# Online learning with dynamics:
# A minimax perspective

**Kush Bhatia**
EECS, UC Berkeley
kush@cs.berkeley.edu

**Karthik Sridharan**
CS, Cornell University
sridharan@cs.cornell.edu

## Abstract

We study the problem of *online learning with dynamics*, where a learner interacts with a stateful environment over multiple rounds. In each round of the interaction, the learner selects a policy to deploy and incurs a cost that depends on both the chosen policy and current state of the world. The state-evolution dynamics and the costs are allowed to be time-varying, in a possibly adversarial way. In this setting, we study the problem of minimizing policy regret and provide non-constructive upper bounds on the minimax rate for the problem.

Our main results provide sufficient conditions for online learnability for this setup with corresponding rates. The rates are characterized by: 1) a complexity term capturing the expressiveness of the underlying policy class under the dynamics of state change, and 2) a dynamic stability term measuring the deviation of the instantaneous loss from a certain counterfactual loss. Further, we provide matching lower bounds which show that both the complexity terms are indeed necessary.

Our approach provides a unifying analysis that recovers regret bounds for several well studied problems including online learning with memory, online control of linear quadratic regulators, online Markov decision processes, and tracking adversarial targets. In addition, we show how our tools help obtain tight regret bounds for a new problems (with non-linear dynamics and non-convex losses) for which such bounds were not known prior to our work.

## 1   Introduction

Machine learning systems deployed in the real-world interact with people through their decision making. Such systems form a feedback loop with their environment: they learn to make decisions from real-world data and decisions made by these systems in turn affect the data that is collected. In addition, people often learn to adapt to such automated decision makers in an attempt to maximize their own utility rendering any assumption on the data generation process futile. Motivated by these aspects of decision making, we propose the problem of *online learning with dynamics* which involves repeated interaction between a learner and an environment with an underlying state. The decisions made by the learner affect this state of the environment which evolves as a dynamical system. Further, we place no distributional assumptions on the learning data and allow this to be adversarial.

Given such a setup, a natural question to ask is how does one measure the performance of the learner? Classical online learning studies one such notion of performance known as regret. This measure compares the performance of the learner to that of a fixed best policy in hindsight, when evaluated on the *same* states which were observed by the learner. Such a measure of performance clearly does not work for the above setup: if we would have deployed a different policy, we would have observed different states of the environment. To overcome this, we study a counterfactual notion of regret, called *Policy Regret*, where the comparator term is the performance of a policy on the states one would have observed if this policy was deployed from the beginning of time.

Such a notion of regret has been studied in the online learning literature for understanding memory based adversaries [20, 4, 5] and more recently, for the study of specific reinforcement learning models [11, 1, 10]. However, a vast majority of these works have focused on known and fixed models of state evolution, often restricting the scope to linear dynamical systems. Further, these works have focused on simplistic policy classes as the comparators in their notion of policy regret. Contrast this with the vast literature on statistical learning [28, 6] and classical online learning [22] which study the question of learnability in full generality; for arbitrary losses and general function classes.

Our work is a step towards addressing this gap. We study the problem of learnability for a class of online learning problems with underlying states evolving as a dynamical system in its full generality. Our main results provide sufficient conditions (along with non-asymptotic upper bounds) on when such problems are learnable, that is, can have vanishing policy regret. Our approach is non-constructive and provides a complexity term that provides upper bounds on the minimax rates for these problems. Further, we provide lower bounds showing that for a large class of problems, our upper bounds are tight up to constant factors. By studying the problem in full generality, we show how several well-studied problems in the literature comprising online Markov decision processes [11], online adversarial tracking [1], online linear quadratic regulator [10], online control with adversarial noise [2], and online learning with memory [5, 4] can be seen as specific examples of our general framework. We recover the best known rates for a majority of these problems, often times even generalizing these setups. We also provide examples where, to the best of our knowledge, previous techniques are not able to obtain useful bounds on regret; however using our minimax tools, we are able to provide tight bounds on the policy regret for these examples.

Formally, we consider the setup where $\mathcal{X}$ denotes an arbitrary set of states, $\Pi$ an arbitrary class of policies and $\mathcal{Z}$ an arbitrary instance space. Given this, the interaction between the learner and nature can be expressed as a $T$ round protocol where on each round $t \in [T]$, the learner picks a policy $\pi_t \in \Pi$, the adversary simultaneously picks instance $(z_t, \zeta_t) \in \mathcal{Z}$. The learner suffers loss $\ell(\pi_t, x_t, z_t)$ and the state of the system evolves[1] as $x_{t+1} \leftarrow \Phi(x_t, \pi_t, \zeta_t)$, where $\Phi$ is known to the learner. The goal of the learner is to minimize policy regret

$$\text{Reg}_{\text{T}}^{\text{pol}} = \sum_{t=1}^{T} \ell(\pi_t, x_t, z_t) - \inf_{\pi \in \Pi} \sum_{t=1}^{T} \ell(\pi, x_t[\pi^{(t-1)}, \zeta_{1:t-1}], z_t) \ ,$$

where $x_t$ are the states of the system based on learners choices of policies and $x_t[\pi^{(t-1)}, \zeta_{1:t-1}]$ represents the state of the system at time $t$ if the policy $\pi$ was used the previous $t-1$ rounds. We refer to the loss $\ell(\pi, x_t[\pi^{(t-1)}, \zeta_{1:t-1}], z_t)$ as the *counterfactual loss* of policy $\pi$. Notice that dynamics $\Phi$ being fixed or known in advance to the learner is not really restrictive since an adversary can encode arbitrary state dynamics mapping in $\zeta_t$'s and $\Phi$ can just be seen as an applicator of these mapping.

**Our contributions.** We are interested in the following question: for a given problem instance $(\Pi, \mathcal{Z}, \Phi, \ell)$, is the problem learnable, that is, does there exists a learning algorithm such that policy regret is such that $\text{Reg}_{\text{T}}^{\text{pol}} = o(T)$. Below we highlight some of the key contributions of this paper.

1. We show that the minimax policy regret for any problem specified by $(\Pi, \mathcal{Z}, \Phi, \ell)$ can be upper bounded by sum of two terms: i) a sequential Rademacher complexity like term for the class of counterfactual losses of the policy class, and ii) a term we refer to as dynamic stability term for the Empirical Risk Minimizer (ERM) (or regularized ERM) algorithm.

2. We analyze the problem in the dual game. While in most cases ERM does not even have low classical regret let alone policy regret, we show that ERM like strategy in the dual game can lead to the two term decomposition of minimax policy regret we mention above.

3. Ours is the first work that studies arbitrary online dynamical systems, and provides an analysis for general policy classes and loss functions (possibly non-convex).

4. We provide lower bounds that show that our sufficient conditions are tight for a large class of problem instances showing that both the terms in our upper bounds are indeed necessary.

5. We delineate a number of previously studied problems including online linear quadratic regulator, and online learning with memory for which we recover rates. More importantly, we provide examples of new non-convex and general online learning with dynamics problems and obtain tight regret bounds. For these examples, none of the previous methods are able to obtain any non-degenerate regret bounds.

**Related work.** The classical online learning setup [8] considers a repeated interactive game between a learner and an environment without any notion of underlying dynamics. Sequential complexity measures were introduced in [22] to get tight characterization of minimax regret rates for the classical online learning setting. They showed that for the class of online supervised learning problems, one can upper and lower bound minimax rate in terms of a sequential Rademacher complexity of the predictor class. The works [18, 7] provided an analog of VC theory for online classification and the sequential complexity measures in work [22] provided such a theory for general supervised online learning. This paper can be seen as deriving such characterization of learnability and tight rates for the problem of online learning with dynamics. In the more general setting we consider, while the main mathematical tools introduced in [22] are useful, they are not by themselves sufficient because of the complexities of policy regret and the state dynamics. This is evident from our upper bound which consists of two terms (both of which we show are necessary) and only one of them is a sequential Rademacher complexity type term.

Another line of work closes related ours is that on the theory of optimal control (see [17] for a review). Linear dynamical systems with simple zero mean noise models like Gaussian noise for state dynamics have been extensively studied (see the surveys [19] and [14] for an extensive review). While majority of the work in control have focused on linear dynamics with fixed noise models, $H_\infty$ control (and more generally robust control) literature has aimed at extending the setting to worst case perturbations (see [26]). However these works focus on cumulative costs and are often not practical for machine learning scenarios where such algorithms tend to be overly conservative.

There has been recent work dealing with adversarial costs and linear dynamics with either stochastic or adversarial noise. Online Markov decision processes [11], Online Adversarial Tracking [1] and Online Linear Quadratic Regulator [10, 25] are all examples of such work that deal with specific form of possibly adversarially chosen cost functions, albeit the loss functions in these problems are very specific and the dynamics are basically linear with either fixed stochastic noise or no noise. Perhaps the closest comparison to our work is the work by Agarwal et. al [2] (and also [3, 12]) where adversarial but convex costs, linear policies and linear dynamics with an adversarial component are considered. In contrast, we consider arbitrary class of policies, both adversarially chosen costs (possibly non-convex) and dynamics that are presented on the fly and arbitrary state space.

## 2 Online learning with dynamics

We now formally define the online learning with dynamics problem. We let $\mathcal{X}$ represent the state space, $\Pi$ denote the set of learner polices, and $\mathcal{Z} = \mathcal{Z}_\ell \times \mathcal{Z}_\Phi$ denote the space of adversary's moves.

### 2.1 Problem setup

The problem of online learning with dynamics proceeds as a repeated game between a learner and an adversary played over $T$ rounds. The state of the system at time $t$, denoted by $x_t \in \mathcal{X}$, evolves according to a stochastic dynamical system as $x_{t+1} = \Phi(x_t, \pi_t, \zeta_t) + w_t$, where $\Phi : \mathcal{X} \times \Pi \times \mathcal{Z}_\Phi \to \mathcal{X}$ is the transition function and $w_t \sim \mathcal{D}_w$ is a zero-mean additive noise. The transition function $\Phi$ is allowed to depend on adversary's action $\zeta_t$ allowing the dynamics to change across time steps. We assume that the dynamics function $\Phi$ and distribution $\mathcal{D}_w$ are fixed apriori and are known to the learner before the game begins.

Given these dynamics, the repeated online game between the learner and the adversary starts at an initial state $x_1$ proceeds via the following interactive protocol:

On round $t = 1, \ldots, T$,

- the learner picks policy $\pi_t \in \Pi$, adversary simultaneously selects instance $(z_t, \zeta_t) \in \mathcal{Z}$
- the learner receives payoff (loss) signal $\ell(\pi_t, x_t, z_t)$
- the state of the system transitions to $x_{t+1} = \Phi(x_t, \pi_t, \zeta_t) + w_t$

We consider the *full information* version of the above game: the learner gets to observe the instances $(z_t, \zeta_t)$ at time $t$. The objective of the learner is to minimize, in expectation, the *policy regret*

$$\text{Reg}_T^{\text{pol}} = \sum_{t=1}^{T} \mathbb{E}_w[\ell(\pi_t, x_t[\pi_{1:t-1}, w_{1:t-1}, \zeta_{1:t-1}], z_t)] - \inf_{\pi \in \Pi} \mathbb{E}_w\left[\sum_{t=1}^{T} \ell(\pi, x_t[\pi^{(t-1)}, w_{1:t-1}, \zeta_{1:t-1}], z_t)\right] \quad (1)$$

with respect to a policy class $\Pi$ and dynamics model $\Phi$. In the above definition, the notation $x_t[\pi_{1:t-1}, w_{1:t-1}, \zeta_{1:t-1}]$ makes the dependence of the state $x_t$ explicit on the previous policies, noise and adversarial actions. For notational convenience, we will often drop dependencies on the noise variables $w$ and adversarial actions $\zeta$ when it is clear from context.

Observe that in the definition of policy regret, the loss depends on the state of the system at that instance which can be potentially very different for the learner and a comparator policy $\pi$. This lends an additional source of complexity to the interactive game, and can make the problem *much* harder than its counterpart without a dynamics.

The problem of online learning with dynamics generalizes the online learning problem where the loss functions $\ell(\pi, x, z) = \tilde{\ell}(\pi, z)$ are independent of the underlying state variables. Indeed, our notion of policy regret in equation (1) reduces to the notion of external regret studied in the online learning literature. Also, the problem of online learning with memory involves adversaries which have bounded memory of length $m$ and thus the loss incurred by the learner at any time is a function of its past $m$ moves. By setting the state variable $x_t = [\pi_{t-m}, \ldots, \pi_{t-1}]$, the dynamics function $\Phi(x_t, \pi_t) = [\pi_{t-m+1}, \ldots, \pi_t]$, and the noise disturbances $w_t = 0$, we can see that the bounded memory adversaries can be seen as a special case of our problem with dynamics.

## 2.2 Minimax Policy Regret

Given the setup of the previous section, we study the online learning with dynamics game between the learner and the adversary through a minimax perspective. Studying this minimax value allows one to understand the limits of learnability for a tuple $(\Pi, \mathcal{Z}, \Phi, \ell)$: upper bounds on this value imply existence of algorithms with corresponding rates while lower bounds on this values represent the information-theoretic limits of learnability. In the following lemma, we formally define the value $\mathcal{V}_T(\Pi, \mathcal{Z}, \Phi, \ell)$ of the minimax policy regret for a given problem which informally is the policy regret of best learning algorithm against the worst case adversary.

**Proposition 1** (Value $\to$ Dual Game). *Let $\mathcal{Q}$ and $\mathcal{P}$ denote the sets of probability distributions over the policy class $\Pi$ and the adversarial actions $\mathcal{Z}$ respectively, satisfying the necessary conditions for the minimax theorem to hold. Then, we have that*[2]

$$\mathcal{V}_T(\Pi, \mathcal{Z}, \Phi, \ell) := \left\langle\!\!\!\left\langle \inf_{q_t \in \mathcal{Q}} \sup_{(z_t, \zeta_t) \in \mathcal{Z}} \mathbb{E}_{\pi_t \sim q_t} \right\rangle\!\!\!\right\rangle_{t=1}^T \left[ \mathrm{Reg}_{\mathrm{T}}^{\mathrm{pol}} \right] = \left\langle\!\!\!\left\langle \sup_{p_t \in \mathcal{P}} \inf_{\pi_t} \mathbb{E}_{(z_t, \zeta_t) \sim p_t} \right\rangle\!\!\!\right\rangle_{t=1}^T \left[ \mathrm{Reg}_{\mathrm{T}}^{\mathrm{pol}} \right]. \tag{2}$$

The proof of the proposition is deferred to Appendix A. The proof proceeds via a repeated application of von Neumann's minimax theorem (for instance see [23, Appendix A]). Notice that the minimax theorem changes the order of the online sequential game defined in the setup above: at every time step $t$, the adversary proceeds first and outputs a distribution $p_t$ over instances and the learner responds back with $\pi_t$ *after* having observed the distribution. The actual loss instance $(z_t, \zeta_t)$ is then sampled from the revealed distribution $p_t$. On the other hand, the comparator remains the same as before: the best policy $\pi \in \Pi$ in hindsight. This reversed game, termed the *Dual Game*, forms the basis of our analysis and allows us to study the complexity of the online learning with dynamics problem.

## 3 Upper bounds on value of the game

Our main result in this section concerns an upper bound on the value of the sequential game $\mathcal{V}_T(\Pi, \mathcal{Z}, \Phi, \ell)$ relating it to the study of certain stability properties of empirical minimizers and stochastic processes associated with them. Before we proceed to describe the main result, we revisit some preliminaries and setup notation which would be helpful in describing the main result.

**Sequential Rademacher complexity.** The notion of Sequential Rademacher Complexity, introduced in [22], is a natural generalization of the Rademacher complexity for online learning. However, observe that the loss of the comparator term in the definition of policy regret in equation (1) depends on the adversarial actions $\zeta_{1:t-1}$ through the dynamics and $z_t$ through the loss function $\ell$. We define the following version of sequential Rademacher complexity for such dynamics based losses.

**Definition 1.** *The Sequential Rademacher Complexity of a policy class* $\Pi$ *with respect to loss function* $\ell : \Pi \times \mathcal{X} \times \mathcal{Z}_\ell \mapsto \mathbb{R}$ *and dynamics* $\Phi : \mathcal{X} \times \Pi \times \mathcal{Z}_\Phi \to \mathcal{X}$ *is defined as*

$$\mathfrak{R}_T^{\mathsf{seq}}(\ell \circ \Pi) := \sup_{(\mathbf{z}, \boldsymbol{\zeta})} \mathbb{E}_\epsilon \left[ \sup_{\pi \in \Pi} \sum_{t=1}^T \epsilon_t \ell(\pi, x_t[\boldsymbol{\zeta}_1(\epsilon), \ldots, \boldsymbol{\zeta}_{t-1}(\epsilon)], \mathbf{z}_t(\epsilon)) \right] ,$$

*where the outer supremum is taken over* $\mathcal{Z} = \mathcal{Z}_\ell \times \mathcal{Z}_\Phi$-*valued trees* [3] *of depth* $T$ *and* $\epsilon = (\epsilon_1, \ldots, \epsilon_T)$ *is a sequence of i.i.d. Rademacher random variables.*

A similar definition was also used by Han et al. [13] in the context of online learning with strategies where the notion of regret was defined w.r.t. a set of strategies rather than a fixed action. As compared with the classical online learning problem, the above comprises problems where the loss at time $t$ depends on the complete history $(\zeta_1, \ldots, \zeta_{t-1})$ of the adversarial choices along with $z_t$. As noted by [13], such dependence on the the adversary's history can often make the online learning problem harder to learn compared with the online learning problem.

**Empirical Risk Minimization (ERM).** Given a sequence of loss functions $\ell_t : \mathcal{F} \mapsto \mathbb{R}$ for $t \in [T]$, the ERM with respect to a function class $\mathcal{F}$ is defined to be the minimizer of the cumulative loss with $f_{\mathsf{ERM}, T} \in \mathrm{argmin}_{f \in \mathcal{F}} \sum_{t=1}^T \ell_t(f)$. In the statistical learning setup, the problems of supervised classification and regression are known to be learnable with respect to a function class $\mathcal{F}$ *if and only if* the empirical risks uniformly converge over this class $\mathcal{F}$ to the population risks. In contrast, our results provide *sufficient* conditions for learnability in terms of certain stability properties of such empirical risk minimizers.

**Dynamic stability.** We introduce the notion of dynamic stability which captures the stability of an algorithm's interaction with the underlying dynamics $\Phi$. In order to do so, we define a notion of counterfactual loss $\ell_t^\Phi$ of a policy $\pi$ as the loss incurred by a learner which selects $\pi$ for time $1 : t$.

**Definition 2** (Counterfactual Losses). *Given a sequence of adversarial actions* $\zeta_{1:t-1}, z_t$, *dynamics function* $\Phi$, *and noise distribution* $\mathcal{D}_w$, *the counterfactual loss of a policy* $\pi$ *at time* $t$ *is*

$$\ell_t^\Phi(\pi, \zeta_{1:t-1}, z_t) := \mathbb{E}_{w_s \sim \mathcal{D}_w} \left[ \ell(\pi, x_t[\pi^{(t-1)}, w_{1:t-1}, \zeta_{1:t-1}], z_t) \right] .$$

With this definition, observe that the comparator term in the value $\mathcal{V}_T$ in equation (2) is in fact a cumulative sum of counterfactual losses for a policy $\pi$. Any algorithm $\mathcal{A}$ that plays a sequence of policies $\{\pi_t\}$ in the online game incurs an instantaneous loss $\ell(\pi_t, x_t[\pi_{1:t-1}, \zeta_{1:t-1}], z_t)$ at time $t$. In comparison, the counterfactual loss $\ell^\Phi(\pi_t, \zeta_{1:t-1}, z_t)$ represents a scenario where the algorithm commits to the policy $\pi_t$ from the beginning of the game. Our notion of dynamic stability of an algorithm is precisely the deviation between these two types of losses: instantaneous and counterfactual.

**Definition 3** (Dynamic Stability). *An algorithm* $\mathcal{A}$ *is said to be* $\{\beta_t\}$-*dynamically stable if for all sequences of adversarial actions* $[(z_1, \zeta_1), \ldots, (z_T, \zeta_T)]$ *and time instances* $t \in [T]$

$$\left| \mathbb{E}_{w_{1:t-1}} [\ell(\pi_t, x_t[\pi_{1:t-1}, w_{1:t-1}, \zeta_{1:t-1}], z_t)] - \ell^\Phi(\pi_t, \zeta_{1:t-1}, z_t) \right| \le \beta_t \quad where \quad \pi_t = \mathcal{A}((z_{1:t-1}, \zeta_{1:t-1})).$$

It is interesting to note that if that loss functions are independent of the underlying states, that is $\ell(\pi, x, z) = \tilde{\ell}(\pi, z)$, then *any* algorithm is dynamically stable in a trivial manner with the stability parameters $\beta_t = 0$ for all time instances $t$.

With these definitions, we now proceed to describe our main result. Recall that Proposition 1 translates the problem of studying the value of the game $\mathcal{V}_T(\Pi, \mathcal{Z}, \Phi, \ell)$ to that of studying the policy regret in a dual game. In this dual game, the learner has access to the set of adversaries distribution $\{p_s\}_{s=1}^t$ at time $t$ and the policy $\pi_t$ can be a function of these. For a regularization function $\Omega : \Pi \mapsto \mathbb{R}_+$, we denote the regularized ERMs with respect to function class $\Pi$ and counterfactual losses $\ell^\Phi$ by

$$\pi_{\mathsf{RERM}, t} \in \mathrm{argmin}_{\pi \in \Pi} \sum_{s=1}^t \mathbb{E}_{z_s} \left[ \ell^\Phi(\pi, \zeta_{1:s-1}, z_s) \right] + \lambda \cdot \Omega(\pi) , \tag{3}$$

where $\lambda \ge 0$ is the regularization parameter. The following theorem provides an upper bound on the value $\mathcal{V}_T$ in terms of the dynamic stability parameters of the regularized ERMs above as well a sequential Rademacher complexity of the *effective* loss class $\ell^\Phi \circ \Pi := \{ \ell^\Phi(\pi, \cdot) : \pi \in \Pi \}$.

**Theorem 1** (Upper bound on value). *For any online learning with dynamics instance $(\Pi, \mathcal{Z}, \Phi, \ell)$, consider the set of regularized ERMs given by eq. (3) with regularization function $\Omega$ and parameter $\lambda \geq 0$ having dynamic stability parameters $\{\beta_{\mathsf{RERM},t}\}_{t=1}^{T}$. Then, we have that the value of the game*

$$\mathcal{V}_T(\Pi, \mathcal{Z}, \Phi, \ell) \leq \sum_{t=1}^{T} \beta_{\mathsf{RERM},t} + 2\mathfrak{R}_T^{\mathsf{seq}}(\ell^\Phi \circ \Pi) + 2\lambda \cdot \sup_{\pi \in \Pi} \Omega(\pi) \; . \tag{4}$$

The complete proof of the above theorem can be found in Appendix B. A few comments on Theorem 1 are in order. The theorem provides sufficient conditions to ensure learnability of the online learning with dynamics problem. In particular, the two terms Term (I) = $\sum_{t=1}^{T} \beta_{\mathsf{RERM},t}$ and Term (II) = $\mathfrak{R}_T^{\mathsf{seq}}(\ell^\Phi \circ \Pi)$ contain the main essence of the upper bound. Term (I) concerns the dynamic mixability property of the regularized ERM in the dual game. If there exist *approximate* minimizers (regularized) of the sequence of counterfactual losses within the policy class $\Pi$ such that $\pi_{\mathsf{RERM},t}$ is uniformly close to $\pi_{\mathsf{RERM},t+1}$ the dynamic stability parameters can be made to be small. Term (II) comprises of the sequential Rademacher complexity of the loss class $\ell^\Phi \circ \Pi$ which involves the underlying policy class $\Pi$ as well as the counterfactual loss $\ell^\Phi$. This measure of complexity can be seen as one which corresponds to an effective online game where the the loss at time $t$ depends on the adversarial actions up to time $t$. Compare this to the instantaneous loss $\ell(\pi_t, x_t[\pi_{1:t-1}, \zeta_{1:t-1}], z_t)$ which depends on both the policies as well as the adversarial actions up to time $t$. Observe that for the classical online learning setup without dynamics, the dynamic stability parameters $\beta_{\mathsf{RERM},t} \equiv 0$. On setting the value of regularization parameter $\lambda = 0$, we recover back the learnability result of Rakhlin et al. [22].

We would like to highlight that the complexity-based learnability guarantees of Theorem 1 are non-constructive in nature. In particular, the theorem says that any non-trivial upper bounds on the stability and sequential complexity terms would guarantee the *existence* of an online learning algorithm with the corresponding policy regret. Our minimax perspective on the problem allows us to study the problem in full generality without making assumptions with respect to the policy class $\Pi$, adversarial actions $\mathcal{Z}$ and the underlying (possibly adversarial) dynamics $\Phi$, and provide sufficient conditions for learnability.

Given the upper bound on the value $\mathcal{V}_T(\Pi, \mathcal{Z}, \Phi, \ell)$, one can observe that there is a possible tension between the two complexity terms: while dynamic stability term promotes using policies which are "similar" across time steps, the regularized complexity term seeks policies which are minimizers of cumulative losses and might vary across time steps. In order to balance similar trade-offs, a natural *Mini-Batching Algorithm* has been proposed in various works on online learning with memory [5] and online learning with switching costs [9]. The key idea is that the learner divides the time $T$ into intervals of length $\tau > 0$ and commits to playing the same strategy over this time period.

Let us denote any such mini-batching algorithm by $\mathcal{A}_\tau$ and the corresponding minimax value restricted to this class of algorithms by $\mathcal{V}_{T,\tau}(\Pi, \mathcal{Z}, \Phi, \ell)$ where the infimum in equation 2 is taken over all mini-batching algorithms $\mathcal{A}_\tau$. Similar to the regularized ERM of equation (3), we define the following mini-batched ERMs:

$$\pi_{\mathsf{ERM},t}^\tau = \begin{cases} \pi_{\mathsf{ERM}}(t) & \text{for } t \equiv 0 \bmod \tau \\ \pi_{\mathsf{ERM}}(\tau\lfloor \frac{t}{\tau} \rfloor) & \text{otherwise} \end{cases} , \tag{5}$$

where we have used the notation $\pi_{\mathsf{ERM}}(t) := \pi_{\mathsf{ERM},t}$. In the following proposition, we prove an upper bound analogous to that of Theorem 1 for this class of mini-batching algorithms[4].

**Proposition 2** (Mini-batching algorithms.). *For any online learning with dynamics game $(\Pi, \mathcal{Z}, \Phi, \ell)$, consider the set of mini-batch ERMs given by equation (5) having dynamic stability parameters $\{\beta_{\mathsf{ERM},t}^\tau\}_{t=1}^{T}$. Then, we have that the value of the game*

$$\mathcal{V}_T(\Pi, \mathcal{Z}, \Phi, \ell) \leq \inf_{\tau > 0} \mathcal{V}_{T,\tau}(\Pi, \mathcal{Z}, \Phi, \ell) \leq \inf_{\tau > 0} \left( \sum_{t=1}^{T} \beta_{\mathsf{ERM},t}^\tau + 2\tau \cdot \sup_{s \in [\tau]} \mathfrak{R}_{T/\tau}^{\mathsf{seq}}(\ell_s^\Phi \circ \Pi) \right) \; , \tag{6}$$

*where $\ell_s^\Phi$ is the counterfactual loss for the $s^{th}$ batch.*

We defer the proof of the above proposition to Appendix B. In comparison with the upper bound of Theorem 1, this bound concerns the dynamic stability of the mini-batched ERMS as compared to

their regularized counterparts. Often times, obtaining bounds on the stability parameters $\{\beta_{\mathsf{ERM},t}^\tau\}_{t=1}^T$ can be much easier than the ones for regularized ERMS. For instance, it is easy to see that for the problem of online learning with memory with adversaries having memory $m$, one can bound $\sum_{t=1}^T \beta_{\mathsf{ERM},t}^\tau = O(\frac{mT}{\tau})$ whenever the losses are bounded, providing a natural trade-off between the two complexity terms.

## 4   Lower bounds on value of the game

Having established sufficient conditions for the learnability of the online learning with dynamics problem in the previous section, we now turn to address the optimality of these conditions. Recall that Theorem 1 and Proposition 2 established upper bounds on the value $\mathcal{V}_T(\Pi, \mathcal{Z}, \Phi, \ell)$ for instances of our problem. The following theorem shows that both the upper bounds of equations (4) and (6) are indeed tight upto constant factors.

**Theorem 2** (Lower Bound). *For the online learning with dynamics problem, there exist problem instances $\{(\Pi, \mathcal{Z}, \Phi, \ell_i)\}_{i=1}^3$, a regularization function $\Omega$ and a universal constant $c > 0$ such that*

$$\mathcal{V}_T(\Pi, \mathcal{Z}, \Phi, \ell_1) \geq c \cdot \mathfrak{R}_T^{\mathsf{seq}}(\ell_1^\Phi \circ \Pi) \tag{7a}$$

$$\mathcal{V}_T(\Pi, \mathcal{Z}, \Phi, \ell_2) \geq c \cdot \inf_{\lambda > 0}\left( \sum_{t=1}^T \beta_{\mathsf{RERM},t} + \lambda \cdot \sup_{\pi \in \Pi} \Omega(\pi) \right) \tag{7b}$$

$$\mathcal{V}_T(\Pi, \mathcal{Z}, \Phi, \ell_3) \geq c \cdot \inf_{\tau > 0}\left( \sum_{t=1}^T \beta_{\mathsf{ERM},t}^\tau + 2\tau \mathfrak{R}_{T/\tau}^{\mathsf{seq}}(\ell_3^\Phi \circ \Pi) \right), \tag{7c}$$

*where $\beta_{\mathsf{RERM},t}$ and $\beta_{\mathsf{ERM},t}^\tau$ are the dynamic mixability parameters of the regularized ERM w.r.t. $\ell_2$ (eq. (3)) and mini-batching ERM w.r.t. $\ell_3$ (eq. (5)) respectively.*

A few comments on Theorem 2 are in order. The theorem exhibits that the sufficiency conditions from Theorem 1 and Proposition 2 are indeed necessary by exhibiting instances whose value is lower bounded by these terms. In particular, equation (7a) shows that the sequential Rademacher term is necessary, (7b) establishes necessity for the dynamic stability of the regularized ERM, while (7c) shows that the mini-batching upper bound is also tight. It is worth noting that these lower bounds are not instance dependent but rather construct specific examples to demonstrate the tightness of our upper bound from the previous section. We next present the key idea for the proof of the theorem and defer the complete details to Appendix C.

**Proof sketch.**   We now describe the example instances which form the crux of the proof for Theorem 2. Consider the online learning with dynamics game between a learner and an adversary with the state space $\mathcal{X} = \{x \in \mathbb{R}^d \mid \|x\|_2 \leq 1\}$ and the set of adversarial actions $\mathcal{Z}_\ell^{\mathsf{Lin}} = \{z \in \mathbb{R}^d \mid \|z\|_2 \leq 1\}$. Further, we consider the constant policy class $\Pi_{\mathsf{Lin}} = \{\pi_f \mid \pi(x) = f \text{ for all states } x \text{ with } f \in \mathbb{B}_d(1)\}$, consisting of policies $\pi_f$ which select the same action $f$ at each state $x$. With a slight abuse of notation, we represent the policy $\pi_t$ played by the learner at time by the corresponding $d$-dimensional vector $f_t$. Further, we let the dynamics function $\Phi_{\mathsf{Lin}}(x_t, f_t, \zeta_t) = f_t$. We now define the loss function which consists of two parts, a linear loss and a $L$-Lipschitz loss involving the dynamics:

$$\ell_L(f_t, x_t, z_t) = \langle f_t, z_t \rangle + \sigma(f_t, x_t) \quad \text{where} \quad \sigma(f_t, x_t) = \begin{cases} L\|f_t - x_t\|_2 & \text{for } \|f_t - x_t\|_2 \leq \frac{1}{L} \\ 1 & \text{otherwise} \end{cases}. \tag{8}$$

Observe that this example constructs a family of instances one for each value of the Lipschitz constant of $L$ of the function $\sigma$. For this family of instances, we establish that the value

$$\mathcal{V}_T(\Pi_{\mathsf{Lin}}, \mathcal{Z}, \Phi_{\mathsf{Lin}}, \ell_L) \geq \begin{cases} \sqrt{T} & \text{for } 0 < L < 1 \\ \sqrt{LT} & \text{for } 1 \leq L \leq (4T)^{\frac{1}{3}} \\ 2^{\frac{1}{3}} T^{\frac{2}{3}} & \text{for } L > (4T)^{\frac{1}{3}} \end{cases}.$$

The proof finally connects these lower bounds to the upper bounds of Theorem 1 and Proposition 2.♣

With the lower bounds given in Theorem 2, it is natural to ask whether the sufficient conditions in Theorem 1 and Proposition 2 are indeed necessary for every instance of the online learning with dynamics problem. The answer to this question is unsurprisingly *No* given the generality in which we study this problem. Consider the following simple instance of the problem:

$$\Pi = \mathcal{X}, \quad \ell(\pi, x, z) = \tilde{\ell}(\pi, z) + \mathbb{I}[\pi = x], \quad \text{and} \quad x_{t+1} = \pi_t,$$

for any non-negative bounded loss $\tilde{\ell}(\pi, z) \in [0, 1]$ for all $\pi \in \Pi$, $z \in \mathcal{Z}_\ell$. Consider any policy class for which $\mathfrak{R}_T^{\mathsf{seq}}(\ell^\Phi \circ \Pi) > 0$. Both bounds (4) and (6) suggest that the problem is learnable with rate at least $\mathfrak{R}_T^{\mathsf{seq}}(\ell^\Phi \circ \Pi)$. However, observe that the indicator term in the loss is quite severe on the comparator; it ensures that the comparator term is at least $T$. Thus, *any* algorithm which selects a policy from $\Pi$ at every instance can ensure that the policy regret is at most 0! While the above example establishes that the sufficient conditions are not necessary in an instance dependent manner, our next proposition establishes that they are indeed tight for large class of problems instances.

**Proposition 3** (Instance-dependent lower bound). *a) Given any online learning problem $(\mathcal{F}, \mathcal{Z}_\ell, \ell)$ with a bounded loss function $\ell : \mathcal{F} \times \mathcal{Z}_\ell \mapsto [-1, 1]$, there exists an online learning with dynamics problem $(\Pi_\mathcal{F}, \mathcal{Z}_\ell \times \{-1, 1\}, \Phi, \tilde{\ell})$ and a universal constant $c > 0$ such that*

$$\mathcal{V}_T(\Pi_\mathcal{F}, \mathcal{Z}_\ell \times \{-1, 1\}, \Phi, \tilde{\ell}) \geq c \cdot \inf_{\tau > 0}\left(\sum_{t=1}^T \beta_{\mathsf{ERM}, t}^\tau + 2\tau \mathfrak{R}_{T/\tau}^{\mathsf{seq}}(\ell^\Phi \circ \Pi)\right),$$

*where $\beta_{\mathsf{ERM}, t}^\tau$ are the dynamic mixability parameters of the mini-batching ERM w.r.t. $\ell$ (eq. (5)).*

*b) Given a policy class $\Pi$ and dynamics function $\Phi$, there exists an online learning with dynamics problem $(\Pi, \mathcal{Z}, \Phi, \ell)$ and a universal constant $c > 0$ such that*

$$\mathcal{V}_T(\Pi, \mathcal{Z}, \Phi, \ell) \geq c \cdot \mathfrak{R}_T^{\mathsf{seq}}(\ell^\Phi \circ \Pi).$$

We defer the proof of the proposition to Appendix C. This proposition can be seen as a strengthening of the lower bounds (7a) and (7c) showing that for a very large class of problems, the upper bound given by the mini-batching algorithm and the sequential complexity terms are in fact necessary.

## 5    Examples

In this section, we look at specific examples of the online learning with dynamics problem and obtain learnability guarantees for these instances using our upper bounds from Theorem 1. For clarity of exposition, our focus in this section on the scaling of the value $\mathcal{V}_T(\Pi, \mathcal{Z}, \Phi, \ell)$ with the time horizon $T$. The proofs in Appendix D explicitly detail out all the problem dependent parameters.

**Example 1: Online Isotron with dynamics.** Single Index Models (SIM) are class of semi-parametric models widely studied in the econometric and operations research community. Kalai and Sastry [16] introduced the Isotron algorithm for learning SIMs and Rakhlin et al [23] established that the online version of this problem is learnable. Here, we introduce a version of this problem with a state variable that requires a component of the model to vary slowly across time:

$$\mathcal{X} = \mathbb{R}, \ \mathcal{F} = \{f = (\sigma, \mathbf{w} = (w_1, w) \mid \sigma : [-1, 1] \mapsto [-1, 1] \text{ 1-Lipschitz}, \ \mathbf{w} \in \mathbb{R}^{d+1} \ |w_1| \leq 1, \|w\|_2 \leq 1\},$$

$$\Pi_\mathcal{F} = \{\pi_f \mid \pi \in \mathcal{F}, \ \pi_f(x) = f \text{ for all } x \in \mathcal{X}\}, \ \mathcal{Z}_\ell = [-1, 1]^{d+1} \times [-1, 1], \ \mathcal{Z}_\Phi = \phi,$$

$$\ell(\pi_f, x, z = (z_1, \mathbf{x}, y)) = (y - \sigma(\langle \mathbf{x}, w \rangle))^2 + (z_1 - w_1)^2 + (x - w_1)^2, \ \Phi(x, \pi_{(\sigma, w)}, \zeta) = w_1, \ \mathcal{D}_w = 0. \quad (9)$$

The following corollary establishes the learnability of the online Isotron problem with dynamics.

**Corollary 1.** *For the online Isotron problem with dynamics given by $(\Pi_\mathcal{F}, \mathcal{Z}, \Phi, \ell)$ in equation (5), we have that the minimax value*

$$\mathcal{V}_{\mathsf{Iso}, T}(\Pi_\mathcal{F}, \mathcal{Z}, \Phi, \ell) = \widetilde{\mathcal{O}}(\sqrt{T}).$$

It is important to note that as with the online learning problem [23], it is not clear whether a computationally efficient method attaining the above guarantee exists.                                              ♣

**Example 2: Online Markov decision processes [11].** This example considers the problem of Online Markov Decision Processes (MDPs) studied in Even-Dar et al. [11]. The setup consists of a finite state space $|\mathcal{X}| = S$, a finite action space $|\mathcal{U}| = A$, and

$$\mathcal{Z}_\ell = \{z \mid z \in [0, 1]^{S \times A}\}, \ \mathcal{Z}_\Phi = \phi, \ \Pi_{\mathsf{MDP}} = \{\pi \mid \pi : \mathcal{X} \mapsto \Delta(\mathcal{U})\}, \ \ell(\pi, x, z) = z(x, \pi(x)),$$

$$\Phi \text{ given by } P : \mathcal{X} \times \mathcal{U} \mapsto \Delta(\mathcal{U}) \text{ with } x' \sim P(x, \pi(x)). \quad (10)$$

With this setup, we now provide a bound on the minimax value $\mathcal{V}_T^{\mathsf{MDP}}$, assuming, as in [11], that the underlying MDP is unichain and satisfies a mixability assumption (see Appendix D for details).

**Corollary 2.** *For the online MDP problem given by $(\Pi_{\mathsf{MDP}}, \mathcal{Z}, \Phi, \ell)$ in equation (5), we have that the minimax value*

$$\mathcal{V}_{\mathsf{MDP}, T}(\Pi_{\mathsf{MDP}}, \mathcal{Z}, \Phi, \ell) = \mathcal{O}(\sqrt{T}).$$

The above corollary helps one recover the same $\mathcal{O}(\sqrt{T})$ regret bound that was obtained by [11].     ♣

**Example 3: Online Linear Quadratic Regulator [10].** The online Linear Quadratic Regulator (LQR) setup studied in this section was first studied in Cohen et al. [10]. The setup consists of a LQ system - with linear dynamics and quadratic costs - where the cost functions are chosen adversarially. The comparator class consists of a set of strongly stable linear policies (see Appendix D).

$$\mathcal{X} = \mathbb{R}^d, \ \mathcal{Z}_\ell = \{(Q,R) \mid Q,R > 0, \ \text{tr}(Q), \text{tr}(R) \le C\}, \ \Pi_{\mathsf{LQR}} = \{K \in \mathbb{R}^{k \times d} \mid K \text{ is } (\kappa, \gamma) - \text{strongly stable}\},$$

$$\mathcal{Z}_\Phi = \phi, \ \ell(\pi, x, z) = x^\top Q x + (Kx)^\top R(Kx), \ \Phi(x, K) = (A + BK)x, \ \mathcal{D}_w = \mathcal{N}(0, I). \quad (11)$$

With this setup, we now establish the learnability of this problem in the following corollary.

**Corollary 3.** *For the online LQR problem given by* $(\Pi_{\mathsf{LQR}}, \mathcal{Z}, \Phi, \ell)$ *in equation* (5)*, we have that the minimax value*

$$\mathcal{V}_{\mathsf{LQR},T}(\Pi_{\mathsf{LQR}}, \mathcal{Z}, \Phi, \ell) = \widetilde{\mathcal{O}}(\sqrt{T}).$$

Note that [10] obtained a similar policy regret bound of $\mathcal{O}(\sqrt{T})$ but their analysis only worked for an oblivious adversary whereas the guarantee of Corollary 3 holds for an adaptive adversary. ♣

**Example 4: Online Control with Adversarial Disturbances [2].** In this example, we consider a simplified version of the setup from Agarwal et al. [2] where the adversary is allowed to perturb the dynamics at each time instance along with the loss functions. We consider the LQ version of the problem where the dynamics are linear and the cost functions quadratic. Similar to the example above, the comparator class consists of a set of strongly stable linear policies (see Appendix D).

$$\mathcal{X} = \mathbb{R}^d, \ \mathcal{Z}_\ell = \{(Q,R) \mid Q,R > 0, \ \text{tr}(Q), \text{tr}(R) \le C\}, \ \Pi_{\mathsf{LQR}} = \{K \in \mathbb{R}^{k \times d} \mid K \text{ is } (\kappa, \gamma) - \text{strongly stable}\},$$

$$\mathcal{Z}_\Phi = \{\zeta \mid \|\zeta\|_2 \le W\}, \ \ell(\pi, x, z) = x^\top Q x + (Kx)^\top R(Kx), \ \Phi(x, K, \zeta) = (A + BK)x + \zeta, \ \mathcal{D}_w = \phi. \quad (12)$$

With this setup, we now establish the learnability of this problem in the following corollary.

**Corollary 4.** *For the Online Control with Adversarial Disturbances problem instance* $(\Pi_{\mathsf{LQR}}, \mathcal{Z}, \Phi, \ell)$ *detailed in equation* (5)*, we have that the minimax value*

$$\mathcal{V}_{\mathsf{adv},T}(\Pi_{\mathsf{LQR}}, \mathcal{Z}, \Phi, \ell) = \widetilde{\mathcal{O}}(\sqrt{T}).$$

In contrast to Example 3 above, the disturbances in the dynamics are actually controlled by the adversary instead of being random. Our result above shows that this harder version of the problem is also learnable at $\sqrt{T}$ rate. ♣

In addition to these four examples, in Appendix D we consider additional examples of the framework including online adversarial tracking [1] and a non-linear generalization of the LQ problem [10, 2] which we call online non-linear control.

## Broader Impact

This work is mainly theoretical in nature and hopes to provide theoretical foundations for learning under dynamical systems. The work is expected to have a broader impact in the future by opening up research on learning and non-linear dynamical systems with complex policy classes. In the future, we hope that our work will enable ML systems to be deployed reliably in more reactive environments.

## Acknowledgments and Disclosure of Funding

We would like to thank Dylan Foster, Mehryar Mohri and Ayush Sekhari for helpful discussions. KB is supported by a JP Morgan AI Fellowship. KS would like to acknowledge NSF CAREER Award 1750575 and Sloan Research Fellowship

## Footnotes

[1]while we consider deterministic dynamics here, Section 2 considers general dynamics with stochastic noise

[2]$\langle\!\langle \ldots \rangle\!\rangle_{t=1}^T$ denotes interleaved application of the sequence of operator inside. For example, for $T = 2$, $\langle\!\langle \sup_{p_t} \inf_{q_t} \rangle\!\rangle_{t=1}^2 [\cdot] = \sup_{p_1} \inf_{q_1} \sup_{p_2} \inf_{q_2} [\cdot]$

[3] A $\mathcal{Z}$-valued tree $\mathbf{z}$ of depth $d$ is defined as a sequence $(\mathbf{z}_1, \ldots, \mathbf{z}_d)$ of mappings $\mathbf{z}_t : \{\pm 1\}^{t-1} \mapsto \mathcal{Z}$ (see [21])

[4]For this class of mini-batching algorithms, we consider an oblivious adversary which cannot adapt to the randomness of the learner.

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
