[Supplementary Material]

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

[5] we supress the dependence of the state $x_t[\pi_{1:t-1}, \zeta_{1:t-1}]$ on the random noise $w_{1:t-1}$.

[6]We assume $T/\tau$ to be an integer; if not, it affects the bound by an additive factor of $\tau$.

[7]Assume $L$ to be an integer; if not, redefine $L = \lfloor L \rfloor$.

[8]since the dynamics are independent of the adversary, we have added an additional time index $t$ to make explicit the number of times policy $\pi$ is run in the environment.

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

## A  Proof of Proposition 1

The minimax value of the policy regret for the online learning with dynamics protocol is acheieved when at every time $t$, the learner picks the best distribution $q_t$, the adversary picks the worst-case $z_t$ and a sample of policy $\pi_t$ is then drawn from $q_t$. This can be succinctly represented as a sequence of infimum, suprememum and expectations as

$$
\begin{aligned}
\mathcal{V}_T(\Pi, \mathcal{Z}, \Phi, \ell) &= \left\langle\!\!\left\langle \inf_{q_t \in \mathcal{Q}} \sup_{(z_t, \zeta_t) \in \mathcal{Z}} \mathbb{E}_{\pi_t \sim q_t} \right\rangle\!\!\right\rangle_{t=1}^{T-1} \inf_{q_T \in \mathcal{Q}} \sup_{(z_T, \zeta_T) \in \mathcal{Z}} \mathbb{E}_{\pi_T \sim q_T} \left[ \mathrm{Reg}_T^{\mathsf{pol}} \right] \\
&\overset{(i)}{=} \left\langle\!\!\left\langle \inf_{q_t \in \mathcal{Q}} \sup_{(z_t, \zeta_t) \in \mathcal{Z}} \mathbb{E}_{\pi_t \sim q_t} \right\rangle\!\!\right\rangle_{t=1}^{T-1} \sup_{p_T \in \mathcal{P}} \inf_{\pi_T} \mathbb{E}_{(z_T, \zeta_T) \sim p_T} \left[ \mathrm{Reg}_T^{\mathsf{pol}} \right] \\
&\overset{(ii)}{=} \left\langle\!\!\left\langle \sup_{p_t \in \mathcal{P}} \inf_{\pi_t} \mathbb{E}_{(z_t, \zeta_t) \sim p_t} \right\rangle\!\!\right\rangle_{t=1}^{T} \left[ \mathrm{Reg}_T^{\mathsf{pol}} \right],
\end{aligned}
$$

where (i) follows from an application of the von Neumann's minimax theorem for the distributions $\mathcal{Q}$ and $\mathcal{P}$ (see [23, Appendix A]) and (ii) follows from repeatedly performing the same step for $t = \{1, \ldots, T-1\}$. This establishes the desired claim. $\qquad\square$

## B  Proofs of upper bounds

### B.1  Proof of Theorem 1

Recall from equation (3) that the dual regularized ERM for a regularization function $\Omega$ and parameter $\lambda \geq 0$ is

$$
\pi_{\mathsf{RERM}, t} \in \operatorname*{argmin}_{\pi \in \Pi} \sum_{s=1}^{t} \mathbb{E}_{z_s} \left[ \ell^\Phi(\pi, \zeta_{1:s-1}, z_s) \right] + \lambda \cdot \Omega(\pi).
$$

The proof of our main result relies on the following intermediate result which relates the performance of the above RERM with that of any policy $\pi \in \Pi$ when compared on the counterfactual losses $\ell^\Phi$.

**Lemma 1.** *For any policy $\pi \in \Pi$ and any sequence of distributions $\{p_t\}_{t=1}^{T}$ over instance space $\mathcal{Z}$, we have*

$$
\sum_{t=1}^{T} \mathbb{E}_{z_t} \left[ \ell^\Phi(\pi_{\mathsf{RERM}, t}, \zeta_{1:t-1}, z_t) \right] \leq \sum_{t=1}^{T} \mathbb{E}_{z_t} \left[ \ell^\Phi(\pi, \zeta_{1:t-1}, z_t) \right] + \lambda \cdot (\Omega(\pi) - \Omega(\pi_{\mathsf{RERM}, 1})). \tag{13}
$$

Taking this lemma as given, let us proceed to the proof of the theorem statement. For the purpose of this proof, we will use the notation $\hat{\pi}_t := \pi_{\mathsf{RERM}, t}$. Let us begin by considering the value of the game and its equivalence to the dual game established by Proposition 1 as[5]

$$
\begin{aligned}
\mathcal{V}_T(\Pi, \mathcal{Z}, \Phi, \ell) &= \left\langle\!\!\left\langle \sup_{p_t \in \mathcal{P}} \inf_{\pi_t} \mathbb{E}_{(z_t, \zeta_t) \sim p_t} \right\rangle\!\!\right\rangle_{t=1}^{T} \left( \sum_{t=1}^{T} \mathbb{E}[\ell(\pi_t, x_t[\pi_{1:t-1}, \zeta_{1:t-1}], z_t)] - \inf_{\pi \in \Pi} \mathbb{E}_w \left[ \sum_{t=1}^{T} \ell(\pi, x_t[\pi^{(t-1)}, \zeta_{1:t-1}], z_t) \right] \right) \\
&\overset{(i)}{\leq} \left\langle\!\!\left\langle \sup_{p_t \in \mathcal{P}} \mathbb{E}_{(z_t, \zeta_t) \sim p_t} \right\rangle\!\!\right\rangle_{t=1}^{T} \left( \sum_{t=1}^{T} \mathbb{E}[\ell(\hat{\pi}_t, x_t[\hat{\pi}_{1:t-1}, \zeta_{1:t-1}], z_t)] - \mathbb{E}_{\zeta_{1:t-1}, z_t}[\ell^\Phi(\hat{\pi}_t, \zeta_{1:t-1}, z_t)] \right. \\
&\qquad\qquad\qquad \left. \sum_{t=1}^{T} \mathbb{E}_{\zeta_{1:t-1}, z_t}[\ell^\Phi(\hat{\pi}_t, \zeta_{1:t-1}, z_t)] - \inf_{\pi \in \Pi} \mathbb{E}_w \left[ \sum_{t=1}^{T} \ell(\pi, x_t[\pi^{(t-1)}, \zeta_{1:t-1}], z_t) \right] \right) \\
&\overset{(ii)}{\leq} \left\langle\!\!\left\langle \sup_{p_t \in \mathcal{P}} \mathbb{E}_{(z_t, \zeta_t) \sim p_t} \right\rangle\!\!\right\rangle_{t=1}^{T} \left( \sum_{t=1}^{T} \mathbb{E}[\ell(\hat{\pi}_t, x_t[\hat{\pi}_{1:t-1}, \zeta_{1:t-1}], z_t)] - \mathbb{E}_{\zeta_{1:t-1}, z_t}[\ell^\Phi(\hat{\pi}_t, \zeta_{1:t-1}, z_t)] \right) \quad \text{[Term (I)]} \\
&\quad + \left\langle\!\!\left\langle \sup_{p_t \in \mathcal{P}} \mathbb{E}_{(z_t, \zeta_t) \sim p_t} \right\rangle\!\!\right\rangle_{t=1}^{T} \left( \sum_{t=1}^{T} \mathbb{E}_{\zeta_{1:t-1}, z_t}[\ell^\Phi(\hat{\pi}_t, \zeta_{1:t-1}, z_t)] - \inf_{\pi \in \Pi} \mathbb{E}_w \left[ \sum_{t=1}^{T} \ell(\pi, x_t[\pi^{(t-1)}, \zeta_{1:t-1}], z_t) \right] \right) \quad \text{[Term (II)]},
\end{aligned}
$$
$$(14)$$

where (i) follows from upper bounding the infimum over the policies $\pi_t$ by the choice of $\pi_t = \hat{\pi}_t$ and (ii) follows from the linearity of the expectation and sub-additivity of the supremum function.

Focussing on the first term in the above decomposition,

$$
\begin{aligned}
\text{Term (I)} &= \left\langle\!\!\left\langle \sup_{p_t \in \mathcal{P}} \mathbb{E}_{(z_t,\zeta_t)\sim p_t} \right\rangle\!\!\right\rangle_{t=1}^{T} \left( \sum_{t=1}^{T} \mathbb{E}_{w}[\ell(\hat{\pi}_t, x_t[\hat{\pi}_{1:t-1}, \zeta_{1:t-1}], z_t)] - \mathbb{E}_{z_t}[\ell^{\Phi}(\hat{\pi}_t, \zeta_{1:t-1}, z_t)] \right) \\
&= \left\langle\!\!\left\langle \sup_{p_t \in \mathcal{P}} \mathbb{E}_{\zeta_t} \right\rangle\!\!\right\rangle_{t=1}^{T} \left( \sum_{t=1}^{T} \mathbb{E}_{w,z_t}\left[\ell(\hat{\pi}_t, x_t[\hat{\pi}_{1:t-1}, \zeta_{1:t-1}], z_t)\right] - \mathbb{E}_{z_t}[\ell^{\Phi}(\hat{\pi}_t, \zeta_{1:t-1}, z_t)] \right) \\
&\overset{(i)}{=} \left\langle\!\!\left\langle \sup_{p_t \in \mathcal{P}} \mathbb{E}_{\zeta_t} \right\rangle\!\!\right\rangle_{t=1}^{T} \left( \sum_{t=1}^{T} \sup_{z_t} \left| \mathbb{E}_{w}[\ell(\hat{\pi}_t, x_t[\hat{\pi}_{1:t-1}, \zeta_{1:t-1}], z_t)] - [\ell^{\Phi}(\hat{\pi}_t, \zeta_{1:t-1}, z_t)] \right| \right) \\
&\overset{(ii)}{\leq} \sum_{t=1}^{T} \beta_{\mathsf{RERM},t},
\end{aligned}
\tag{15}
$$

where (i) follows from an application of Hölder's inequality and (ii) folllows from the definition of the dynamic stability of the regularized ERM algorithm.

Having established the upper bound on the first term, we now proceed to the second term of equation (14).

$$
\begin{aligned}
\text{Term (II)} &\overset{(i)}{=} \left\langle\!\!\left\langle \sup_{p_t \in \mathcal{P}} \mathbb{E}_{(z_t,\zeta_t)\sim p_t} \right\rangle\!\!\right\rangle_{t=1}^{T} \sup_{\pi \in \Pi} \left( \sum_{t=1}^{T} \mathbb{E}_{z_t}[\ell^{\Phi}(\hat{\pi}_t, \zeta_{1:t-1}, z_t)] - \sum_{t=1}^{T} \ell^{\Phi}(\pi, \zeta_{1:t-1}, z_t) \right) \\
&\overset{(ii)}{\leq} \left\langle\!\!\left\langle \sup_{p_t \in \mathcal{P}} \mathbb{E}_{(z_t,\zeta_t),z_t'} \right\rangle\!\!\right\rangle_{t=1}^{T} \sup_{\pi \in \Pi} \left( \sum_{t=1}^{T} \ell^{\Phi}(\pi, \zeta_{1:t-1}, z_t') - \sum_{t=1}^{T} \ell^{\Phi}(\pi, \zeta_{1:t-1} z_t) \right) + \lambda \sup_{\pi \in \Pi} \Omega(\pi) \\
&\overset{(iii)}{\leq} \left\langle\!\!\left\langle \sup_{p_t \in \mathcal{P}} \mathbb{E}_{(z_t,\zeta_t),z_t'} \right\rangle\!\!\right\rangle_{t=1}^{T} \mathbb{E}_{\epsilon_{1:T}} \sup_{\pi \in \Pi} \left( \sum_{t=1}^{T} \epsilon_t(\ell^{\Phi}(\pi, \zeta_{1:t-1}, z_t') - \ell^{\Phi}(\pi, \zeta_{1:t-1} z_t)) \right) + \lambda \sup_{\pi \in \Pi} \Omega(\pi) \\
&\leq 2 \left\langle\!\!\left\langle \sup_{p_t \in \mathcal{P}} \mathbb{E}_{(z_t,\zeta_t)} \right\rangle\!\!\right\rangle_{t=1}^{T} \mathbb{E}_{\epsilon_{1:T}} \sup_{\pi \in \Pi} \left( \sum_{t=1}^{T} \epsilon_t \ell^{\Phi}(\pi, \zeta_{1:t-1}, z_t) \right) + \lambda \sup_{\pi \in \Pi} \Omega(\pi),
\end{aligned}
$$

where (i) follows from rewriting the comparator in terms of the counterfactual loss $\ell^{\Phi}$, (ii) follows from Lemma 1, and in (iii) we introduce the Rademacher variables $\epsilon_t$. Using Jensen's inequality, we can obtain a further upper bound on Term (II) as

$$
\begin{aligned}
\text{Term (II)} &\leq 2 \left\langle\!\!\left\langle \sup_{p_t \in \mathcal{P}} \mathbb{E}_{(z_t,\zeta_t),\epsilon_t} \right\rangle\!\!\right\rangle_{t=1}^{T} \sup_{\pi \in \Pi} \left( \sum_{t=1}^{T} \epsilon_t \ell^{\Phi}(\pi, \zeta_{1:t-1}, z_t) \right) + \lambda \sup_{\pi \in \Pi} \Omega(\pi) \\
&\leq 2 \sup_{\mathbf{z},\boldsymbol{\zeta}} \mathbb{E}_{\epsilon_{1:T}} \sup_{\pi \in \Pi} \left( \sum_{t=1}^{T} \epsilon_t \ell^{\Phi}(\pi, [\boldsymbol{\zeta}_1(\epsilon), \dots, \boldsymbol{\zeta}_{t-1}(\epsilon)], \mathbf{z}_t(\epsilon)) \right) + \lambda \sup_{\pi \in \Pi} \Omega(\pi)
\end{aligned}
\tag{16}
$$

where in the last line, we have replaced the worst case joint distributions over the $\mathcal{Z}$ space by the corresponding worst case $\mathcal{Z}$-valued trees (see [13, 23] for more details). The upper bound on the value $\mathcal{V}_T(\Pi, \mathcal{Z}, \Phi, \ell)$ now follows from combining the bounds obtained in equations (15) and (16). $\qquad\square$

**Proof of Lemma 1.** For the purpose of this proof, we will use the short hand $\hat{\pi}_t := \pi_{\mathsf{RERM},t}$. We will prove the statement of the lemma via an inductive argument on the number of time steps $t$.

*Base Case:* For time step $t = 1$, we have that $\pi_{\mathsf{RERM},1}$ is the minimizer of the regularized loss implying

$$
\mathbb{E}_{z_1}\left[\ell^{\Phi}(\hat{\pi}_1, \zeta_\phi, z_1)\right] \leq \mathbb{E}_{z_1}\left[\ell^{\Phi}(\pi, \zeta_\phi, z_1)\right] + \lambda(\Omega(\pi) - \Omega(\hat{\pi}_1))
$$

for any $\pi \in \Pi$.

*Inductive Step:* Assume that the equation (13) holds for some time step $s$ and consider the cumulative loss at time step $s+1$

$$\sum_{t=1}^{s+1} \mathbb{E}_{z_t}\left[\ell^\Phi(\hat{\pi}_t, \zeta_{1:t-1}, z_t)\right] = \sum_{t=1}^{s} \mathbb{E}_{z_t}\left[\ell^\Phi(\hat{\pi}_t, \zeta_{1:t-1}, z_t)\right] + \mathbb{E}_{z_{s+1}}\left[\ell^\Phi(\hat{\pi}_{s+1}, \zeta_{1:s}, z_{s+1})\right]$$

$$\overset{(i)}{\leq} \sum_{t=1}^{s} \mathbb{E}_{z_t}\left[\ell^\Phi(\hat{\pi}_{s+1}, \zeta_{1:t-1}, z_t)\right] + \lambda(\Omega(\hat{\pi}_{s+1}) - \Omega(\hat{\pi}_1))$$

$$+ \mathbb{E}_{z_{s+1}}\left[\ell^\Phi(\hat{\pi}_{s+1}, \zeta_{1:s}, z_{s+1})\right]$$

$$\overset{(ii)}{\leq} \sum_{t=1}^{s+1} \mathbb{E}_{z_t}\left[\ell^\Phi(\pi, \zeta_{1:t-1}, z_t)\right] + \lambda(\Omega(\pi) - \Omega(\hat{\pi}_1)),$$

where (i) follows from the induction hypothesis for time $s$ and applying it for $\pi = \hat{\pi}_{s+1}$, and (ii) follows from the fact that $\hat{\pi}_{s+1}$ is the minimizer of the regularized objective at time $s+1$. This concludes the proof of the lemma. $\square$

## B.2 Proof of Proposition 2

For the purpose of this proof, we restrict our attention to an oblivious adversary wherein the adversary selects instances $\{z_t\}_{t=1}^T$ before the game begins. Several recent works [5, 9] have studied specific versions of a mini-batching algorithms under such an oblivious adversary.

For any mini-batching algorithm with parameter $\tau$, we consider denote by $\hat{T} = T/\tau$ as the effective time horizon[6] of the game. We now look at the mini-batched value of the game

$$\mathcal{V}_{T,\tau}(\Pi, \mathcal{Z}, \Phi, \ell) \leq \left\langle\!\left\langle \inf_{q_t} \left\langle\!\left\langle \sup_{z_{t,s}, \zeta_{t,s}} \right\rangle\!\right\rangle_{s=1}^{\tau} \mathbb{E}_{\pi_{t,s}\sim q_t} \right\rangle\!\right\rangle_{t=1}^{\hat{T}} \mathbb{E}_w[\mathrm{Reg}_{\mathrm{T}}^{\mathrm{pol}}]$$

represents the minimax policy regret for any such mini-batching algorithm $\mathcal{A}_\tau$ in the presence of an oblivious adversary. Let us denote the comparator term by

$$\psi(\zeta_{1:T}, z_{1:T}) := \inf_{\pi\in\Pi}\left[\mathbb{E}_w \sum_{t=1}^{T} \ell(\pi, x_t[\pi^{(t-1)}, \zeta_{1:t-1}], z_t)\right].$$

Following a repeated application of von Neumann's minimax theorem similar to the proof of Proposition 1, we upper bound the value

$$\mathcal{V}_{T,\tau}(\Pi, \mathcal{Z}, \Phi, \ell) \leq \left\langle\!\left\langle \sup_{\bar{p}_t} \inf_{\pi_t} \mathbb{E}_{(\bar{z}_t, \bar{\zeta}_t)\sim\bar{p}_t} \right\rangle\!\right\rangle_{t=1}^{\hat{T}} \left[\mathbb{E}_w \sum_{t=1}^{\hat{T}} \sum_{s=1}^{\tau} \ell\left(\pi_t, x_t[\pi_{1:t-1}^{(\tau)}, \pi_t^{(s-1)}, \bar{\zeta}_{1:t-1}, \bar{\zeta}_t^{1:s-1}], \bar{z}_t^s\right) - \psi(\bar{\zeta}_{1:\hat{T}}, \bar{z}_{1:\hat{T}})\right],$$

(17)

where the distribution $\bar{p} \in \mathcal{P}^\tau$ is a joint distribution over instances $(\bar{z}, \bar{\zeta}) \in \mathcal{Z}^\tau$ and we have explicitly indicated the dependence of the state variable on the past sequence of policies and adversarial instances. Define the mini-batched loss at time $t$

$$L_\tau(\pi_t, \bar{\zeta}_{1:t-1}, \bar{z}_t, \bar{\zeta}_t; \pi_{1:t-1}) := \mathbb{E}_w \sum_{s=1}^{\tau} \ell\left(\pi_t, x_t[\pi_{1:t-1}^{(\tau)}, \pi_t^{(s-1)}, \bar{\zeta}_{1:t-1}, \bar{\zeta}_t^{1:s-1}], \bar{z}_t^s\right),$$

and the corresponding mini-batched counterfactual loss

$$L_\tau^\Phi(\pi_t, \bar{\zeta}_{1:t-1}, \bar{z}_t, \bar{\zeta}_t) := \mathbb{E}_w \sum_{s=1}^{\tau} \ell\left(\pi_t, x_t[\pi_t^{(\tau)}, \pi_t^{(s-1)}, \bar{\zeta}_{1:t-1}, \bar{\zeta}_t^{1:s-1}], \bar{z}_t^s\right).$$

Given these definitions, we can rewrite equation (17) as

$$\mathcal{V}_{T,\tau}(\Pi, \mathcal{Z}, \Phi, \ell) \leq \left\langle\!\left\langle \sup_{\bar{p}_t} \inf_{\pi_t} \mathbb{E}_{(\bar{z}_t, \bar{\zeta}_t)\sim\bar{p}_t} \right\rangle\!\right\rangle_{t=1}^{\hat{T}} \left[\sum_{t=1}^{\hat{T}} L_\tau(\pi_t, \bar{\zeta}_{1:t-1}, \bar{z}_t, \bar{\zeta}_t; \pi_{1:t-1}) - \psi(\bar{\zeta}_{1:\hat{T}}, \bar{z}_{1:\hat{T}})\right].$$

(18)

The expression on the right can be seen as a dual game between a learner and an adversary of $\hat{T}$ rounds. At each round, the adversary reveals a joint distribution $\bar{p}_t$ over the instances and the learner selects a policy $q_t$. The learner then receives the loss $L_\tau$ for that round. We further bound the value by selecting the mini-batched dual ERM strategies for the learner, given by

$$\hat{\pi}_t^\tau := \pi_{\mathsf{ERM},t}^\tau = \operatorname*{argmin}_\pi \left( \sum_{s=1}^t \mathbb{E}_{\bar{z}_s, \bar{\zeta}_s} \left[ L_\tau^\Phi(\pi, \bar{\zeta}_{1:s-1}, \bar{z}_s, \bar{\zeta}_s) \right] \right).$$

Substituting the above mini-batched policies in equation (18) and following a similar set of steps as in proof of Theorem 1, we get,

$$\mathcal{V}_{T,\tau}(\Pi, \mathcal{Z}, \Phi, \ell) \leq \left\langle\!\!\left\langle \sup_{\bar{p}_t} \mathbb{E}_{(\bar{z}_t, \bar{\zeta}_t) \sim \bar{p}_t} \right\rangle\!\!\right\rangle_{t=1}^{\hat{T}} \left[ \sum_{t=1}^{\hat{T}} L_\tau(\hat{\pi}_t^\tau, \bar{\zeta}_{1:t-1}, \bar{z}_t, \bar{\zeta}_t; \hat{\pi}_{1:t-1}^\tau) - \psi(\bar{\zeta}_{1:\hat{T}}, \bar{z}_{1:\hat{T}}) \right]$$

$$\leq \left\langle\!\!\left\langle \sup_{\bar{p}_t} \mathbb{E}_{(\bar{z}_t, \bar{\zeta}_t) \sim \bar{p}_t} \right\rangle\!\!\right\rangle_{t=1}^{\hat{T}} \left[ \sum_{t=1}^{\hat{T}} L_\tau(\pi_t, \bar{\zeta}_{1:t-1}, \bar{z}_t, \bar{\zeta}_t; \pi_{1:t-1}) - L_\tau^\Phi(\pi_t, \bar{\zeta}_{1:t-1}, \bar{z}_t, \bar{\zeta}_t) \right]$$

$$+ \left\langle\!\!\left\langle \sup_{\bar{p}_t} \mathbb{E}_{(\bar{z}_t, \bar{\zeta}_t) \sim \bar{p}_t} \right\rangle\!\!\right\rangle_{t=1}^{\hat{T}} \left[ \sum_{t=1}^{\hat{T}} \mathbb{E}_{\bar{z}_t, \bar{\zeta}_t} \left[ L_\tau^\Phi(\pi_t, \bar{\zeta}_{1:t-1}, \bar{z}_t, \bar{\zeta}_t) \right] - \psi(\bar{\zeta}_{1:\hat{T}}, \bar{z}_{1:\hat{T}}) \right]$$

$$\overset{(i)}{\leq} \sum_{t=1}^T \beta_{\mathsf{ERM},t}^\tau + \left\langle\!\!\left\langle \sup_{\bar{p}_t} \mathbb{E}_{(\bar{z}_t, \bar{\zeta}_t) \sim \bar{p}_t} \right\rangle\!\!\right\rangle_{t=1}^{\hat{T}} \left[ \sum_{t=1}^{\hat{T}} \mathbb{E}_{\bar{z}_t, \bar{\zeta}_t} \left[ L_\tau^\Phi(\pi_t, \bar{\zeta}_{1:t-1}, \bar{z}_t, \bar{\zeta}_t) \right] - \psi(\bar{\zeta}_{1:\hat{T}}, \bar{z}_{1:\hat{T}}) \right],$$

where step (i) follows from upper bounding the sequence of joint distributions by the worst-case sequence of the adversary instances $\bar{z}_t, \bar{\zeta}_t$. The second term in the above expression can be upper bounded by using an induction argument, similar to that used in Lemma 1. The resulting bound is given by

$$\mathcal{V}_{T,\tau}(\Pi, \mathcal{Z}, \Phi, \ell) \leq \sum_{t=1}^T \beta_{\mathsf{ERM},t}^\tau + \left\langle\!\!\left\langle \sup_{\bar{p}_t} \mathbb{E}_{(\bar{z}_t, \bar{\zeta}_t)} \right\rangle\!\!\right\rangle_{t=1}^{\hat{T}} \sup_{\pi \in \Pi} \left[ \sum_{t=1}^{\hat{T}} \mathbb{E}_{\bar{z}_t, \bar{\zeta}_t} \left[ L_\tau^\Phi(\pi, \bar{\zeta}_{1:t-1}, \bar{z}_t, \bar{\zeta}_t) \right] - L_\tau^\Phi(\pi, \bar{\zeta}_{1:t-1}, \bar{z}_t, \bar{\zeta}_t) \right].$$

Symmetrizing the above expression and introducing Rademacher variables, we get,

$$\mathcal{V}_{T,\tau}(\Pi, \mathcal{Z}, \Phi, \ell) \leq \sum_{t=1}^T \beta_{\mathsf{ERM},t}^\tau + 2 \left\langle\!\!\left\langle \sup_{\bar{p}_t} \mathbb{E}_{(\bar{z}_t, \bar{\zeta}_t)} \mathbb{E}_{\epsilon_t} \right\rangle\!\!\right\rangle_{t=1}^{\hat{T}} \sup_{\pi \in \Pi} \left[ \sum_{t=1}^{\hat{T}} \epsilon_t L_\tau^\Phi(\pi, \bar{\zeta}_{1:t-1}, \bar{z}_t, \bar{\zeta}_t) \right]$$

$$\overset{(i)}{\leq} \sum_{t=1}^T \beta_{\mathsf{ERM},t}^\tau + 2 \left\langle\!\!\left\langle \sup_{\bar{p}_t} \mathbb{E}_{(\bar{z}_t, \bar{\zeta}_t)} \mathbb{E}_{\epsilon_t} \right\rangle\!\!\right\rangle_{t=1}^{\hat{T}} \sup_{\pi \in \Pi} \sum_{s=1}^\tau \left[ \left| \sum_{t=1}^{\hat{T}} \epsilon_t \ell_{s,t}^\Phi(\pi, \bar{\zeta}_{1:t-1}, \bar{z}_t, \bar{\zeta}_t) \right| \right]$$

$$\leq \sum_{t=1}^T \beta_{\mathsf{ERM},t}^\tau + 2 \sum_{s=1}^\tau \left\langle\!\!\left\langle \sup_{\bar{p}_t} \mathbb{E}_{(\bar{z}_t, \bar{\zeta}_t)} \mathbb{E}_{\epsilon_t} \right\rangle\!\!\right\rangle_{t=1}^{\hat{T}} \sup_{\pi \in \Pi} \left[ \left| \sum_{t=1}^{\hat{T}} \epsilon_t \ell_{s,t}^\Phi(\pi, \bar{\zeta}_{1:t-1}, \bar{z}_t, \bar{\zeta}_t) \right| \right]$$

$$\leq \sum_{t=1}^T \beta_{\mathsf{ERM},t}^\tau + 2\tau \cdot \sup_{s \in [\tau]} \mathfrak{R}_{T/\tau}^{\mathsf{seq}}(\ell_s^\Phi \circ \Pi),$$

where step (i) follows from swapping the supremum with the summation and in the last step we have used the definition of sequential Rademacher complexity with an absolute value. This establishes the desired claim. $\qquad\square$

## C   Proofs of lower bounds

### C.1   Proof of Theorem 2

We begin by recalling the example instance described in the proof sketch of Theorem 2. The online learning game between learner and adversary is given comprises of the state space $\mathcal{X} = \{x \in \mathbb{R}^d \mid \|x\|_2 \leq 1\}$ and the set of adversarial actions $\mathcal{Z}_\ell^{\mathsf{Lin}} = \{z \in \mathbb{R}^d \mid \|z\|_2 \leq 1\}$ for some dimension $d \geq 3$. In our setup, the adversarial instance space for the dynamics is empty. Given this state space, our policy class $\Pi_{\mathsf{Lin}}$ is a constant class of policies

$$\Pi_{\mathsf{Lin}} = \{\pi_f \mid \pi(x) = f \text{ for all states } x \text{ with } f \in \mathbb{B}_d(1)\},$$

consisting of policies $\pi_f$ which select the same action $f$ at each state $x$. With a slight abuse of notation, we represent the policy $\pi_t$ played by the learner at time by the corresponding $d$-dimensional vector $f_t$. Further, we let the dynamics function $\Phi_{\mathsf{Lin}}(x_t, f_t, \zeta_t) = f_t$ with the noise distribution $\mathcal{D}_w = 0$. Observe that the dynamics simply remembers the last action played by the learner and sets the next state as $x_{t+1} = f_t$ in a deterministic way with the starting state $x_1 = 0$. We now define the loss function which consists of two parts, a linear loss and a $L$-Lipschitz loss involving the dynamics:

$$
\ell_L(f_t, x_t, z_t) = \langle f_t, z_t \rangle + \sigma(f_t, x_t) \quad \text{where} \quad \sigma(f_t, x_t) = \begin{cases} L\|f_t - x_t\|_2 & \text{for } \|f_t - x_t\|_2 \leq \frac{1}{L} \\ 1 & \text{otherwise} \end{cases}. \quad (19)
$$

Observe that this example constructs a family of instances one for each value of the Lipschitz constant of $L$ of the function $\sigma$. For this setup, the loss function $\ell^\Phi$ for any time $t > 1$ is just the linear part of the loss

$$
\ell^\Phi(f, x[f^{t-1}], z) = \langle f, z \rangle + \underbrace{\sigma(f, x)}_{=0} = \langle f, z \rangle.
$$

Let us now break down the lower bound analysis into two cases: that of the Lipschitz constant $L \leq 1$ and $L > 1$.

**Case 1:** $L \leq 1$. For the case when $L \leq 1$, we lower bound the value of the game by ignoring the dynamics loss $\sigma$.

$$
\mathcal{V}_T(\Pi_{\mathsf{Lin}}, \mathcal{Z}^{\mathsf{Lin}}, \Phi_{\mathsf{Lin}}, \ell) = \left\langle\!\!\!\left\langle \inf_{q_t} \sup_{z_t} \mathbb{E}_{f_t} \right\rangle\!\!\!\right\rangle_{t=1}^T \sum_{t=1}^T \left( \langle f_t, z_t \rangle + \sigma(f_t, f_{t-1}) - \inf_{f \in \Pi_{\mathsf{Lin}}} \sum_{t=1}^T \langle f, z_t \rangle + \sigma(f, x_1) \right)
$$

$$
\overset{(i)}{\geq} \left\langle\!\!\!\left\langle \inf_{q_t} \sup_{z_t} \mathbb{E}_{f_t} \right\rangle\!\!\!\right\rangle_{t=1}^T \left( \sum_{t=1}^T \langle f_t, z_t \rangle - \inf_{f \in \Pi_{\mathsf{Lin}}} \left( \sum_{t=1}^T \langle f, z_t \rangle \right) \right) + 1,
$$

where (i) follows by noting that $\sigma(f_t, f_{t-1}) \in [0, 1]$. The above lower bound reduces the value to that of a online linear game between a learner and an adversary. A lower bound on the value of this game can be shown to be $\sqrt{T}/2$ (see [21]) and thus for the case when $L < 1$, we have that the value $\mathcal{V}_T(\Pi_{\mathsf{Lin}}, \mathcal{Z}^{\mathsf{Lin}}, \Phi_{\mathsf{Lin}}, \ell) \geq c\sqrt{T}$ for some $c = 0.5$.

**Case 2:** $L > 1$. We now proceed to the case when the Lipschitz constant[7] $L > 1$. In order to prove the requisite lower bound, we will describe the adversaries choice of action $z_t$. The adversaries strategy is to stick to some action $z$ and only *switch* to a new action when one of events E1 or E2 happen.

- E1 The time $t = \lambda L$ for $\lambda = \{1, \ldots, T/L\}$.
- E2 Let $t_0$ denote the last time the adversary had switched and denote the expected deviation from the previous move by $\delta_t := \mathbb{E}_{f_t, f_{t-1}} \|f_t - f_{t-1}\|$. Further, let $\Delta_{t_0, t} = \sum_{s=t_0}^t \delta_t$ denote the cumulative deviation of the moves from time $t_0$ upto time $t$. The adversary switches at time $t$ whenever $\Delta_{t_0, t} > \frac{1}{L}$.

Given the above events, we now define the adversarial action when it switches. Let $t$ be a time instance when one of E1 or E2 happens. Then the adversary selects $z_t$ such that

$$
\|z_t\|_2 = 1, \quad \langle Z_{t-1}, z_t \rangle = 0. \quad \mathbb{E}_{f_t \sim q_t} \langle f_t, z_t \rangle = 0,
$$

where $Z_{t-1} = \sum_{s=1}^{t-1} z_s$ is the cumulative sum of the adversary's past actions. Note that our choice of dimensions $d \geq 3$ ensures that such a $z_t$ will always exist.

In order to undersatnd the performance of any algorithm, let us partition the time interval into $T/L$ blocks each of length $L$ and denote each such bock $I_i := [L(i-1) + 1, Li]$. Let $k_i$ denote the number of times the learner causes event E2 to occur in the interval $I_i$. Observe that the cumulative loss within an interval $I_i$

$$
\left\langle\!\!\!\left\langle \mathbb{E}_{\pi_t} \right\rangle\!\!\!\right\rangle_{t \in I_i} \sum_{t \in I_i} \langle f_t, z_t \rangle + \sigma(f_t, f_{t-1}) \begin{cases} = 0 & \text{if } k_i = 0 \\ \geq k_i - 1 & \text{if } k_i \geq 1 \end{cases} \quad (20)
$$

where the lower bound for the case $k_i \geq 1$ follows since at each round the learner can only obtain inner product $\mathbb{E}_{f_t}\langle f_t, z_t \rangle \geq -1/L$ at each round of the interval. As soon as $\mathbb{E}_{f_t}\langle f_t, z_t \rangle < -1/L$, the adversary switches and ensures that $\mathbb{E}_{f_t}\langle f_t, z_t \rangle = 0$ for that time. The lower bound of $k_i - 1$ follows since the total length of the interval is $L$ and each time event E2 occurs, the learner pays a cumulative cost of 1. Note that the case for $k_i = 0$ is equivalent to the case $k_i = 1$ and hence going forward, we assume each $k_i \geq 1$.

Let $K = \sum_{i=1}^{T/L} k_i$ denote the total number of times an algorithm causes event E2 to happen and let $K = \beta T / L$ for some $\beta \in [1, L]$. Then, for any sequence of learner distributions $[q_1, \ldots, q_T]$, we have that the policy regret is lower bounded as

$$
\mathrm{Reg}_{\mathrm{T}}^{\mathsf{pol}}(\mathcal{A}) = \left\langle\!\!\left\langle \underset{f_t \sim q_t}{\mathbb{E}} \right\rangle\!\!\right\rangle_{t=1}^{T} \left( \langle f_t, z_t \rangle + \sigma(f_t, f_{t-1}) \inf_{f \in \Pi_{\mathsf{Lin}}} \sum_{t=1}^{T} \langle f, z_t \rangle \right) \overset{\text{(i)}}{\geq} K - \frac{T}{L} + \|Z_T\|_2, \tag{21}
$$

where (i) follows from an application of the Cauchy-Schwarz inequality and from the bound in (20). In order to lower bound the term $\|Z_T\|_2$, we define the set of times $T_{\mathsf{sw}} = [\hat{t}_1, \ldots, \hat{t}_{K_{\mathsf{ad}}}]$ where $K_{\mathsf{ad}} \leq T/L + K$ is the total number of switches that the adversary makes. Further, let us denote by $\hat{z}_i$ the choice of the adversary at time $\hat{t}_i$ and by $\gamma_i := \hat{t}_{i+1} - \hat{t}_i$ as the length of the interval for which the adversary played $\hat{z}_i$. Then, the squared norm

$$
\|Z_T\|_2^2 = \|\sum_{i=1}^{K_{\mathsf{ad}}} \gamma_i \hat{z}_i\|_2^2 = \sum_{i=1}^{K_{\mathsf{ad}}} \gamma_i^2 \|\hat{z}_i\|_2^2 + 2 \sum_{j=2}^{K_{\mathsf{ad}}} \gamma_j \langle \sum_{i=1}^{j-1} \gamma_i \hat{z}_i, \hat{z}_j \rangle = \sum_{i=1}^{K_{\mathsf{ad}}} \gamma_i^2,
$$

where the last ineqality follows from the choice of adversary ensuring that $\langle Z_{i-1}, z_i \rangle = 0$ and noting that $\|z_t\| = 1$ for all time $t$. We can now obtain a lower bound on $\|Z_T\|_2$ by an application of the cauchy-Schwarz inequality as

$$
\|Z_T\|_2 = \sqrt{\sum_{i=1}^{K_{\mathsf{ad}}} \gamma_i^2} \geq \frac{\sum_{i=1}^{K_{\mathsf{ad}}} \gamma_i}{\sqrt{K_{\mathsf{ad}}}} = \frac{T}{\sqrt{K_{\mathsf{ad}}}}.
$$

Substituting the above value in equation (21) and taking an infimum over all algorithms, we have that the minimax value

$$
\mathcal{V}_T(\Pi_{\mathsf{Lin}}, \mathcal{Z}^{\mathsf{Lin}}, \Phi_{\mathsf{Lin}}, \ell) \geq \inf_{\beta \in [1, L]} \left( (\beta - 1)\frac{T}{L} + \frac{\sqrt{LT}}{\sqrt{\beta + 1}} \right),
$$

where the inequality above follows from setting $K = \beta T / L$ and the fact that $K_{\mathsf{ad}} \leq T/L + K$. Optimizing for the value of $\beta$, we get that the minimax value

$$
\mathcal{V}_T(\Pi_{\mathsf{Lin}}, \mathcal{Z}, \Phi_{\mathsf{Lin}}, \ell_L) \geq \begin{cases} \frac{\sqrt{T}}{2} & \text{for } 0 < L < 1 \\ \frac{\sqrt{LT}}{\sqrt{2}} & \text{for } 1 \leq L \leq (32T)^{\frac{1}{3}} \\ 2^{\frac{1}{3}} T^{\frac{2}{3}} & \text{for } L > (32T)^{\frac{1}{3}} \end{cases} . \tag{22}
$$

Thus, we have that the value is lower bounded by these three different terms each corresponding to different ranges of the Lipschitz constant $L$. In order to obtain the requisite lower bounds, we now evaluate each term on the right hand side of equations (7a)- (7c).

**Bound (7a).** This corresponds to the sequential Rademacher complexity of the class $\Pi$ which corresponds to the unit Euclidean ball with respect to the linear loss. Following the calculations in Rakhlin and Sridharan (see[21, Chapter 10]), we have that

$$
\mathfrak{R}_T^{\mathsf{seq}}(\ell^\Phi \circ \mathcal{F}) \leq \sqrt{T}. \tag{23}
$$

**Bound (7b).** In order to establish an upper bound on the dynamic stability parameters, we consider the regularization given by the squared loss as $\Omega(f) = \frac{\|f\|_2^2}{2}$ with some regularization parameter $\lambda \geq 0$. Given that the form of the counterfactual loss $\ell^\Phi$, the regularized ERM

$$
f_{\mathsf{RERM},t} = \mathrm{Proj}_{\mathbb{B}_d(1)} \left( \frac{1}{\lambda} \sum_{s=1}^{t} \underset{z_s \sim p_s}{\mathbb{E}} [z_s] \right)
$$

for the dual game and the adversarial distributions given by $\{p_t\}$. Consequently, the stability parameters

$$\beta_{\mathsf{RERM},t} = \sigma(f_{\mathsf{RERM},t}, f_{\mathsf{RERM},t-1}) \le L\|f_{\mathsf{RERM},t} - f_{\mathsf{RERM},t-1}\|_2 \le \frac{L}{\lambda}.$$

Finally, the bound of equation (7b) can now be evaluated as

$$\inf_{\lambda>0}\left(\sum_{t=1}^{T}\beta_{\mathsf{RERM},t} + \lambda \cdot \sup_{\pi \in \Pi}\Omega(\pi)\right) \le \inf_{\lambda \ge 0}\left(\frac{LT}{\lambda} + \frac{\lambda}{2}\right) = \sqrt{\frac{LT}{2}}. \tag{24}$$

**Bound (7c).** We now proceed to the bound given by the mini-batching ERMs with parameter $\tau > 0$. The stability parameters for the mini-batching ERM can be upper bounded as

$$\beta_{\mathsf{ERM},t}^{\tau} : \begin{cases} \le 2 & \text{for } t \equiv 0 \bmod \tau \\ = 0 & \text{otherwise} \end{cases},$$

where the first case follows trivially from the fact that two unit norm vectors can have a distance at most 2 and the second case is a consequence of the fact that $\ell = \ell^{\Phi}$ anytime an algorithm repeats the past two policies. Combining this with the sequential Rademacher bound of equation (23) we have

$$\inf_{\tau>0}\left(\sum_{t=1}^{T}\beta_{\mathsf{ERM},t}^{\tau} + 2\tau\mathfrak{R}_{T/\tau}^{\mathsf{seq}}(\ell^{\Phi} \circ \Pi)\right) \le \inf_{\tau>0}\frac{2T}{\tau} + 2\tau\sqrt{\frac{T}{\tau}} = 2T^{\frac{2}{3}}. \tag{25}$$

Comparing equations (23), (24) and (25) with the lower bounds on the value $\mathcal{V}_T$ in equation (22), we see that the sequential Rademacher bound is tight up to constant factors in the regime $L \le 1$, the dynamic stability bounds are tight for the regime $1 < L < (32T)^{\frac{1}{3}}$ and the mini-batching bounds are tight for the range $(32T)^{\frac{1}{3}} \le L \le T$. This establishes the desired claim. $\square$

### C.2 Proof of Proposition 3

We establish both parts of the proposition separately. For both the subparts, we lower bound the value $\mathcal{V}_T$ be first describing a problem instance $(\Pi, \mathcal{Z}, \Phi, \ell)$ and compute the value for a specific choice of adversarial actions. We assume that the loss function $|\ell(f, z)| \le 1$ for all $f \in \mathcal{F}$ and $z \in \mathcal{Z}_\ell$. The bounds for larger loss values can be obtained by a corresponding scaling.

#### C.2.1 Proof of part (a)

We denote by $K = T/\tau$ the number of times a mini-batching algorithm changes its policy.

**Constructing online learning with dynamics instance.** Given an instance of the online learning problem $(\mathcal{F}, \mathcal{Z}_\ell, \ell)$, we construct the online learning with dynamics instance with state space $\mathcal{X} = \mathcal{F}$ and policy class

$$\Pi_{\mathcal{F}} = \{\pi_f \mid f \in \mathcal{F}, \pi_f(x) = f \text{ for all } x \in \mathcal{X}\},$$

which plays the same action $f$ for all states $x \in \mathcal{X}$. Going forward, with a slight abuse of notation we use the action $f$ and the constant policy $\pi_f$ interchangeably.

The adversary's loss instance space is given by $\tilde{\mathcal{Z}}_\ell = \mathcal{Z}_\ell \times \{-1, +1\}$ with the actions $z_t \in \mathcal{Z}_\ell$ and $\epsilon_t \in \{-1, +1\}$. The dynamics function $\Phi(x, \pi_f, \zeta) = f$ represent the deterministic dynamics which remembers the last action played by the learner and is not affect by the adversary. The instantaneous loss $\tilde{\ell}(f_t, x_t, (z_t, \epsilon_t))$ is given as

$$\tilde{\ell}(f_t, x_t, (z_t, \epsilon_t)) = \epsilon_t \ell(f_t, z_t) + \mathbb{I}[f_t \ne x_t].$$

With the above loss function, notice that the counterfactual loss $\ell^{\Phi}(f_t, (z_t, \epsilon_t)) = \epsilon_t \ell(f_t, z_t)$ for all time $t > 1$ and the dynamic stability parameters for any algorithm $\beta_t = \mathbb{E}_{\mathcal{A}}[\mathbb{I}[f_t \ne f_{t-1}]]$.

**Specifying the adversary.** Given the online learning with dynamics problem above, we now specify an adversary for this setup. Let $K^* = T/\tau^*$ denote the optimal number of switches given by

$$K^* = \underset{K}{\mathrm{argmin}}\left(K + 2\frac{T}{K}\mathfrak{R}_K^{\mathsf{seq}}(\ell \circ \Pi_{\mathcal{F}})\right).$$

Note that such a value of $K^*$ is an equalizer of the two terms and ensures that $K^*$ and $\frac{2T}{K^*}\mathfrak{R}^{\text{seq}}_{K^*}$ are equal. Now, consider the worst case $\mathcal{Z}_\ell$-valued tree $\mathbf{z}_T$ of depth $T$ corresponding to the online learning problem $(\mathcal{F}, \mathcal{Z}_\ell, \ell)$

$$\mathbf{z}_T = \operatorname*{argsup}_{\mathbf{z}} \mathbb{E}_\epsilon\left[\sup_{f\in\mathcal{F}}\sum_{t=1}^{T}\epsilon_t\ell(f,\mathbf{z}(\epsilon))\right].$$

The adversary computes the tree $\mathbf{z}_{2K^*}$ produces instances $(z_t,\epsilon_t)$ as

Case 1. Whenever $t = \lambda\tau^*/2$ for $\lambda = \{1,\ldots,2T/K^*\}$, the adversary samples $\epsilon_t$ as a Rademacher random variable and sets $z_t = \mathbf{z}_{2K^*}(\epsilon_{1:2(t-1)/\tau^*})$.

Case 2. For any time $t \neq \lambda\tau^*$, the adversary computes the probability of switch $p_t^{\text{sw}} = \mathbb{E}_{\mathcal{A}}\mathbb{I}[f_t \neq f_{t-1}]$ and selects instance $(z_t,\epsilon_t)$ as

$$(z_t,\epsilon_t) = \begin{cases} (z_{t-1},\epsilon_t \sim \text{Rad}) & \text{if } p_t^{\text{sw}} > \frac{1}{2} \\ (z_{t-1},\epsilon_{t-1}) & \text{otherwise} \end{cases}.$$

**Lower bound on the value.** For any algorithm $\mathcal{A}$ producing distributions $q_1,\ldots,q_T$, the expected policy regret is

$$\mathbb{E}_{\mathcal{A},\epsilon}[\text{Reg}_{\text{T}}^{\text{pol}}] \overset{(i)}{\geq} \sum_{t=1}^{T}\mathbb{I}[p_t^{\text{sw}} > 0.5] - \mathbb{E}_\epsilon\inf_{f\in\mathcal{F}}\sum_{t=1}^{T}\ell^\Phi(f,z_t,\epsilon_t)$$

$$= \sum_{t=1}^{T}\mathbb{I}[p_t^{\text{sw}} > 0.5] + \mathbb{E}_\epsilon\sup_{f\in\mathcal{F}}\sum_{t=1}^{T}\epsilon_t\ell(f,z_t)$$

where inequality (i) follows from fact that whenever $p_t^{\text{sw}} > 0.5$, the adversary samples a new rademacher variable $\epsilon_t$. For any algorithm, let $K^{\text{sw}} = \sum_t\mathbb{I}[p_t^{\text{sw}} > 0.5]$ denote the number of time periods for which the switching probability is greater than half. We break the lower bound in two separate cases depending on the value of $K^{\text{sw}}$.

**Case 1: $K^{\text{sw}} \geq K^*$.** For this case, the policy regret for any algorithm can be lower bounded as

$$\mathbb{E}_{\mathcal{A},\epsilon}[\text{Reg}_{\text{T}}^{\text{pol}}] \geq K^* \overset{(i)}{=} \frac{1}{2}\left(K^* + 2\frac{T}{K^*}\mathfrak{R}^{\text{seq}}_{K^*}(\ell \circ \Pi_\mathcal{F})\right), \tag{26}$$

where (i) follows from our previous observation that $K^* = \frac{2T}{K^*}\mathfrak{R}^{\text{seq}}_{K^*}$.

**Case 2: $K^{\text{sw}} < K^*$.** For this case, not that the complete time horizon can be divided into atmost $3K^*$ intervals wherein the adversary selects the same instances $(z,\epsilon)$, each of length at most $T/2K^*$. By the pigeonhole principle, we must have at least $K^*$ intervals having length $T/2K^*$ beginning at time $t = \lambda\tau^*/2$ for some integral $\lambda$. Denote the collection of times in these intervals by $\mathcal{I}$. We can now lower bound the policy regret as

$$\mathbb{E}_{\mathcal{A},\epsilon}[\text{Reg}_{\text{T}}^{\text{pol}}] \geq \mathbb{E}_\epsilon\sup_{f\in\mathcal{F}}\sum_{t=1}^{T}\epsilon_t\ell(\pi,\mathbf{z}_{2K^*}(\epsilon))$$

$$\overset{(i)}{\geq} \mathbb{E}_{\epsilon_t:t\in\mathcal{I}}\left[\sup_{f\in\mathcal{F}}\sum_{t\in\mathcal{I}}\epsilon_t\ell(\pi,\mathbf{z}_{2K^*}(\epsilon))\right]$$

$$\overset{(ii)}{\geq} \frac{T}{2K^*}\cdot\mathfrak{R}^{\text{seq}}_{K^*}(\ell\circ\mathcal{F})$$

$$= \frac{1}{4}\left(K^* + 2\frac{T}{K^*}\mathfrak{R}^{\text{seq}}_{K^*}(\ell\circ\Pi_\mathcal{F})\right) \tag{27}$$

whre (i) follows from the an application of Jensen's inequality and the fact that the respampled $\epsilon_t$ when adversary switched because of the learner are not used to parse the tree $\mathbf{z}_{2K^*}$ and (ii) follows from noting that each pair $(z,\epsilon)$ was used exaclty $T/2K^*$ times.

Combining equations (26) and (27) along with the observation that the minimax value of the online learning with dynamics $\mathcal{V}_T(\Pi, \mathcal{Z}_\ell \times \{+1,-1\}, \Phi, \tilde{\ell})$ is the minimum policy regret for any algorithm establishes the desired claim. $\qquad\square$

### C.2.2 Proof of part (b)

We will proof a slighlty stronger version of the lower bound from which the desired statement will follow. We follow a strategy similar to the one used in the proof of part (a) above.

**Constructing online learning with dynamics instance.** Let the dynamics function be defined over states space $\mathcal{X}$ and adversary instance space $\mathcal{Z}_\Phi$. Consider any loss function $\tilde{\ell} : \Pi \times \mathcal{X} \times \tilde{\mathcal{Z}}_\ell \mapsto \mathbb{R}$ for some instance space $\tilde{\mathcal{Z}}_\ell$. We define the space of adversarial loss actions $\mathcal{Z}_\ell = \tilde{\mathcal{Z}}_\ell \times \{-1, +1\}$ and the corresponding loss $\ell(\pi, x, (z, \epsilon)) = \epsilon \cdot \tilde{\ell}(\pi, x, z)$. This defines an instance of the online learning with dynamics problem $(\Pi, \mathcal{Z} = \mathcal{Z}_\ell \times \mathcal{Z}_\Phi, \Phi, \ell)$.

**Specifying the adversary.** Consider the $\tilde{\mathcal{Z}}_l$ and $\mathcal{Z}_\Phi$ valued trees $\mathbf{z}_T$ and $\boldsymbol{\zeta}_T$ defined as

$$
(\mathbf{z}_T, \boldsymbol{\zeta}_T) = \operatorname*{argsup}_{\mathbf{z}, \boldsymbol{\zeta}} \mathbb{E}_\epsilon \left[ \sup_{\pi \in \Pi} \sum_{t=1}^{T} \epsilon_t \ell^\Phi(\pi, \boldsymbol{\zeta}_{1:t-1}(\epsilon), \mathbf{z}(\epsilon)) \right],
$$

which correspond to the worst-case trees of the sequential Rademacher complexity of the class $\ell^\Phi \circ \Pi$. At every time $t$, the adversary selects $(z_t, \epsilon_t, \zeta_t)$ by sampling a uniform Rademacher variable and traversing the two trees as

$$
\epsilon_t \sim \text{Rad}, \quad z_t = \mathbf{z}_T(\epsilon_{1:t-1}) \quad \text{and} \quad \zeta_t = \boldsymbol{\zeta}_T(\epsilon_{1:t-1}).
$$

**Lower bound on the value.** For any algorithm $\mathcal{A}$, the expected policy regret is given by

$$
\mathbb{E}_{\mathcal{A}, \epsilon}[\text{Reg}_T^{\text{pol}}] \overset{(i)}{=} \mathbb{E}_\epsilon \left[ \sup_{\pi \in \Pi} \sum_{t=1}^{T} \epsilon_t \ell^\Phi(\pi, \zeta_{1:t-1}, z_t) \right] \overset{(ii)}{=} \mathfrak{R}_T^{\text{seq}}(\ell^\Phi \circ \Pi),
$$

where (i) follows from noting that the loss at time $t$ is a zero-mean random variable and (ii) is implied by the definition of the trees $\mathbf{z}_T$ and $\boldsymbol{\zeta}_T$.

Finally, observing that the minimax value is equal to the policy regret of the best algorithm completes the proof. $\qquad\square$

## D   Details of examples

In this section, we work out the examples mentioned in Section 5 in detail and prove the rates for their respective value functions.

Before proceeding to the examples, we introduce some notation. Most of the examples that we consider have dynamics which are not affected by the adversary, that is, the instance space $\mathcal{Z}_\Phi$ is empty. We focus on this special case and derive a few results which will be helpful in deriving bounds for the examples.

Borrowing from the theory of stochastic processes, we next define ergodicity of the dynamics which relates a sequence of instantaneous losses to a notion of stainary loss $\ell_*^\Phi : \Pi \times \mathcal{Z} \mapsto \mathbb{R}$.

**Definition 4** (Ergodicity). *We say that the dynamics $\Phi$ are ergodic with respect to the loss $\ell$ if for any policy $\pi \in \Pi$ and adversarial action $z \in \mathcal{Z}_\ell$, the expected loss converges to a stationary loss starting from any state $x_1$ as*

$$
\lim_{t \to \infty} \mathbb{E}_{\{w_t\}} \ell(\pi, x_t[\pi^{(t-1)}], z) = \ell_*^\Phi(\pi, z).
$$

The loss function $\ell_*^\Phi$ can be seen as the limit of the counterfactual losses $\ell^\Phi$ and as we shortly show, the losses and dynamics in most of our examples satisfy this ergodicity assumption. For setups where such a stationary loss exists, we define the ergodic stability parameters $\beta_t^*$ analogous to the dynamic stability parameters.

**Definition 5** (Ergodic Stability). *An algorithm $\mathcal{A}$ is said to be $\{\beta_t^*\}$-ergodic stable if for all sequences of adversarial actions $[z_1, \ldots, z_T]$ and time instances $t \in [T]$*

$$
\left| \mathbb{E}_{w_{1:t-1}}[\ell(\pi_t, x_t[\pi_{1:t-1}, w_{1:t-1}], z_t)] - \ell_*^\Phi(\pi_t, z_t) \right| \leq \beta_t^* \quad \text{where} \quad \pi_t = \mathcal{A}(z_{1:t-1}).
$$

Observe that the ergodic stability parameters are defined with respect to the stationary loss as compared to their dynamic stability counterparts which were defined with respect to the counterfactual losses. Next, we define the set of regularized ERMs $\pi^*_{\mathsf{RERM}}$ with respect to these stationary loss as

$$\pi^*{}_{\mathsf{RERM},t} = \underset{\pi \in \Pi}{\operatorname{argmin}} \sum_{t=1}^{T} \underset{z_t \sim p_t}{\mathbb{E}} \left[ \ell_*^{\Phi}(\pi, z_t) \right] + \lambda \cdot \Omega(\pi) \;, \tag{28}$$

for some regularization function $\Omega$ and parameter $\lambda \geq 0$. Given this notation, the following corollary upper bounds the value of the game $\mathcal{V}_T(\Pi, \mathcal{Z}, \Phi, \ell)$ in terms of the sequential Rademacher complexity of the loss class $\ell_*^{\Phi} \circ \Pi$ and the ergodic stability of the RERMs $\pi^*_{\mathsf{RERM}}$.

**Corollary 5.** *For any online learning with dynamics instance $(\Pi, \mathcal{Z}, \Phi, \ell)$ with ergodic dynamics $\Phi$, consider the set of regularized ERMs given by eq. (28) with regularization function $\Omega$ and parameter $\lambda \geq 0$ having ergodic stability parameters $\{\beta^*_{\mathsf{RERM},t}\}_{t=1}^{T}$. Then, we have that the value of the game*

$$\mathcal{V}_T(\Pi, \mathcal{Z}, \Phi, \ell) \leq \sum_{t=1}^{T} \beta^*_{\mathsf{RERM},t} + 2\mathfrak{R}_T^{\mathsf{seq}}(\ell_*^{\Phi} \circ \Pi) + 2\lambda \sup_{\pi \in \Pi} \Omega(\pi) + \underbrace{\sup_{\pi \in \Pi} \sum_{t=1}^{T} \left| \ell^{\Phi}(\pi, z_t, t) - \ell_*^{\Phi}(\pi, z) \right|}_{\textit{Mixing Gap}}. \tag{29}$$

Compared with the corrresponding upper bound in Theorem 1, the above bound has an additional term: the worst case deviation of the counterfactual losses[8] from the stationary losses. This term, which we call the *Mixing Gap*, captures how quickly the dynamics mix to these stationary stationary losses when the same policy is repeatedly played over a period of time. The proof of the corollary is very similar to that of Theorem 1 and we provide it below for completeness.

*Proof of Corollary 5.* We begin by considering the value of the game and its equivalence to the dual game established by Proposition 1 as

$$\mathcal{V}_T(\Pi, \mathcal{Z}, \Phi) = \left\langle\!\left\langle \sup_{p_t \in \mathcal{P}} \inf_{\pi_t} \underset{z_t \sim p_t}{\mathbb{E}} \right\rangle\!\right\rangle_{t=1}^{T} \left[ \underset{w}{\mathbb{E}}\left[ \sum_{t=1}^{T} \ell(\pi_t, x_t[\pi_{1:t-1}], z_t) \right] - \inf_{\pi \in \Pi} \underset{w}{\mathbb{E}}\left[ \sum_{t=1}^{T} \ell(\pi, x_t[\pi^{(t-1)}], z_t) \right] \right]$$

$$\overset{(i)}{\leq} \left\langle\!\left\langle \sup_{p_t \in \mathcal{P}} \underset{z_t \sim p_t}{\mathbb{E}} \right\rangle\!\right\rangle_{t=1}^{T} \left[ \underset{w}{\mathbb{E}}\left[ \sum_{t=1}^{T} \ell(\pi_{\mathsf{RERM},t}, x_t[\pi_{\mathsf{RERM},1:t-1}], z_t) \right] - \inf_{\pi \in \Pi} \underset{w}{\mathbb{E}}\left[ \sum_{t=1}^{T} \ell(\pi, x_t[\pi^{(t-1)}], z_t) \right] \right]$$

$$\overset{(ii)}{\leq} \left\langle\!\left\langle \sup_{p_t \in \mathcal{P}} \underset{z_t \sim p_t}{\mathbb{E}} \right\rangle\!\right\rangle_{t=1}^{T} \left[ \underset{w}{\mathbb{E}}\left[ \sum_{t=1}^{T} \underset{z_t \sim p_t}{\mathbb{E}} \left[ \ell(\pi^*_{\mathsf{RERM},t}, x_t[\pi^*_{\mathsf{RERM},1:t-1}], z_t) \right] \right] - \inf_{\pi \in \Pi} \left( \sum_{t=1}^{T} \ell_*^{\Phi}(\pi, z_t) \right) \right] \quad \text{[Term (I)]}$$

$$+ \left\langle\!\left\langle \sup_{p_t \in \mathcal{P}} \underset{z_t \sim p_t}{\mathbb{E}} \right\rangle\!\right\rangle_{t=1}^{T} \left[ \sup_{\pi \in \Pi} \left( \sum_{t=1}^{T} \ell_*^{\Phi}(\pi, z_t) - \underset{w}{\mathbb{E}}\left[ \sum_{t=1}^{T} \ell(\pi, x_t[\pi^{(t-1)}], z_t) \right] \right) \right],$$

where (i) follows from replacing the $\inf_{\pi_t}$ at every time step with $\pi^*_{\mathsf{RERM},t}$ and (ii) follows from the subadditivity of the sup function and the fact that $\inf_y(g(y) + h(y)) \geq \inf_y g(y) + \inf_y h(y)$. The second term in the expression now corresponds to the worst-case deviation of the stationary loss from the counterfactual losses.

Further, observe that Term (I) above is similar to the term obtained in equation 14 and the desired upper bound can be obtained by following the same sequence of steps as in the proof of Theorem 1. $\qquad\square$

Having established the above corollary, we proceed to studying the examples from Section 5 in detail.

### D.1 Online Isotron with dynamics

In this section, we look at the online Isotron with dynamics problem introduced in Section 5. The setup consists of a real valued state space $\mathcal{X} = \mathbb{R}$. The policy class $\Pi$ is based on a function class $\mathcal{F}$ consisting of a 1-Lipschitz function along with a $d + 1$ unit dimensional vector and is given as

$$\mathcal{F} = \{ f = (\sigma, \mathbf{w} = (w_1, w)) \mid \sigma : [-1, 1] \mapsto [-1, 1] \text{ 1-Lipschitz}, \; \mathbf{w} \in \mathbb{R}^{d+1} \; |w_1| \leq 1 \; \|w\|_2 \leq 1 \},$$
$$\Pi_{\mathcal{F}} = \{ \pi_f \mid \pi \in \mathcal{F}, \; \pi_f(x) = f \text{ for all } x \in \mathcal{X} \}.$$

The adversary selects instances in the space $\mathcal{Z} = [-1, 1]^{d+1} \times [-1, 1]$ and we represent each instance $z = (z_1, \mathbf{x}, y)$. Given this setup, we now formalize the online learning protocol, starting from initial state $x_1 = 0$.

On round $t = 1, \ldots, T$,

- the learner selects a policy $\pi_t \in \Pi_{\mathcal{F}}$ and the adversary selects $z_t \in \mathcal{Z}$
- the learner receives $\text{loss}\ell(\pi_t, x_t, z_t) = (y_t - \sigma(\langle \mathbf{x}_t, w_t \rangle))^2 + (z_{t,1} - w_{t,1})^2 + (x_t - w_{t,1})^2$
- the state of the system transitions to $x_{t+1} = w_{t,1}$

Given this setup, the next corollary provides a bound on the value of this game $\mathcal{V}_{\text{Iso},T}(\Pi_{\mathcal{F}}, \mathcal{Z}, \Phi, \ell)$.

**Corollary 6** (Online Isotron with dynamics). *For the online Isotron with dynamics problem, there exists a universal constant $c > 0$ such that*

$$\mathcal{V}_{\text{Iso},T}(\Pi_{\mathcal{F}}, \mathcal{Z}, \Phi, \ell) \leq c\sqrt{T}\log^{3/2}(T).$$

It is worth recalling that the above game is an dynamical extension of the online Isotron problem instance studied by [23]. We are not aware of any primal algorithm which can get a rate of $\sqrt{T}$ for both the online learning version as well the dynamical version of this game. Our non-constructive analysis on the other hand proved a way to gaurantee learnabaility at this rate for the Isotron problem.

*Proof of Corollary 6.* We prove the above statement by bounding the mixing gap and the ergodic stability parameters for the appropriate regularized ERMs.

**Bound on mixing gap.** Note that for any time $t > 1$, the losses $\ell_*^\Phi$ and $\ell^\Phi$ are identical since the state variable only depends on the policy at time $t - 1$. Therefore, one can upper bound the loss by constant $c = 12$.

**ERMs.** For the dual game, we consider the ERM at time $t$ given by

$$f_{\text{ERM},t} = (\sigma_t, \mathbf{w}_t) = \underset{\sigma, \mathbf{w}}{\operatorname{argmin}} \left\{ \sum_{s=1}^{t} \left( \underset{z_s \sim p_s}{\mathbb{E}} \left[ (y_s - \sigma(\langle \mathbf{x}_s, w \rangle))^2 + (z_{s,1} - w_1)^2 \right] \right) \right\} ,$$

and set $\pi_t = \pi_{f_{\text{ERM},t}}$.

**Ergodic stability parameters.** Note that objective function in the above equation is strongly-convex with respect to the paramter $w_1$ and a simple calculation shows that $|w_{t,1} - w_{t-1,1}| \leq \frac{2}{t}$. We can now bound the ergodic stability parameter as

$$\beta_{\text{RERM},t}^* = |\ell(\pi_t, x_t[\pi_{1:t-1}], z_t) - \ell_*^\Phi(\pi_t, z_t)| = |w_{t-1,1} - w_{t,1}|^2 \leq \frac{4}{t^2}. \tag{30}$$

**Bound on the value.** Having established bounds on the mixing gap and the ergodic stability parameters of the ERM, we now use Corollary 5 to upper bound the value of the game as

$$\mathcal{V}_{\text{Iso},T}(\Pi_{\mathcal{F}}, \mathcal{Z}, \Phi, \ell) \overset{\text{(i)}}{\leq} \sum_{t=1}^{T} \beta_{\text{RERM},t}^* + 2\mathfrak{R}_T^{\text{seq}}(\ell_*^\Phi \circ \Pi_{\mathcal{F}}) + 16$$

$$\overset{\text{Eq. (30)}}{\leq} 8 + 2\mathfrak{R}_T^{\text{seq}}(\ell_*^\Phi \circ \Pi_{\mathcal{F}}) + 16$$

$$\overset{\text{(ii)}}{\leq} c\sqrt{T}\log^{3/2}(T) ,$$

where (i) follows by the upper bound of 16 on the mixing gap and (ii) follows by the corresponding bound on the sequential Rademacher complexity $2\mathfrak{R}_T^{\text{seq}}(\ell_*^\Phi \circ \Pi_{\mathcal{F}})$ from [23, Proposition 18]. $\square$

## D.2 Online Markov decision processes

In this section, we revisit the problem of Online Markov Decision Processes (MDPs) studied in [11]. The setup consists of a finite state space such that $|\mathcal{X}| = S$ and a finite action space with $|\mathcal{U}| = A$. The policy class $\Pi$ consists of all stationary policies, that is,

$$\Pi_{\text{MDP}} = \{\pi \mid \pi : \mathcal{X} \mapsto \Delta(\mathcal{U})\},$$

where $\Delta(\mathcal{U})$ represents the set of all probability disitributions over the action space. Un addition, the transitions are drawn according to a known function $P : \mathcal{X} \times \mathcal{U} \mapsto \Delta(\mathcal{U})$. The sequential game then proceeds as follows, starting from some state $x_1 \sim d$:

On round $t = 1, \ldots, T$,

- the learner selects a policy $\pi_t \in \Pi_{\mathsf{MDP}}$ and the adversary selects $z_t \in \mathcal{Z} = [0,1]^{S \times A}$
- the learner receives loss $\ell(\pi_t, x_t, z_t) = z_t(x_t, \pi_t(x_t))$
- the state of the system transitions to $x_{t+1} \sim P(x_t, u_t)$

For every stationary policy $\pi$, we let $P^f$ denote the transition function induced by $\pi$, that is,

$$P^f(x, x') := \sum_{u \in \mathcal{U}} \pi^u(x) P^{x'}(x, u),$$

where we have used superscript to denote the relevant coordinate of the vector. As in [11], we make the following mixability assumptions about the underlying MDP.

**Assumption 1** (MDP Unichain). *We assume that the underlying MDP given by the transition function $P$ is unichain. Further, there exists $\tau \geq 1$ such that for all policies $\pi$ and distributions $d, d' \in \Delta(\mathcal{U})$ we have*

$$\|dP^\pi - d'P^f\|_1 \leq e^{-1/\tau} \|d - d'\|_1.$$

The parameter $\tau$ is often referred to as the mixing time of the MDP. Since the MDP is assumed to be unichain, every policy $\pi$ has a well defined unique stationary distribution $d_\pi$ with the stationary loss given by $\ell_*^\Phi(\pi, z) = \mathbb{E}_{x \sim d_\pi} \mathbb{E}_{u \sim \pi(x)} z(x, u)$. Given this setup, we can obtain an upper bound on the value $\mathcal{V}_{\mathsf{MDP}, T}$ as follows:

**Corollary 7** (Online MDP). *For the online Markov Decision Process sequential game satisfying Assumption 1, the value $\mathcal{V}_{\mathsf{MDP}, T}(\Pi_{\mathsf{MDP}}, \mathcal{Z}, \Phi)$ is bounded by*

$$\mathcal{V}_{\mathsf{MDP}, T}(\Pi_{\mathsf{MDP}}, \mathcal{Z}, \Phi, \ell) \leq 4\tau \sqrt{TS \log A} + 2\tau(1 + e^{1/\tau}).$$

The above corollary helps one recover the same $\mathcal{O}(\sqrt{T})$ regret bound that was obtained by [11]. In terms of the dependence of problem specific parameters, while our bound above shows a $\sqrt{S}$ dependence, their bound was independent of $S$. However note that while the setting studied by [11] consisted of the weaker oblivious adversary, we consider the stronger adaptive adversary which can adapt to the learners strategy.

*Proof of Corollary 7.* In order to establish the bound, we begin by bounding the ergodic stability parameters as well as the mixing gap for loss $\ell^\Phi$ and $\ell_*^\Phi$.

**Bound on mixing gap.** Consider any policy $\pi \in \Pi_{\mathsf{MDP}}$ and the associated steady state distribution $d_\pi$. The stationary loss for this problem is then

$$\ell_*^\Phi(\pi, z) = \mathbb{E}_{x \sim d_\pi} \mathbb{E}_{u \sim \pi(x)} [z(x, u)].$$

Consider now the difference between the stationary loss and the counterfactual loss at any time $t$

$$
\begin{aligned}
\left| \ell^\Phi(\pi, z, t) - \ell_*^\Phi(\pi, z) \right| &= \left| \mathbb{E}_{x_t^\pi \sim d_\pi^t} \mathbb{E}_{u \sim \pi(x_t^\pi)} [z(x_t^\pi, u)] - \mathbb{E}_{x \sim d_\pi} \mathbb{E}_{u \sim \pi(x)} [z(x, u)] \right| \\
&\overset{(i)}{=} \left| \mathbb{E}_{x \sim d_\pi^t} [\tilde{z}_\pi(x)] - \mathbb{E}_{x \sim d_\pi} [\tilde{z}_\pi(x)] \right| \\
&\overset{(ii)}{\leq} \|\tilde{z}_\pi\|_\infty \cdot \|d_\pi^t - d_\pi\|_1 \\
&\overset{(iii)}{\leq} 2e^{-(t-1)/\tau},
\end{aligned}
\tag{31}
$$

where in (i), we use the redefined loss function $\tilde{z}_\pi(x) := \mathbb{E}_{u \sim \pi(x)} z(x, u)$, (ii) follows from an application of Hölder's inequality, and (iii) follows from Assumption 1 and the fact the $\|d_1 - d_\pi\| \leq 2$.

**Ergodic stability parameters.** For this setup, we will be using a regularized ERM and parameterize the policy $\pi_{\mathsf{RERM},t}$ as a distribution over the deterministic policies present in $\Pi_{\mathsf{MDP}}$. Let us denote this subset of policies by $\Pi_{\mathsf{MDP}}^{\mathsf{det}}$. Note that a distribution $q$ in $\mathcal{Q}_{\mathsf{MDP}}^{\mathsf{det}}$ is randomized policy in the class $\Pi_{\mathsf{MDP}}$. We will work with the negative entropy function as the regularizer.

$$q_{\mathsf{RERM},t} \in \underset{q \in \mathcal{Q}_{\mathsf{MDP}}}{\operatorname{argmin}} \left( \underset{\pi \sim q}{\mathbb{E}} \left[ \sum_{s=1}^{t} \underset{x \sim d_\pi}{\mathbb{E}} \underset{u \sim \pi(x)}{\mathbb{E}} \left[ \bar{z}_2(x,u) \right] \right] + \lambda \cdot \sum_{i=1}^{|\Pi_{\mathsf{MDP}}^{\mathsf{det}}|} q_i \ln q_i \right),$$

where we denote by $\bar{z}_s = \mathbb{E}_{z_s \sim p_t} z_s$ the expected loss at time $s$. Now, we can encode the loss at time $s$ for every policy $\pi \in \Pi_{\mathsf{MDP}}^{\mathsf{det}}$ in a vector $\ell_s^{\mathsf{det}} \in [0,1]^{|\Pi_{\mathsf{MDP}}^{\mathsf{det}}|}$ where the $\pi^{th}$ coordinate $\ell_{s,\pi}^{\mathsf{det}}$ is the loss for policy $\pi$. Given this, we can show that the distribution $q_{\mathsf{RERM},t}$ is given by:

$$(q_{\mathsf{RERM},t})_\pi = \frac{\exp\left( \frac{-1}{\lambda} \sum_{s=1}^{t} \ell_{s,\pi}^{\mathsf{det}} \right)}{\sum_j \exp\left( \frac{-1}{\lambda} \sum_{s=1}^{t} \ell_{s,j}^{\mathsf{det}} \right)}.$$

Going forward, we drop the RERM term from the distribution $q_{\mathsf{RERM},t}$ for ease of readability. In addition, the boundedness of the loss function $|\ell_{s,\pi}^{\mathsf{det}}| \le 1$ ensures that the RERM solutions satisfy the following stability property:

$$\|q_t - q_{t+1}\|_1 \le \frac{1}{\lambda}. \tag{32}$$

Given the above stability, one can also obtain a bound on the action distribution between the randomized policy $\pi_t = \mathbb{E}_{\pi \sim q_t}[\pi]$ and the corresponding $\pi_{t+1}$:

$$\|\pi_t(x) - \pi_{t+1}(x)\|_1 = \left\| \underset{\pi \sim q_t}{\mathbb{E}} [f(x)] - \underset{\pi \sim q_{t+1}}{\mathbb{E}} [f(x)] \right\|_1 = \|q_t - q_{t+1}\|_1 \le \frac{1}{\lambda},$$

where the second equality follows from the fact that $\|\pi(x)\|_1 = 1$ since they are distributions over the action space $\mathcal{U}$. Now, following a similar calculation as Lemma 5.2 in [11], we can obtain a bound on the variation in state distributions while playing policies $q_{1:t-1}$ as compared to the steady state distribution $d_{q_t}$.

$$\|d[q_{1:t-1}] - d_{q_t}\|_1 \le \frac{2\tau^2}{\lambda} + 2e^{-t/\tau}.$$

With this bound in place, we can now bound the ergodic stability parameters $\beta_{\mathsf{RERM},t}^*$ for the ERM procedure as

$$
\begin{aligned}
\beta_{\mathsf{RERM},t}^* &= \left| \mathbb{E}\left[ \ell(\pi_t, x_t[\pi_{1:t-1}], z) \right] - \ell_*^\Phi(\pi_t, z) \right| \\
&= \left| \underset{x \sim d[q_{1:t-1}]}{\mathbb{E}} \bar{z}_{\pi_t}(x) - \underset{x \sim d_{q_t}}{\mathbb{E}} \bar{z}_{\pi_t}(x) \right| \\
&\le \frac{2\tau^2}{\lambda} + 2e^{-t/\tau}.
\end{aligned}
\tag{33}
$$

**Bound on the value.** Having established bounds on the mixing gap and the RERM ergodic stability paramters, we now proceed to obtain the requisite bound on the value $\mathcal{V}_{\mathsf{MDP},T}(\Pi_{\mathsf{MDP}}, \mathcal{Z}, \Phi, \ell)$.

$$
\begin{aligned}
\mathcal{V}_{\mathsf{MDP},T}(\Pi_{\mathsf{MDP}}, \mathcal{Z}, \Phi, \ell) &\overset{(i)}{\le} \sum_{t=1}^{T} \beta_{\mathsf{RERM},t}^* + 2\mathfrak{R}_T^{\mathsf{seq}}(\ell_*^\Phi \circ \Pi_{\mathsf{MDP}}) + \sup_{\pi \in \Pi} \sum_{t=1}^{T} \left| \ell^\Phi(\pi, z_t, t) - \ell_*^\Phi(\pi, z) \right| + \lambda S \log A \\
&\overset{\mathsf{Eq.\ (31)}}{\le} \sum_{t=1}^{T} \beta_{\mathsf{RERM},t}^* + 2\mathfrak{R}_T^{\mathsf{seq}}(\ell_*^\Phi \circ \Pi_{\mathsf{MDP}}) + 2\tau e^{1/\tau} + \lambda S \log A \\
&\overset{\mathsf{Eq.\ (33)}}{\le} \frac{2\tau^2}{\lambda} T + 2\mathfrak{R}_T^{\mathsf{seq}}(\ell_*^\Phi \circ \Pi_{\mathsf{MDP}}) + 2\tau(1 + e^{1/\tau}) + \lambda S \log A \\
&\overset{(ii)}{\le} 2\tau\sqrt{TS \log A} + 2\mathfrak{R}_T^{\mathsf{seq}}(\ell_*^\Phi \circ \Pi_{\mathsf{MDP}}) + 2\tau(1 + e^{1/\tau})
\end{aligned}
$$

where (i) follows since the entropy over the class $\Pi_{\mathsf{MDP}}^{\mathsf{det}}$ is upper bounded by $\log \Pi_{\mathsf{MDP}}^{\mathsf{det}}$, and (ii) follows by setting $\lambda = \tau\sqrt{\frac{T}{S \log A}}$. Finally, bounding the sequential Rademacher complexity of the finite loss class $\ell_*^\Phi \circ \Pi_{\mathsf{MDP}}$ by $2\sqrt{ST \log(A)}$ completes the proof of the corollary. $\qquad\square$

### D.3 Online linear quadratic regulator

The online Linear Quadratic Regulator (LQR) setup studied in this section was first studied in [10]. The setup consists of a LQ system - with linear dynamics and quadratic costs - where the cost functions can be adversarial in nature. The comparator class $\Pi_{\mathsf{LQR}}$ comprises a subset of linear policies $K$ which satisfy the following strong stability property.

**Definition 6** (Strongly Stable Policy). *A policy $K$ is $(\kappa, \gamma)$-strongly stable (for $\kappa > 0$ and $0 < \gamma < 1$) if $\|K\|_2 \leq \kappa$, and there exists matrices $L$ and $H$ such that $A + BK = HLH^{-1}$, with $\|L\|_2 \leq 1 - \gamma$ and $\|H\|_2 \|H^{-1}\|_2 \leq \kappa$.*

The policy class $\Pi_{\mathsf{LQR}}$ is then defined as $\Pi_{\mathsf{LQR}} = \{K \mid K \text{ is } (\kappa, \gamma) - \text{strongly stable}\}$. Given this policy class, the sequential protocol for this game proceeds as follows, starting from state $x_0 = 0$

On round $t = 1, \ldots, T$,

- the learner selects a policy $K_t \in \Pi_{\mathsf{LQR}}$ and the adversary selects instance $z_t \in \mathcal{Z} = (Q_t, R_t)$ such that $Q_t \geq 0, R_t \geq 0$ and $\operatorname{tr}(Q_t), \operatorname{tr}(R_t) \leq C$
- the learner receives loss $\ell(\pi_t, x_t, z_t) = x_t^\top Q_t x_t + u_t^\top R_t u_t$
- the state of the system transitions to $x_{t+1} = A x_t + B u_t + w_t$

where we assume that the stochastic noise $w_t \sim \mathcal{N}(0, W)$ with $\|W\|_2 \leq \sigma_w$, $\operatorname{tr}(W) \leq \Psi_w$ and $W \geq \tau_w I$. The transition matrices $A$ and $B$, as well as the noise covariance matrix $W$ are assumed to be known to both the learner and the adversary in advance. Given this setup, the stationary loss is given by

$$\ell_*^\Phi(K, z) = \langle Q + K^\top RK, X_K \rangle = \operatorname{tr}[(Q + K^\top RK) X_K],$$
$$\text{where} \quad X_K = (A + BK) X_K (A + BK)^\top + W. \tag{34}$$

The following lemma establishes certain structural properties of the stationary loss, namely, boundedness over the policy class $\Pi_{\mathsf{LQR}}$ and Lipschitzness with respect to the operator norm.

**Lemma 2.** *The loss function $\ell_*^\Phi : \Pi_{\mathsf{LQR}} \times \mathcal{Z} \mapsto \mathbb{R}_+$ described in equation (34) satisfies*

$$\ell_*^\Phi(K, z) \leq B_{\mathsf{max}} \quad \text{for all } K \in \Pi_{\mathsf{LQR}}, \ z \in \mathcal{Z}$$
$$|\ell_*^\Phi(K_1, z) - \ell_*^\Phi(K_2, z)| \leq L_{\mathsf{Lip}} \|K_1 - K_2\|_2 \quad \text{for all } K_1, K_2 \in \Pi_{\mathsf{LQR}}, \ z \in \mathcal{Z},$$

*where $B_{\mathsf{max}} := C(1 + \kappa^2) \frac{\sigma_w \kappa^2}{\gamma}$ and $L_{\mathsf{Lip}} := 4C(1 + \kappa^2) \frac{\sigma_b \kappa^5 \sigma_w}{\gamma^2}$.*

We defer the proof of the lemma to the end of section and now proceed to obtain an upper bound on the value $\mathcal{V}_{\mathsf{LQR}, T}$ for the above problem.

**Corollary 8** (Online Linear Quadratic Regulator). *For the online LQR sequential game, the value $\mathcal{V}_{\mathsf{LQR}, T}$ is bounded as*

$$\mathcal{V}_{\mathsf{LQR}, T}(\Pi_{\mathsf{LQR}}, \mathcal{Z}, \Phi, \ell) \leq \mathcal{O}\left(\sqrt{T \log(T)}\right),$$

*where the $\mathcal{O}$ notation hides the dependence of the bound on problem-specific parameters (see equation (40) for the exact dependencies).*

*Proof.* As before, our strategy is to establish upper bounds on the mixing gap and the RERM ergodic stability parameter for the LQR problem, and using these with Corollary 5 to establish an upper bound on the value $\mathcal{V}_{\mathsf{LQR}, T}$.

**Existence of stationary loss.** Consider any stable policy $K \in \Pi_{\mathsf{LQR}}$. It is well known that a repeated application of the policy $K$ in the linear dynamics ensures that the state $x_t$ converges to a steady-state distribution, that is, the distribution of $x_t$ and $(A + BK)x_t + w_t$ is the same. Since the noise $w_t$ is assumed to be $\mathcal{N}(0, W)$, the steady-state distribution will also be a normal distribution with mean 0 and steady-state covariance $X_K$ satisfying the following recurrence equation:

$$X_K = (A + BK) X_K (A + BK)^\top + W \quad \text{or equivalently} \quad X_K = \sum_{s=0}^{\infty} (A + BK)^s W ((A + BK)^s)^\top,$$

and the corresponding steady-state loss is given by:

$$\ell_*^\Phi(K, z) = \langle Q + K^\top RK, X_K \rangle = \operatorname{tr}[(Q + K^\top RK) X_K].$$

**Bound on mixing gap.** We now proceed to obtain upper bounds on the mixing gap for this problem instance. Going forward, we define $X_{K,t}$ to be the state-covariance matrix at time $t$ when policy $K$ has been used for all preceding timesteps. For the purpose of readability, we will drop the dependence of the covariance matrix on the underlying policy $K$ when it is clear from the context. We begin by looking at the convergence of $X_t$ to the stationary matrix $X$:

$$\|X_t - X\|_2 = \left\| \sum_{s=0}^{t-1} (A+BK)^s W (A+BK)^s)^\top - \sum_{s=0}^{\infty} (A+BK)^s W (A+BK)^s)^\top \right\|_2$$

$$= \left\| \sum_{s=t}^{\infty} (A+BK)^s W (A+BK)^s)^\top \right\|_2$$

$$\overset{(i)}{\le} \sigma_w \sum_{s=t}^{\infty} \kappa^2 (1-\gamma)^{2s}$$

$$\le \frac{\sigma_w \kappa^2 (1-\gamma)^{2t}}{\gamma}$$

where (i) follows from the fact that $\|A+BK\|^s \le \kappa(1-\gamma)^s$ from the strong-stability of $K$. The above analysis shows that the covariance matrix $X_t$ converges to its stationary distribution exponentially fast. One can also obtain a bound similar to above on $\mathrm{tr}(X - X_t)$ with $\sigma_w$ replaced by $\Psi_w$. Having established this convergence, we establish a bound on the mixing gap as

$$|\mathbb{E}[\ell^\Phi(\pi, z, t)] - \ell_*^\Phi(\pi, z)| = |\langle Q + K^\top R K, X_t - X \rangle|$$

$$\le (\sigma_q + \kappa^2 \sigma_r) \cdot \mathrm{tr}(X - X_t)$$

$$\le (\sigma_q + \kappa^2 \sigma_r) \cdot \frac{\Psi_w \kappa^2 (1-\gamma)^{2t}}{\gamma}. \tag{35}$$

Since the above bound is independent of the underlying policy $K$, we have thus established a bound on the mixing gap for the policy class $\Pi_{\mathsf{LQR}}$.

**Regularized ERMs.** We now define the class of RERM's we use for the function class $\Pi_{\mathsf{LQR}}$. Instead of working with a fixed regularization function, we shall look at random perturbations as regularizations. Such an idea is popular in the study of online learning algorithms and is often termed as Follow the Perturbed Leader (FTPL); for a detailed study, see [24, 15]. Thus, the regularized ERM solutions at time $t$ are given by:

$$K_{t,\sigma} = \underset{K \in \Pi_{\mathsf{LQR}}}{\mathrm{argmin}} \left( \sum_{s=1}^{t} \mathbb{E}_{z_s \sim p_s} \left[ \langle Q_s + K^\top R K, X_K \rangle \right] - \langle \sigma, K \rangle \right),$$

where $\sigma \in \mathbb{R}^{k \times d}$ such that each coordinate of $\sigma \sim \mathrm{Exp}(\lambda)$, the exponential distribution with parameter $\lambda > 0$. It was established by [27] that if each of the loss function above is $L_{\mathsf{Lip}}$-Lipschitz, the iterates produced by the FTPL strategy above satisfy:

$$\mathbb{E}_\sigma \left[ \|K_{t,\sigma} - K_{t+1,\sigma}\|_1 \right] \le c\lambda \cdot L_{\mathsf{Lip}} (kd)^2 \kappa := \lambda_K,$$

where the norm above is defined element-wise. In Lemma 2, we establish that the losses given by $\ell_*^\Phi(\pi, z)$ are indeed Lipschitz over the space of policies $\Pi_{\mathsf{LQR}}$. With these set of regularized empirical minimizers, we proceed to now bound the ergodic stability parameters of these regularized ERM's, each one of which is strongly-stable.

**Sequential strong-stability of solutions.** We first establish that the set of RERM solutions produced by the algorithm satisfy the sequential strong-stability property (see [10] for details) with the appropriate parameters. Note that since each of the $K_t$ (we drop the dependece on the random noise $\sigma$) belongs to the class $\Pi_{\mathsf{LQR}}$, we have that $\|K_t\|_2 \le \kappa$.

Let $X_t := X_{K_t}$ be the steady-state covariance of the $t^{th}$ solution and $\hat{X}_t$ denote the covariance of the state reached when policies $\{K_1, \ldots, K_{t-1}\}$ are applied at the first $t$ timesteps. Consider the following decomposition for $A + BK_t$:

$$A + BK_t = H_t L_t H_t^{-1} \quad \text{where} \quad L_t = X_t^{-1/2}(A+BK_t)X_t^{-1/2}, \ H_t = X_t^{1/2}.$$

**Bound on $\|H_t\|_2$ and $\|H_t^{-1}\|_2$.** Using the recursive definition of $X_t$, we have:

$$\|X_t\|_2 = \left\| \sum_{s=0}^{\infty} (A + BK_t)^s W((A + BK)^s)^\top \right\|_2 \leq \frac{\sigma_w \kappa^2}{\gamma} \tag{36}$$

The above equation allows us to bound $\|H_t\|_2 \leq \kappa \sqrt{\sigma_w/\gamma} = \beta_h$. Also, by the definition of the matrix $X_t$, we have that $X \geq W$ and hence $\|H_t^{-1}\| \leq 1/\sqrt{\tau_w} = 1/\alpha_h$. Define $\tilde{\kappa} = \beta_h/\alpha_h$ and note that $\tilde{\kappa} \geq \kappa$.

**Bound on $\|L_t\|_2$.** Starting from the recursive definition of $X_t$, we have,

$$\begin{aligned}
I &= X_t^{-1/2}(A + BK)X_t(A + BK)^\top X_t^{-1/2} + X_t^{-1/2} W X_t^{-1/2} \\
&\geq L_t L_t^\top + \tau_w X_t^{-1} \\
&\geq L_t L_t^\top + \frac{\tau_w \gamma}{\sigma_w \max(\kappa^2, 1)} I \,,
\end{aligned}$$

which implies that $\|L_t\| \leq 1 - \tilde{\gamma}$ where $\tilde{\gamma} = \frac{\tau_w \gamma}{2\sigma_w \max(\kappa^2, 1)}$.

**Bound on $\|X_t - X_{t+1}\|$.** As before, we begin with the recursive definitions of $X_t$ and $X_{t+1}$ to get:

$$\begin{aligned}
X_{t+1} - X_t &= (A + BK_{t+1})X_{t+1}(A + BK_{t+1})^\top - (A + BK_t)X_t(A + BK_t)^\top \\
&= (A + BK_{t+1})(X_{t+1} - X_t)(A + BK_{t+1})^\top + \underbrace{B\Delta_t X_t(A + BK_{t+1})^\top}_{T_1} + \underbrace{(A + BK_t)(B\Delta_t)^\top}_{T_2} \\
&= \sum_{s=0}^{\infty} (A + BK_{t+1})^s (T_1 + T_2)((A + BK_{t+1})^s)^\top \,,
\end{aligned}$$

where $\Delta_t = K_{t+1} - K_t$. Taking norms on both sides, we get:

$$\|X_{t+1} - X_t\|_2 \leq \frac{2\sigma_b \kappa^5 \sigma_w}{\gamma^2} \|\Delta_t\|_2. \tag{37}$$

**Bound on $\|H_{t+1}^{-1} H_t\|_2$.** Recall that $H_t = X_t^{1/2}$. In order to bound the required term, we proceed as follows:

$$\begin{aligned}
\mathbb{E}\|X_{t+1}^{-1/2} X_t^{1/2}\|_2^2 &= \mathbb{E}\|X_{t+1}^{-1/2} X_t X_{t+1}^{-1/2}\| \\
&\leq \mathbb{E}\|X_{t+1}^{-1/2} X_{t+1} X_{t+1}^{-1/2}\|_2 + \mathbb{E}\|X_{t+1}^{-1/2}(X_{t+1} - X_t)X_{t+1}^{-1/2}\| \\
&\leq 1 + \frac{\mathbb{E}\|X_{t+1} - X_t\|_2}{\tau_w} \\
&\leq 1 + \frac{2\sigma_b \kappa^5 \sigma_w}{\tau_w \gamma^2} \lambda_K \\
&\overset{(i)}{\leq} 1 + \tilde{\gamma}
\end{aligned}$$

where we bound the term $\|X_{t+1} - X_t\|_2$ using Eq. (37) and (i) follows by setting $\lambda \leq \frac{\tilde{\gamma}\gamma^2 \tau_w}{c\sigma_b \sigma_w L_{\text{Lip}} \kappa^6 (kd)^2}$. Finally, using the fact that $\sqrt{1+x} \leq 1 + x/2$ for $x \in [0, 1]$, we have that $\mathbb{E}\|H_{t+1}^{-1} H_t\|_2 \leq 1 + \tilde{\gamma}/2$.

**Ergodic stability parameters.** We now proceed to obtain an upper bound on the ergodic stability parameters. Before doing so, we obtain some auxiliary results which will be useful in establishing the final bound.

**Bound on $\|\hat{X}_t - X_t\|_2$.** We will now obtain a bound on the difference between the observed covariance $\hat{X}_t$ when a sequence of ERMs are played and the steady-state covariance matrix $X_t$. Let us set some notation before we begin with bounding this.

$$\Delta_{x,t} := H_t^{-1}(\hat{X}_t - X_t)(H_t^{-1})^\top \quad \text{and} \quad \mathbb{E}_\sigma\|X_t - X_{t+1}\|_2 \leq \tilde{\lambda}.$$

We then have the following recursion for the term $\Delta_{x,t}$ with the expectation with respect to the sampling of the noise variable $\sigma$:

$$\mathbb{E}\|\Delta_{x,t+1}\|_2 \leq \mathbb{E}\|(H_{t+1}^{-1}H_t L_t)\Delta_{x,t}(H_{t+1}^{-1}H_t L_t)^\top\|_2 + \mathbb{E}\|(H_{t+1}^{-1})(X_t - X_{t+1})((H_{t+1}^{-1})^\top\|_2$$

$$\leq \mathbb{E}\|L_t\|_2^2\|H_{t+1}^{-1}H_t\|_2^2\|\Delta_{t,x}\|_2 + \frac{\tilde{\lambda}}{\alpha^2}$$

$$\overset{(i)}{\leq} \left(1 - \frac{\tilde{\gamma}}{2}\right)^2 \mathbb{E}\|\Delta_{x,t}\|_2 + \frac{\tilde{\lambda}}{\alpha^2}$$

$$\leq e^{-\tilde{\gamma}t}\|\Delta_{x,1}\|_2 + \frac{\tilde{\lambda}}{\alpha^2\tilde{\gamma}}\,,$$

where (i) follows from the bound on $\|L_t\| \leq 1-\tilde{\gamma}$ and $\|H_{t+1}^{-1}H_t\|_2 \leq (1+\tilde{\gamma}/2)$. Substituting the value for $\Delta_{x,t}$ in the above bound, we get that:

$$\mathbb{E}\|X_{t+1} - \hat{X}_{t+1}\|_2 \leq \frac{\beta_h^2}{\alpha_h^2}\left(e^{-\tilde{\gamma}t}\mathbb{E}\|\hat{X}_1 - X_1\| + \frac{\tilde{\lambda}}{\tilde{\gamma}}\right). \tag{38}$$

Let us now bound the ergodic stability parameters $\beta_{\mathsf{RERM},t}^*$ as

$$\beta_{\mathsf{RERM},t}^* = \left|\mathbb{E}_{\sigma}\mathbb{E}_{w}[\ell(\pi_t, x_t[\pi_{1:t-1}], z)] - \mathbb{E}_{\sigma}[\ell_*^\Phi(\pi_t, z)]\right|$$

$$= \left|\mathbb{E}_{\sigma}\left[\mathrm{tr}((Q + K_{t,\sigma}^\top R K_{t,\sigma}^\top)(\hat{X}_{t,\sigma} - X_{t,\sigma}))\right]\right|$$

$$\leq d(\sigma_q + \kappa^2\sigma_r)\mathbb{E}_{\sigma}\|\hat{X}_{t,\sigma} - X_{t,\sigma}\|_2$$

$$\overset{\text{Eq. (38)}}{\leq} d(\sigma_q + \kappa^2\sigma_r) \cdot \frac{\beta_h^2}{\alpha_h^2}\left(e^{-\tilde{\gamma}t}\mathbb{E}\|\hat{X}_1 - X_1\| + \frac{\tilde{\lambda}}{\tilde{\gamma}}\right)$$

$$\leq d(\sigma_q + \kappa^2\sigma_r) \cdot \frac{\beta_h^2}{\alpha_h^2}\left(e^{-\tilde{\gamma}t} \cdot \frac{2\sigma_w\kappa^2}{\gamma} + \lambda \cdot \frac{c\sigma_b\kappa^6\sigma_w d^2 k^2 L_{\mathsf{Lip}}}{\tilde{\gamma}\gamma^2}\right), \tag{39}$$

where $\lambda > 0$ is a free parameter corresponding to the noise in the perturbation $\sigma$.

**Bound on the value.** Having established upper bounds on the mixing gap and the ergodic stability parameters, we now bound the value $\mathcal{V}_{\mathsf{LQR},T}$ as

$$\mathcal{V}_{\mathsf{LQR},T}(\Pi_{\mathsf{LQR}}, \mathcal{Z}, \Phi, \ell) \overset{(i)}{\leq} \sum_{t=1}^{T} \beta_{\mathsf{RERM},t}^* + 2\mathfrak{R}_T^{\mathsf{seq}}(\ell_*^\Phi \circ \Pi_{\mathsf{LQR}}) + \sup_{\pi \in \Pi_{\mathsf{LQR}}}\sum_{t=1}^{T}\left|\ell^\Phi(\pi, z_t, t) - \ell_*^\Phi(\pi, z)\right| + \frac{\kappa k d}{\lambda}$$

$$\overset{\text{Eq. (35)}}{\leq} \sum_{t=1}^{T} \beta_{\mathsf{RERM},t}^{\mathsf{LQR}} + 2\mathfrak{R}_T^{\mathsf{seq}}(\ell_*^\Phi \circ \Pi_{\mathsf{LQR}}) + (\sigma_q + \kappa^2\sigma_r) \cdot \frac{\Psi_w\kappa^2}{\gamma^2} + \frac{\kappa k d}{\lambda}$$

$$\overset{\text{Eq. (39)}}{\leq} d(\sigma_q + \kappa^2\sigma_r) \cdot \frac{\beta_h^2}{\alpha_h^2}\left(\frac{2\sigma_w\kappa^2}{\gamma^2} + \lambda T \cdot \frac{c\sigma_b\kappa^6\sigma_w d^2 k^2 L_{\mathsf{Lip}}}{\tilde{\gamma}\gamma^2}\right)$$

$$+ 2\mathfrak{R}_T^{\mathsf{seq}}(\ell_*^\Phi \circ \Pi_{\mathsf{LQR}}) + (\sigma_q + \kappa^2\sigma_r) \cdot \frac{\Psi_w\kappa^2}{\gamma^2} + \frac{\kappa k d}{\lambda} \tag{40}$$

where (i) follows from the fact that $\mathbb{E}[\sigma_i] = 1/\lambda$.

To obtain a bound on the sequential Rademacher complexity of the class, observe the the matrices $K \in \mathbb{R}^{k \times d}$. Also, by Lemma 2, we have that the loss $\ell_*^\Phi$ is bounded by $B_{\mathsf{max}}$ and Lipschitz with respect to policies $K$ with constant $L_{\mathsf{Lip}}$. Using a standard covering number argument, one can get an $\epsilon$-net of the class $\Pi_{\mathsf{LQR}}$ in the frobenius norm with almost $O(dk(\frac{1}{\epsilon})^{dk})$ elements. Given this cover, one can upper bound the complexity as

$$\mathfrak{R}_T^{\mathsf{seq}}(\ell_*^\Phi \circ \Pi_{\mathsf{LQR}}) \leq cB_{\mathsf{max}}\sqrt{kd \cdot T \log(kdTL_{\mathsf{Lip}})}$$

for some universal constant $c > 0$. Setting $\lambda = O(1/\sqrt{T})$ concludes the proof of the corollary. $\square$

### D.3.1 Proof of Lemma 2

We establish both parts of the claim separately.

**Boundedness of stationary loss.** Consider the loss $\ell_*^\Phi$ given by

$$\ell_*^\Phi(K, z) = \text{tr}[(Q + K^\top R K) X_K]$$
$$\overset{(i)}{\leq} C(1 + \kappa^2)\|X_K\|_2$$
$$\overset{(ii)}{\leq} C(1 + \kappa^2)\frac{\sigma_w \kappa^2}{\gamma},$$

where inequality (i) follows from an applicaion of von Neumann's trace inequality and the trace bounds on the matrices $Q$ and $R$, and step (ii) follows from equation (36).

**Lipschitzness of stationary loss.** For any two matrices $K_1, K_2 \in \Pi_{\mathsf{LQR}}$ and instance $z \in \mathcal{Z}$, consider the difference between the stationary losses

$$|\ell_*^\Phi(K_1, z) - \ell_*^\Phi(K_2, z)| \leq |\text{tr}[Q(X_{K_1} - X_{K_2})]| + |\text{tr}[R(K_1 X_{K_1} K_1^\top - K_2 X_{K_2} K_2^\top)]|$$
$$\leq C\left((1 + \kappa^2)\|X_{K_1} - X_{K_2}\|_2 + \frac{2\kappa^3 \sigma_w}{\gamma}\|K_1 - K_2\|_2\right)$$
$$\overset{(i)}{\leq} 4C(1 + \kappa^2)\frac{\sigma_b \kappa^5 \sigma_w}{\gamma^2}\|K_1 - K_2\|_2 ,$$

where step (i) follows from equation (37). This concludes the proof. $\qquad\square$

## D.4 Online adversarial tracking

The problem of online tracking of adversarial targets in Linear Quadratic Regulators was first posed in Abbasi et al. [1]. The problem setup involves a state space given by $\mathbb{R}^d$ and a action space $\mathbb{R}^k$. The sequential game proceeds as follows starting from state $x_1 = 0$

On round $t = 1, \ldots, T$,

- the learner selects a policy $\pi_t \in \Pi_{\mathsf{track}}$ and adversary selects $z_t \in \mathcal{Z} = \mathbb{R}^d$ such that $\|z_t\|_2 \leq c_z$
- the learner receives loss $\ell(\pi_t, x_t, z_t) = (x_t - z_t)^\top Q(x_t - z_t) + \pi_t(x_t)^\top \pi_t(x_t)$
- the state of the system transitions to $x_{t+1} = Ax_t + Bu_t$

where the matrices $A, B, Q$ are known in advance to the learner and the adversary. In addition, the matrix $Q$ is positive definite, the pair $(A, B)$ is assumed to be controllable while the pair $(A, Q^{1/2})$ is assumed to be observable. The comparator policy class $\Pi_{\mathsf{track}}$ is assumed to be the following restricted class of linear policies:

$$\Pi_{\mathsf{track}} = \{\pi = (K, \eta) \mid \|A + BK\|_2 \leq \rho; \|K\|_2 \leq c_K; \|\eta\|_2 \leq c_\eta\} ,$$

such that the action is given by $u_t = K_t x_t + \eta_t$. For this setup, as we establish later, the stationary loss for any policy $\pi = (K, \eta)$ is given by:

$$\ell_*^\Phi(\pi, z) = (x_*^\pi - z)^\top Q(x_*^\pi - z) + \|Kx_*^\pi + \eta\|_2^2, \quad \text{where} \quad x_*^\pi = (I - (A + BK))^{-1}B\eta$$

Given these preliminaries, we obtain a bound on the value $\mathcal{V}_{\mathsf{tar},T}$ through the following corollary.

**Corollary 9** (Online Tracking). *For the online adversarial tracking sequential game, the value $\mathcal{V}_{\mathsf{tar},T}$ is bounded by:*

$$\mathcal{V}_{\mathsf{tar},T}(\Pi_{\mathsf{track}}, \mathcal{Z}, \Phi) \leq \mathcal{O}\left(\sqrt{T\log(T)}\right) ,$$

*where the $\mathcal{O}$ notation hides the dependence of the bound on problem-specific parameters (see equation (45) for the exact dependencies).*

In contrast to the result obtained above, [1] provide an algorithm for which the regret for the above problem is bounded by $\mathcal{O}(\log^2 T)$. Obtaining such fast rates in our general framework is an interesting open problem.

*Proof of Corollary 9.* Our general strategy is to obtain bounds on the the mixing gap and the ergodic stability paramters for certain regularized ERMs. We then use these upper bounds together with Corollary 5 to establish the required upper bound.

**Bound on mixing gap.** Consider any policy $\pi = (K, \eta)$. We are interested in obtaining a bound on the mixability for this function as:

$$\left| \ell^{\Phi}(\pi, z, t) - \ell_*^{\Phi}(\pi, z) \right| \leq \beta_{\pi, t}.$$

Let us abbreviate the state $x_t[\pi^{(t-1)}]$ by $x_t^{\pi}$. If we run any policy with the linear dynamics, a steady state $x_*^{\pi}$ is reached with

$$x_*^{\pi} = (A + BK)x_*^{\pi} + B\eta \quad \text{and therefore} \quad x_*^{\pi} = \underbrace{(I - (A + BK))^{-1} B}_{:= M_K} \eta = M_K \eta.$$

Then, the corresponding loss at this stationary point is given as

$$\ell_*^{\Phi}(\pi, z) = (x_*^{\pi} - z)^{\top} Q (x_*^{\pi} - z) + \|K x_*^{\pi} + \eta\|_2^2.$$

In order to obtain a bound on the mixing gap, we analyse the convergence of the state $x_{t+1}^{\pi}$ to the stationary state $x_*^{\pi}$.

$$\|x_{t+1}^{\pi} - x_*^{f}\|_2 = \| \sum_{s=0}^{\infty} (A - BK)^s B\eta - \sum_{s=0}^{t-1} (A - BK)^s B\eta \|_2$$

$$\overset{(i)}{\leq} \rho^t \|M_K \eta\|_2 \,,$$

where (i) follows from the assumption that $\|A + BK\|_2 \leq \rho$.

Next, we consider a bound on the norm of the state $x_t^{\pi}$ that is reached by any policy.

$$\|x_{t+1}^{\pi}\|_2 = \|(A + BK)x_t^{\pi} + B\eta\|_2$$

$$\overset{(i)}{=} \| \sum_{s=1}^{t} (A + BK)^{t-s} B\eta \|_2$$

$$\overset{(ii)}{\leq} \frac{\|B\|c_{\eta}}{1 - \rho} := c_x,$$

where (i) follows from recursively applying the definition of the state evolution and the fact that $x_1 = 0$, and (ii) follows from the assumption that $\|A + BK\|_2 \leq \rho$. Having established the above, we now proceed to obtain a bound on the mixing gap as

$$\left| \ell^{\Phi}(\pi, z, t) - \ell_*^{\Phi}(\pi, z) \right| = \left| (x_t^{\pi} - z)^{\top} Q(x_t^{\pi} - z) + \|K x_t^{\pi} + \eta\|_2^2 - (x_*^{\pi} - z)^{\top} Q(x_*^{\pi} - z) + \|K x_*^{\pi} + \eta\|_2^2 \right|$$

$$\overset{(i)}{\leq} \left| (x_t^{\pi} - x_*^{\pi})^{\top} Q(x_t^{\pi} - z) \right| + \left| (x_t^{\pi} - x_*^{\pi})^{\top} Q(x_*^{\pi} - z) \right| + \|K(x_t^{\pi} - x_*^{\pi})\|_2^2$$

$$+ 2\langle K x_*^{\pi} + \eta, K(x_t^{\pi} - x_*^{\pi}) \rangle$$

$$\leq 2\|Q\|(c_x + c_z) \cdot \|x_t^{\pi} - x_*^{\pi}\|_2 + c_K^2 \cdot \|x_t^{\pi} - x_*^{\pi}\|_2^2 + 2c_K(c_K c_x + c_{\eta}) \cdot \|x_t^{\pi} - x_*^{\pi}\|_2$$

$$\leq \rho^{t-1} \cdot \underbrace{c_x(2\|Q\|(c_x + c_z) + 2c_K(c_K c_x + c_{\eta}))}_{C_{\text{tar},1}} + \rho^{2(t-1)} \cdot \underbrace{c_x^2 c_K^2}_{C_{\text{tar},2}} , \qquad (41)$$

where (i) follows from adding and subtracting $x_*^{\pi}$ in both the terms follwed by an application of triangle inequality. For ease of presentation, let us represent the above using constants $C_{\text{tar},1}$ and $C_{\text{tar},2}$ with the knowledge that these depend on the underlying problem parameters but independent of the underlying policy $\pi$, that is,

$$\left| \ell^{\Phi}(\pi, z, t) - \ell_*^{\Phi}(\pi, z) \right| \leq \rho^{t-1} C_{\text{tar},1} + \rho^{2(t-1)} C_{\text{tar},2} \qquad (42)$$

**Ergodic stability parameters.** For obtaining a bound on the ergodic stability parameters, we require a few structural results for the loss $\ell_*^{\Phi}$ defined above. We present these next and defer their proofs to the end of the section.

**Lemma 3** (Equivalence of Tracking Cost). *Consider any policy $\pi = (K, \eta)$ and another stable matrix $K'$. There exists an $\eta'$ such that we have $\ell_*^{\Phi}((K, \eta), z) = \ell_*^{\Phi}((K', \eta'), z)$ such that*

$$\|\eta'\|_2 \leq 2c_{\eta} \left( \frac{\|B\|c_K}{(1 - \rho)} + 1 \right).$$

Thus going forward, we conside the ERM procedure on the class of functions $\Pi'_{\text{track}}(K)$, parameterized for a fixed stable policy $K$, where the bias

$$\eta \le c'_\eta := \max\left(2c_\eta\left(\frac{\|B\|c_K}{1-\rho}+1\right), c_\eta\frac{c_z\|Q\|\|B\|}{\sigma_Q\sigma_B^2(1-\rho)}\right),$$

with $\sigma_X$ denotes the smallest non-zero singular value of $X$. Note that the conclusions of Corollary 5 are still valid with the mixability parameters $\beta_{\Pi'_{\text{track}}(K),t}$. The next lemma establishes the stability of the ERM solutions obtained in consecutive rounds.

**Lemma 4.** *Fix any $\rho$-stable policy $K$. The ERM solutions $\pi_{\text{ERM},t} = (K, \eta_t)$ and $\pi_{\text{ERM},t+1} = (K, \eta_{t+1})$ satisfy the following stability bound:*

$$\|\eta_t - \eta_{t+1}\| \le \frac{2c_z}{t+1}\cdot\frac{c_\eta\|Q\|\|B\|}{\sigma_Q\sigma_B^2(1-\rho)} := \frac{\psi_\eta}{t+1}$$

Having established the stability bound above, one can proceed in a manner similar to [1, Lemma 8], one can establish that for $t > \rho\log(T)/(1-\rho)$

$$\left\|x_t[\pi_{\text{ERM},1:t-1}] - x_*^{\pi_{\text{ERM},t}}\right\|_2 \le \underbrace{\frac{\|B\|\psi_\eta}{1-\rho}\cdot\frac{2\log t}{t-\log t} + \rho^{t-1}\frac{\|B\|c'_\eta}{1-\rho}}_{\psi_{x,t}}. \tag{43}$$

Having established the above, we can now obtain a bound on the ergodic stability parameters for the ERM procedure for the online tracking problem. The calculation is similar to the one done for the mixing gap (see Eq. (41)).

$$\begin{aligned}\left|\ell(\pi_{\text{ERM},t}, x_t[\pi_{\text{ERM},1:t-1}], z_t) - \ell_*^\Phi(\pi_{\text{ERM},t}, z_t)\right| &\le 2\|Q\|(c_x + c_z)\cdot\left\|x_t[\pi_{\text{ERM},1:t-1}] - x_*^{\pi_{\text{ERM},t}}\right\|_2 \\ &\quad + c_K^2\cdot\left\|x_t[\pi_{\text{ERM},1:t-1}] - x_*^{\pi_{\text{ERM},t}}\right\|_2^2 \\ &\quad + 2c_K(c_Kc_x + c'_\eta)\cdot\left\|x_t[\pi_{\text{ERM},1:t-1}] - x_*^{\pi_{\text{ERM},t}}\right\|_2 \\ &\le \psi_{x,t}(2\|Q\|(c_x + c_z) + 2c_K(c_Kc_x + c'_\eta)) + \psi_{x,t}^2c_K^2 ,\end{aligned}$$

where we have substituted the bound for $\|x_t[\pi_{\text{ERM},1:t-1}] - x_*^{\pi_{\text{ERM},t}}\|$ from Eq. (43). Thus, we that the ERM ergodic stability parameters are

$$\beta_{\text{ERM},t}^* = \psi_{x,t}\cdot(2\|Q\|(c_x + c_z) + 2c_K(c_Kc_x + c_\eta)) + \psi_{x,t}^2c_K^2 . \tag{44}$$

**Bound on the value.** We now proceed to obtain a bound on the value $\mathcal{V}_{\text{tar},T}$, beginning from the statement of Corollary 5.

$$\begin{aligned}\mathcal{V}_{\text{tar},T}(\Pi_{\text{track}}, \mathcal{Z}, \Phi, \ell) &\le \sum_{t=1}^T\beta_{\text{ERM},t}^{\text{tar}} + 2\mathfrak{R}_T^{\text{seq}}(\ell_*^\Phi\circ\Pi'_{\text{track}}(K)) + \sup_{\pi\in\Pi_{\text{LQR}}}\sum_{t=1}^T\left|\ell^\Phi(\pi, z_t, t) - \ell_*^\Phi(\pi, z)\right| \\ &\stackrel{\text{Eq. (42)}}{\le} \sum_{t=1}^T\beta_{\text{ERM},t}^{\text{tar}} + 2\mathfrak{R}_T^{\text{seq}}(\ell_*^\Phi\circ\Pi'_{\text{track}}(K)) + \frac{C_{\text{tar},1}}{1-\rho} + \frac{C_{\text{tar},2}}{1-\rho^2} \\ &\stackrel{\text{Eq. (44)}}{\le} \frac{\rho\log T}{1-\rho}\left(\|Q\|(c_x + c_z)^2 + (c_Kc_x + c'_\eta)^2\right) + 2\mathfrak{R}_T^{\text{seq}}(\ell_*^\Phi\circ\Pi'_{\text{track}}(K)) + \frac{C_{\text{tar},1}}{1-\rho} + \frac{C_{\text{tar},2}}{1-\rho^2} \\ &\quad + \left(\frac{2\log^2 T\|B\|\psi_\eta}{1-\rho} + \frac{\|B\|c'_\eta}{(1-\rho)^2}\right)\cdot(2\|Q\|(c_x + c_z) + 4c_K(c_Kc_x + c_\eta)) ,\end{aligned} \tag{45}$$

where in the last inequality, we have upper buonded the lower order term $\psi_{x,t}^2$ by $\psi_{x,t}$.

Finally, one can obtain a bound on the sequential complexity by noting that the loss $\ell_*^\Phi$ is bounded since the state $\|x_*^\pi\|_2 \le c_x$ and is Lipschitz in the bias parameter $\eta$ with respect to the $ell_2$ norm. Using an argument similar to that from the proof of Corollary 8, we have

$$\mathfrak{R}_T^{\text{seq}}(\ell_*^\Phi\circ\Pi'_{\text{track}}(K)) \le O\left(\sqrt{dT\cdot\log(dT)}\right).$$

Substituting this bound in equation (45) establishes the corollary.  □

### D.4.1  Proof of Lemma 3

Let $V_{\pi,z}(x,u)$ represent the value function for state-action pair $(x,u)$ with respect to policy $\pi$ and loss function $z$. Following Lemma 12 from [1] we have that:

$$\ell_*^\Phi(\pi',z) - \ell_*^\Phi(\pi,z) = V_{\pi',z}(x_*^\pi, u_{\pi'}) - V_{\pi',z}(x_*^\pi, u_\pi).$$

The action taken by policy $\pi$ is given by $u_\pi = Kx_*^\pi + \eta$, while that taken by $\pi'$ is given by $K'x_*^\pi + \eta'$. If we set the value of $\eta'$ as:

$$\eta' = (K - K')x_*^\pi + \eta \quad \Rightarrow \quad \ell_*^\Phi(\pi',z) = \ell_*^\Phi(\pi,z).$$

Also, note that one can obtain an upper bound on the norm of $\eta'$ as $\|\eta'\|_2 \le 2c_x c_K + c_\eta$ using the bounds on the state $x_*^\pi$. $\qquad\square$

### D.4.2  Proof of Lemma 4

We begin by characterizing the ERM solution $\eta_t$ as follows:

$$
\begin{aligned}
\eta_t &= \operatorname*{argmin}_{\eta}\left(\sum_{t=1}^{T}\mathbb{E}_{z_t\sim p_t}\left[(x_*^\pi - z_t)^\top Q(x_*^\pi - z_t) + \|Kx_*^\pi + \eta\|_2^2\right]\right)\\
&= \operatorname*{argmin}_{\eta}\left(\sum_{t=1}^{T}\mathbb{E}_{z_t\sim p_t}\left[(x_*^\pi)^\top[Q + K^\top K]x_*^\pi - 2z_t Qx_*^\pi + \eta^\top\eta + \eta^\top Kx_*^\pi + (Kx_*^\pi)^\top\eta\right]\right)\\
&= \operatorname*{argmin}_{\eta}\left(\sum_{t=1}^{T}\mathbb{E}_{z_t\sim p_t}\left[\eta^\top\underbrace{(M_K^\top(Q + K^\top K)M_K + I + KM_K + M_K^\top K^\top)}_{W}\eta - 2z_t^\top QM_K\eta\right]\right)\\
&= W^{-1}\left(\frac{1}{t}\sum_{s=1}^{t}\mathbb{E}_{z_s\sim p_s}[z_s]\right)QM_K\,,
\end{aligned}
$$

where the last equality follows by minimizing the quadratic and the existence of the inverse because $W \succeq B^\top QB$. and the fact that $\eta$ does not lie in the null space of $B$ (it is always beeter to set it to zero in that case). This ensures that $\|\eta_t\|_2 \le c_\eta\frac{c_z\|Q\|\|B\|}{\sigma_Q\sigma_B^2(1-\rho)}$ and hence the policy $\pi_t = (K,\eta_t) \in \Pi'_{\text{track}}$. We can now obtain the stability bounds as:

$$
\begin{aligned}
\|\eta_t - \eta_{t+1}\|_2 &= \left\|W^{-1}\left(\frac{1}{t}\sum_{s=1}^{t}\mathbb{E}_{z_s\sim p_s}[z_s] - \frac{1}{t+1}\sum_{s=1}^{t+1}\mathbb{E}_{z_s\sim p_s}[z_s]\right)QM_K\right\|_2\\
&\le \frac{2c_z}{t+1}\cdot\frac{c_\eta\|Q\|\|B\|}{\sigma_Q\sigma_B^2(1-\rho)}\,,
\end{aligned}
$$

where the final inequality follows from using the bound on $\|M_K\|_2$ as well as the fact that $\|z\|_2 \le c_z$. This concludes the proof of the lemma. $\qquad\square$

## D.5  Online non-linear control

In this section, we look at a non-linear control problem: one formed by extending the LQR problem above to have non-linear deterministic dynamics. We parameterize the dynmaics using a non-linear function $\sigma_{\mathsf{NL}} : \mathbb{R}^d \mapsto \mathcal{X}$ as follows:

$$x_{t+1} = \sigma_{\mathsf{NL}}[Ax_t + Bu_t]\,,$$

We assume that the function $\sigma_{\mathsf{NL}}$ is 1-Lipschitz and $\|\sigma_{\mathsf{NL}}(x)\| \le c_x$ for some $c_x > 0$. This is done to ensure that the dynamics satisfy the ergodicity assumption. We now proceed to define the associated policy class $\Pi_{\mathsf{NL}}$ as

$$\Pi_{\mathsf{NL}} = \{\pi_\theta \mid \theta \in \mathbb{R}^{d_\theta}, \|\theta\|_2 \le c_\theta, \|[Ax + B\pi_\theta(x)] - [Ax' + B\pi_\theta(x')]\|_2 \le (1-\gamma)\|x - x'\|_2\}\,,$$

where the last condition on the function class establishes a stability condition. In addition, we assume that the function class $\Pi_{\mathsf{NL}}$ satisfies a Lipschitz property:

$$\|\pi_\theta(x) - \pi_{\theta'}(x)\|_2 \le L_\pi\|\theta - \theta'\|_2 \quad \text{for all} \quad x \in \mathcal{X}\,.$$

The above basically means that if two parameters $\theta, \theta'$ are close in the parameter space, then the policies parameterized by them are uniformly close for all states. Note that the class of linear policies $\Pi_{\mathsf{track}}$ (without the bias term) defined for the adversarial tracking problem satisfies the above properties. We next outline the learning protocol, with the game starting with $x_1 = 0$.

On round $t = 1, \ldots, T$,

- the learner selects policy $\pi_t \in \Pi_{\mathsf{NL}}$ and the adversary selects $z_t \in \mathcal{Z}$.
- the learner receives loss $\ell(\pi_t, x_t, z_t) \in [0, 1]$
- the state of the system transitions to $x_{t+1} = \sigma_{\mathsf{NL}}[Ax_t + Bu_t]$

For the above setup, we shortly establish that the stationary loss for any policy $\pi$ is given by

$$\ell_*^\Phi(\pi, z) = \ell(\pi, x_*^\pi, z) \quad \text{where} \quad x_*^\pi = \sigma_{\mathsf{NL}}[Ax_*^\pi + B\pi(x_*^\pi)], \tag{46}$$

where the existence of the fixed point is guaranteed by the stability assumption on the function class in conjunction with the Brouwer fixed-point theorem. In the following lemma, we show that the loss function $\ell_*^\Phi$ above is Lipschitz with respect to the parameter $\theta$.

**Lemma 5.** *The loss function $\ell_*^\Phi$ given in equation (46) satisfies*

$$|\ell_*^\Phi(\pi_{\theta_1}, z) - \ell_*^\Phi(\pi_{\theta_2}, z)| \leq \underbrace{\left( L_{l,\theta} + L_{l,x}\frac{\|B\|_2 L_\pi}{\gamma} \right)}_{L_{\mathsf{Lip}}} \|\theta_1 - \theta_2\|_2 \quad \text{for all } \pi_{\theta_1}, \pi_{\theta_2} \in \Pi_{\mathsf{NL}}, \ \ z \in \mathcal{Z}.$$

We prove the lemma at the end of the section. Taking this as given, we now establish the learnability of the function class $\Pi_{\mathsf{NL}}$ in the following corollary.

**Corollary 10** (Online Non-Linear Control). *Consider any value of $\lambda > 0$ and loss function $\ell$ which is $L_{l,x}$-Lipschitz in the state space and $L_{l,\theta}$-Lipschitz in the parameter space with respect to the $\ell_2$ norm. For the online non-linear control problem described above, we have that the value*

$$\mathcal{V}_{\mathsf{NL},T}(\Pi_{\mathsf{NL}}, \mathcal{Z}, \Phi) \leq \mathcal{O}\left( \sqrt{T \log(T)} \right),$$

*where the $\mathcal{O}$ notation hides the dependence of the bound on problem-specific parameters (see equation (52) for the exact dependencies).*

Notice that the above corollary establishes an upper bound of $\widetilde{\mathcal{O}}(\sqrt{T})$ for the value $\mathcal{V}_{\mathsf{NL},T}$. Thus, despite the fact that the setup does not have the nice structure of the LQR problem, we are able to establish the learnability of the class $\Pi_{\mathsf{NL}}$ in the online learning with dynamics framework.

*Proof of Corollary 10.* We begin by establishing a bound on the mixing gap for the class $\Pi_{\mathsf{NL}}$ as well as the ERM ergodic stability parameters. Throughout this section, we would often drop the dependence of the function $\pi_\theta$ on the underlying parameter $\theta$ when it is clear from the context.

**Bound on mixing gap.** Consider any policy $\pi \in \Pi_{\mathsf{NL}}$ and the associated stationary loss

$$\ell_*^\Phi(\pi, z) = \ell(\pi, x_*^\pi, z) \quad \text{where} \quad x_*^\pi = \sigma_{\mathsf{NL}}[Ax_*^\pi + B\pi(x_*^\pi)],$$

where the non-linearity $\sigma_{\mathsf{NL}}$ is applied element-wise to its arguments. Consider now the difference between the stationary and the counterfactual loss as

$$
\begin{aligned}
\left|\ell^\Phi(\pi, z, t) - \ell(\pi, x_*^\pi, z)\right| &\overset{(i)}{\leq} L_{l,x}\|x_t^\pi - x_*^\pi\|_2 \\
&\leq L_{l,x}\|\sigma_{\mathsf{NL}}[Ax_{t-1}^\pi + B\pi(x_{t-1}^\pi)] - \sigma_{\mathsf{NL}}[Ax_*^\pi + B\pi(x_*^\pi)]\|_2 \\
&\overset{(ii)}{\leq} L_{l,x}(1-\gamma)\|x_{t-1}^\pi - x_*^\pi\|_2 \\
&\overset{(iii)}{\leq} 2L_{l,x}c_x(1-\gamma)^{t-1},
\end{aligned}
\tag{47}
$$

where (i) follows from the $L_{l,x}$-Lipschitzness of the loss function, (ii) follows from the fact that $\pi \in \Pi_{\mathsf{NL}}$, and (iii) follows from the boundedness of the states.

**Regularized ERM.** Similar to the proof for Corollary 8, we consider the following FTPL based ERM

$$\theta_{t,\sigma} = \operatorname*{argmin}_{\theta \in \Theta_{\mathsf{NL}}} \left( \sum_{s=1}^{t} \underset{z_s \sim p_s}{\mathbb{E}} \left[ \ell_*^{\Phi}(\pi_\theta, z) \right] - \langle \sigma, \theta \rangle \right),$$

where $\sigma \in \mathbb{R}^{d_\theta}$ such that each coordinate of $\sigma \sim \mathrm{Exp}(\lambda)$, the exponential distribution with parameter $\lambda > 0$. We establish in Lemma 5 that the loss functions defined by $\ell_*^{\Phi}(\cdot, z)$ are $L_{\mathsf{NL}}$-Lipschitz in the parameter $\theta$ and hence the iterates satisfy

$$\underset{\sigma}{\mathbb{E}} \left[ \| \theta_{t,\sigma} - \theta_{t+1,\sigma} \|_1 \right] \leq c\lambda \cdot L_{\mathsf{NL}} (d_\theta)^2 \kappa := \lambda_\theta \;,$$

where the norm above is defined element-wise.

**Ergodic stability parameters.** With these set of regularized ERMs, we proceed to now bound the stability parameters of these solutions. Consider again the difference between the stationary and the instantaneous loss:

$$\left| \underset{\sigma}{\mathbb{E}} \left[ \ell(\pi_t, x_t[\pi_{1:t-1}]), z \right] - \underset{\sigma}{\mathbb{E}} \left[ \ell_*^{\Phi}(\pi_t, z) \right] \right| \leq L_{l,x} \underset{\sigma}{\mathbb{E}} \left[ \| x_t[\pi_{1:t-1}] - x_t^{\pi_t} \|_2 + \| x_*^{\pi_t} - x_t^{\pi_t} \|_2 \right] \;, \quad (48)$$

where the above inequality follows from the $L_{l,x}$ Lipschitz property of the loss function in the state space. We have dropped the dependence of $\pi$ on the noise perturbation $\sigma$, underlying parameter $\theta$ as well as the fact that these are RERM solutions. In order to obtain the stability parameters, we proceed to obtain a bound on the terms on the right.

**Bound on $\| x_*^{\pi_t} - x_t^{\pi_t} \|_2$.** The upper bound on this difference is similar to the one we obtained while bounding the mixing gap, the only difference being we have to handle the expectation with respect to the random perturbation $\sigma$. Consider,

$$\begin{aligned}
\underset{\sigma}{\mathbb{E}} \left[ \| x_*^{\pi_t} - x_t^{\pi_t} \|_2 \right] &= \underset{\sigma}{\mathbb{E}} \left[ \| \sigma_{\mathsf{NL}} [A x_*^{\pi_t} + B\pi_t(x_*^{K_t})] - \sigma_{\mathsf{NL}} [A x_{t-1}^{\pi_t} + B\pi_t(x_{t-1}^{\pi_t})] \|_2 \right] \\
&\leq (1-\gamma) \underset{\sigma}{\mathbb{E}} \left[ \| x_*^{K_t} - x_{t-1}^{K_t} \|_2 \right] \\
&\leq (1-\gamma)^{t-1} \cdot 2c_x \;,
\end{aligned} \quad (49)$$

where the sequence of inequalities follows since we have $\pi_{\theta,t,\sigma} \in \Pi_{\mathsf{NL}}$ for any sampling of the perturbation variables $\sigma$.

**Bound on $\| x_t[\pi_{1:t-1}] - x_t^{\pi_t} \|_2$.** Consider a parameter $\tau \geq 1$ to be specified later. We can then decompose the desired difference as follows:

$$\begin{aligned}
\| x_t[\pi_{1:t-1}] - x_t^{\pi_t} \|_2 &= \sum_{i=1}^{\tau} \left( \| x_t[\pi_1, \ldots, \pi_{t-i}, \pi_t, \ldots, \pi_t] - x_t[\pi_1, \ldots, \pi_{t-i-1}, K_t, \ldots, \pi_t] \|_2 \right) \\
&\quad + \| x_t[\pi_1, \ldots, \pi_{t-\tau-1}, \pi_t, \ldots, \pi_t] - x_t[\pi_t, \ldots, \pi_t] \|_2 \\
&\leq \sum_{i=1}^{\tau} \left( \| x_t[\pi_1, \ldots, \pi_{t-i}, \pi_t, \ldots, \pi_t] - x_t[\pi_1, \ldots, \pi_{t-i-1}, \pi_t, \ldots, \pi_t] \|_2 \right) + 2c_x(1-\gamma)^{\tau-1} \;,
\end{aligned}$$

where the last inequality follows from a similar calculation as in equation (49). We now focus on the terms in the summation above, focussing on a general term $i$. Let us redefine the state to be $x_0^i = x_{t-i}[\pi_1, \ldots, \pi_{t-i-1}]$. Now, denote by $\hat{x}_j = x_{t-i+j}[\pi_{t-i}, \pi_t, \ldots, \pi_t]$ to be the state reached when we select $\pi_{t-i}$ at the $(t-i)^{th}$ time instance, followed by $\pi_t$ for $j-1$ steps. Similarly, $\tilde{x}_j = x_{t-i+j}[\pi_t, \pi_t, \ldots, \pi_t]$ is the state reached when one begins from $x_0^i$ and selects $\pi_t$ for the next $j$ time steps. Bounding the sum above is equivalent to bounding the difference $\tilde{x}_i - \hat{x}_i$.

$$\begin{aligned}
\| \tilde{x}_i - \hat{x}_i \|_2 &= \| \sigma_{\mathsf{NL}} [A\tilde{x}_{i-1} + B\pi_t(\tilde{x}_{i-1})] - \sigma_{\mathsf{NL}} [A\hat{x}_{i-1} + B\pi_t(\hat{x}_{i-1})] \|_2 \\
&\leq (1-\gamma)^{i-1} \| \tilde{x}_1 - \hat{x}_1 \|_2 \\
&= (1-\gamma)^{i-1} \| \sigma_{\mathsf{NL}} [A x_0^i + B\pi_{t-i}(x_0^i)] - \sigma_{\mathsf{NL}} [A x_0^i + B\pi_t(x_0^i)] \|_2 \\
&\leq (1-\gamma)^{i-1} \cdot \| B \|_2 \cdot \| \pi_{t-i}(x_0^i) - \pi_t(x_0^i) \| \\
&\overset{(i)}{\leq} (1-\gamma)^{i-1} \cdot \| B \|_2 \cdot L_\pi \| \theta_{t-i} - \theta_t \| \\
&\leq i(1-\gamma)^{i-1} \cdot \| B \|_2 \cdot L_\pi \lambda_\theta
\end{aligned}$$

where (i) follows from the Lipschitz assumption on the function class in the parameter space. Setting $\tau = t$ and summing up the above inequalities, we get,

$$\|x_t[\pi_{1:t-1}] - x_t^{\pi_t}\|_2 \le \frac{\|B\|_2 \cdot L_\pi \lambda_\theta}{\gamma^2} + 2c_x(1-\gamma)^{t-1}. \tag{50}$$

Finally, substituting the bounds obtained in eq. (49) and eq. (50) in eq. (48), we get that:

$$\left| \mathbb{E}_\sigma \left[ \ell(\pi_t, x_t[\pi_{1:t-1}]), z \right] - \mathbb{E}_\sigma \left[ \ell_*^\Phi(\pi_t, z) \right] \right| \le L_{l,x} \left( \frac{\|B\|_2 \cdot L_\pi \lambda_\theta}{\gamma^2} + 4c_x(1-\gamma)^{t-1} \right) := \beta_{\mathsf{RERM},t}^* \tag{51}$$

**Bound on the value.**   Having established the mixing gap and the RERM stability paramters, we now upper bound the value $\mathcal{V}_{\mathsf{NL},T}(\Pi_{\mathsf{NL}}, \mathcal{Z}, \Phi, \ell)$

$$
\begin{aligned}
\mathcal{V}_{\mathsf{NL},T}(\Pi_{\mathsf{NL}}, \mathcal{Z}, \Phi, \ell) &\overset{(i)}{\le} \sum_{t=1}^T \beta_{\mathsf{RERM},t}^* + 2\mathfrak{R}_T^{\mathsf{seq}}(\ell_*^\Phi \circ \Pi_{\mathsf{NL}}) + \sup_{\pi \in \Pi_{\mathsf{NL}}} \sum_{t=1}^T \left| \ell_*^\Phi(\pi, z_t, t) - \ell_*^\Phi(\pi, z) \right| + \frac{c_\theta d_\theta}{\lambda} \\
&\overset{\text{Eq. (47)}}{\le} \sum_{t=1}^T \beta_{\mathsf{RERM},t}^* + 2\mathfrak{R}_T^{\mathsf{seq}}(\ell_*^\Phi \circ \Pi_{\mathsf{NL}}) + \frac{2L_{l,x} c_x}{\gamma} + \frac{c_\theta d_\theta}{\lambda} \\
&\overset{\text{Eq. (51)}}{\le} \frac{L_{l,x} L_\pi \|B\|_2}{\gamma^2} \cdot \lambda_\theta T + 2\mathfrak{R}_T^{\mathsf{seq}}(\ell_*^\Phi \circ \Pi_{\mathsf{NL}}) + \frac{6L_{l,x} c_x}{\gamma} + \frac{c_\theta d_\theta}{\lambda} \tag{52}
\end{aligned}
$$

where (i) follows from the fact that $\mathbb{E}[\sigma_i] = 1/\lambda$.

Following the proof technique of Corollary 8, it suffices to establish the stationary loss is bounded (by definition) and is Lipszhitz with respect to the underlying parameter (Lemma 5). Combining this with the fact that the parameter $\theta \in \mathbb{R}^d$, we have that the sequential Rademacher complexity is bounded as

$$\mathfrak{R}_T^{\mathsf{seq}}(\ell_*^\Phi \circ \Pi_{\mathsf{LQR}}) \le c\sqrt{d \cdot T \log(dTL_{\mathsf{Lip}})},$$

for some universal constant $c > 0$. Setting a value of $\lambda = 1/\sqrt{T}$ then conlcudes the proof of the corollary. $\qquad\square$

### D.5.1   Proof of Lemma 5

For any policies $\pi_1 := \pi_{\theta_1} \in \Pi_{\mathsf{LQR}}$ and $\pi_2 := \pi_{\theta_2} \in \Pi_{\mathsf{LQR}}$, and instance $z \in \mathcal{Z}$, consider the difference in the stationary loss

$$
\begin{aligned}
|\ell_*^\Phi(\pi_1, z) - \ell_*^\Phi(\pi_2, z)| &\le |\ell(\pi_1, x_*^{\pi_1}, z) - \ell(\pi_1, x_*^{\pi_2}, z)| + |\ell(\pi_1, x_*^{\pi_2}, z) - \ell(\pi_2, x_*^{\pi_2}, z)| \\
&\overset{(i)}{\le} L_{l,\theta} \|\theta_1 - \theta_2\|_2 + L_{l,x} \|x_*^{\pi_1} - x_*^{\pi_2}\|_2 \\
&\overset{(ii)}{\le} \left( L_{l,\theta} + L_{l,x} \frac{\|B\|_2 L_\pi}{\gamma} \right) \|\theta_1 - \theta_2\|_2,
\end{aligned}
$$

where inequality (i) follows from the Lipschitz property of the loss function with respect to the policy and state space while inequality (ii) follows from the Lipszhitz property of the policy. This establishes the deisred claim. $\qquad\square$

### D.6   Online LQR with adversarial disturbances

In this section, we consider the example of an online learning with dynamics problem where the adversary is allowed to perturb the dynamics in addition to the adversarial losses at each time step. We will focus on the Linear-Quadratic setup where the dynamics function is linear and the costs quadratic in the state $x_t$ and action $u_t$. Agarwal et al. [2] studied a general version of this problem where they considered the convex cost functions with linear dynamics.

As in the Online LQR example in Section D.3, we consider the class of linear policies $\Pi_{\mathsf{LQR}}$ which are $(\kappa, \gamma)$-strongly stable. Given this policy class, the online learning with dynamics game proceeds as follows, starting from state $x_0 = 0$

On round $t = 1, \ldots, T$,

- the learner selects a policy $K_t \in \Pi_{\mathsf{LQR}}$ and the adversary selects instance $z_t = (Q_t, R_t)$ such that $Q_t \succeq 0, R_t \succeq 0$ and $\mathrm{tr}(Q_t), \mathrm{tr}(R_t) \leq C$ and $\zeta_t$ such that $\|\zeta_t\|_2 \leq W$
- the learner receives loss $\ell(\pi_t, x_t, z_t) = x_t^\top Q_t x_t + u_t^\top R_t u_t$ where action $u_t = K_t x_t$
- the state of the system transitions to $x_{t+1} = Ax_t + Bu_t + \zeta_t$

where we assume that the transition matrices $A$ and $B$ are known to both the learner and adversary in advance. Observe that in this case, a stationary loss $\ell_*^\Phi$ does not exist because of the adversarial perturbations $\zeta_t$ in the dynamic; indeed, if a learner repeatedly plays the same policy $K \in \Pi_{\mathsf{LQR}}$, the state of the system is not guaranteed to converge to a unique stationary state. We now proceed to obtain an upper bound on the value $\mathcal{V}_{\mathsf{adv},\mathsf{T}}$ in the following corollary, by directly controlling the dynamic stability parameters $\{\beta_{\mathsf{RERM},t}\}$ for this policy class $\Pi_{\mathsf{LQR}}$ with a similar FTPL based regularized ERM as used in the proof of Corollary 8.

**Corollary 11** (LQR with adversarial disturbances). *For the online LQR with adversarial disturbances problem, the value $\mathcal{V}_{\mathsf{adv},\mathsf{T}}$ is bounded as*

$$\mathcal{V}_{\mathsf{adv},\mathsf{T}} \leq \mathcal{O}(\sqrt{T \log(T)}),$$

*where the $\mathcal{O}$ notation hides the dependence on problem-specific parameters.*

*Proof.* As we discussed above, the stationary losses $\ell_*^\Phi$ do not exist for this setup. Instead, we will work directly with the counterfactual losses $\ell^\Phi$ for this setup. Recall from Definition 2, the counterfactual loss at time $t$ for some linear policy $K \in \Pi_{\mathsf{LQR}}$ is defined as

$$\ell_t^\Phi(K_t, \zeta_{1:t}, z_t) = \ell(K_t, x_t[K_t^{(t-1)}, \zeta_{1:t-1}], z_t).$$

To instantiate the above counterfactual for the LQR problem, we will define some notation. Let us denote by $x_t = x_t[K_{1:t-1}, \zeta_{1:t-1}]$ the state at time reached by playing the sequence of policies $K_{1:t-1}$ and by $\tilde{x}_t = x_t[K_t^{(t-1)}, \zeta_{1:t-1}]$ the state when the learner plays polices $K_t$ for the first $t-1$ time steps. Further, let us denote by $X_t = x_t x_t^\top$ the rank 1 covariance matrix at time $t$ for state $x_t$ and similarly $\widetilde{X}_t = \tilde{x}_t \tilde{x}_t^\top$ for state $\tilde{x}_t$. With this notation, we have the losses

$$\ell_t(K_t, x[K_{1:t-1}, \zeta_{1:t-1}], z_t) = \mathrm{tr}((Q_t + K_t^\top R_t K_t) X_t) \quad \text{and} \quad \ell_t^\Phi(K_t, \zeta_{1:t-1}, z_t) = \mathrm{tr}((Q_t + K_t^\top R_t K_t)\widetilde{X}_t). \tag{53}$$

We now proceed to define the regularized ERM that we shall use and derive an upper bound on the dynamic stability parameters.

**Regularized ERM.** As in the proof of Corollary 8, we will consider the class of dual regularized ERM derived by the FTPL strategy

$$K_{t,\sigma} = \underset{K \in \Pi_{\mathsf{LQR}}}{\mathrm{argmin}} \left( \sum_{s=1}^{t} \underset{z_s \sim p_s}{\mathbb{E}} \left[ \langle Q_s + K^\top RK, \widetilde{X}_s \rangle \right] - \langle \sigma, K \rangle \right) \tag{54}$$

where $\sigma \in \mathbb{R}^{d \times k}$ such that each coordinate of $\sigma$ is sampled i.i.d. from the exponential distribution with parameter $\lambda > 0$. Following a similar argument as the one in the proof of Corollary 8, we have

$$\underset{\sigma}{\mathbb{E}}[\|K_{t,\sigma} - K_{t+1,\sigma}\|_1] \leq c\lambda \cdot L_{\mathsf{Lip}}(kd)^2 \kappa := \lambda_K, \tag{55}$$

where the constant $L_{\mathsf{Lip}}$ represents the Lipschitz constant of the function $\ell^\Phi$ (see Lemma 6).

**Dynamic stability parameters.** For any time $t > 0$ and the policies $\{K_t\}$ given by equation (54) (we drop the dependence on the noise $\sigma$) and any sequence of adversarial instances $(\zeta_{1:t}, z_{1:t})$, we have

$$|\ell_t^\Phi(K, \zeta_{1:t-1}, z_t) - \ell_t(K_t, x_t, z_t)| = |\langle Q_t + K_t^\top R_t K_t, X_t - \widetilde{X}_t \rangle|$$
$$\leq \mathrm{tr}(Q_t + K_t^\top R_t K_t) \cdot \|X_t - \widetilde{X}_t\|_2. \tag{56}$$

Thus, in order to obtain a bound on the dynamic stability parameters, we need to obtain a bound on the spectral norm of the difference $X_t - \widetilde{X}_t$. To do so, we begin by bounding the distance between

the states $x_t$ and $\tilde{x}_t$ as

$$
\begin{aligned}
x_t - \tilde{x}_t &= (A + BK_{t-1})x_{t-1} + \zeta_{t-1} - (A + BK_t)\tilde{x}_{t-1} - \zeta_{t-1} \\
&= (A + BK_t)(x_{t-1} - \tilde{x}_{t-1}) + B(K_{t-1} - K_t)x_{t-1} \\
&= (A + BK_t)^{t-1}(x_1 - \tilde{x}_1) + \sum_{s=2}^{t-1}(A + BK_t)^{t-s}B(K_s - K_t)x_s \,,
\end{aligned} \tag{57}
$$

where the final inequality follows by unrolling the recursion. Observe that the first term in the above equality is 0 since both $x_1 = \tilde{x}_1 = \zeta_1$. Taking the $\ell_2$ norm on both sides, we get,

$$
\begin{aligned}
\|x_t - \tilde{x}_t\|_2 &\overset{(i)}{\leq} C_x \sigma_B \kappa \lambda_K \sum_{s=2}^{t-1}(1-\gamma)^{t-s}(t-s) \\
&\overset{(ii)}{\leq} \frac{C_x \sigma_B \kappa \lambda_K}{\gamma} =: C_{x,2}
\end{aligned} \tag{58}
$$

where inequality (i) follows by using the fact that $K_t$ is $(\kappa, \gamma)$-strongly stable and the bound on the norm of the state $\|x_s\|_2 \leq \frac{\kappa W}{\gamma} := C_x$ and (ii) follows by summing up the series. With this bound, we obtain an expression for the difference between the covariances at time $t+1$ as

$$
\begin{aligned}
\widetilde{X}_{t+1} - X_{t+1} &= \left( \zeta_t((A + BK_{t+1})\tilde{x}_t)^\top + (A + BK_{t+1})\tilde{x}_t\zeta_t^\top + (A + BK_{t+1})\widetilde{X}_t(A + BK_{t+1})^\top \right) \\
&\quad - \left( \zeta_t((A + BK_t)x_t)^\top + (A + BK_t)x_t\zeta_t^\top + (A + BK_t)X_t(A + BK_t)^\top \right) \\
&= \underbrace{\zeta_t(\tilde{x}_t - x_t)^\top(A + BK_{t+1})^\top + (A + BK_{t+1})(\tilde{x}_t - x_t)\zeta_t^\top}_{\Delta_1^t} \\
&\quad + \underbrace{\zeta_t x_t^\top(K_{t+1} - K_t)^\top B^\top + B(K_{t+1} - K_t)x_t\zeta_t^\top}_{\Delta_2^t} \\
&\quad + \underbrace{B(K_{t+1} - K_t)X_t(A + BK_t)^\top + (A + BK_{t+1})X_t(B(K_{t+1} - K_t))^\top}_{\Delta_3^t} \\
&\quad + (A + BK_{t+1})(\widetilde{X}_t - X_t)(A + BK_{t+1})^\top.
\end{aligned}
$$

Let us denote by $\Delta_X^{t+1} := \widetilde{X}_{t+1} - X_{t+1}$ the difference between the covariance at time $t+1$ and by $\tilde{K}_{t+1} := A + BK_{t+1}$. With this notation, we can rewrite the above as

$$
\begin{aligned}
\Delta_X^{t+1} &= (\tilde{K}_{t+1})\Delta_X^t(\tilde{K}_{t+1})^\top + \sum_{i=1}^{3}\Delta_i^t \\
&= \tilde{K}_{t+1}^2\Delta_X^{t-1}(\tilde{K}_{t+1}^2)^\top + \tilde{K}_{t+1}\sum_{i=1}^{3}\Delta_i^{t-1}\tilde{K}_{t+1}^\top + \sum_{i=1}^{3}\Delta_i^t \\
&= \tilde{K}_{t+1}^t\Delta_X^1(\tilde{K}_{t+1}^t)^\top + \sum_{i=1}^{3}\sum_{s=2}^{t}\tilde{K}_{t+1}^{t-s+1}\Delta_i^s(\tilde{K}_{t+1}^{t-s+1})^\top,
\end{aligned} \tag{59}
$$

where in the last equality observe that $\Delta_X^1 = 0$. In order to bound the deviation $\|\Delta_X^{t+1}\|_2$, we will now bound each of three terms in the above equation separately.

**Bound for $\Delta_1$.** To obtain a bound on the term with the error $\Delta_1$, recall from equation (58) that we have $\|x_t - \tilde{x}_t\|_2 \leq C_{x,2}$. With this, we have,

$$
\begin{aligned}
\sum_{s=2}^{t}\tilde{K}_{t+1}^{t-s+1}\Delta_1^s(\tilde{K}_{t+1}^{t-s+1})^\top &\leq 2W C_{x,2}\kappa^3 \cdot \sum_{s=2}^{t}(1-\gamma)^{2(t-s+1)} \\
&\leq \frac{2W\kappa^3}{\gamma} \cdot C_{x,2} \,.
\end{aligned} \tag{60}
$$

**Bound for $\Delta_2$.** For the error term corresponding to $\Delta_2$, recall that we have $\|K_{t+1} - K_s\|_2 \leq (t+1-s) \cdot \lambda_K$. Substituting this in the error term, we have,

$$\sum_{s=2}^{t} \tilde{K}_{t+1}^{t-s+1} \Delta_2^s (\tilde{K}_{t+1}^{t-s+1})^\top \leq 2\sigma_B W C_x \lambda_K \kappa^2 \sum_{s=2}^{t} (1-\gamma)^{2(t-s+1)}(t-s+1)$$

$$\leq \frac{2\sigma_B W C_x \kappa^2}{\gamma^2} \cdot \lambda_K . \tag{61}$$

**Bound for $\Delta_3$.** Finally, for the error term $\Delta_3$, observe that the spectral norm of the covariance $\|X_t\|_2 \leq C_x^2$, and substituting this in the sum, we have

$$\sum_{s=2}^{t} \tilde{K}_{t+1}^{t-s+1} \Delta_3^s (\tilde{K}_{t+1}^{t-s+1})^\top \leq 2C_x^2 \sigma_B \kappa^3 \lambda_K \sum_{s=2}^{t} (1-\gamma)^{2(t-s+1)}(t-s+1)$$

$$\leq \frac{C_x^2 \sigma_B \kappa^3}{\gamma^2} \cdot \lambda_K . \tag{62}$$

Substituting the bounds obtained in equations (60), (61) and (62) in the upper bound on the stability parameters in equation (56), we have that the dynamic stability parameters $\beta_{\mathsf{RERM},t} = c_\beta \lambda_K$, where the constant $c_\beta$ depends on problem dependent parameters and can be obtained from the above equations. Having established a bound on the dynamic stability, we now upper bound the value for this problem.

**Bound on the value.** The value of the online LQR with adversarial disturbance problem is

$$\mathcal{V}_{\mathsf{adv},\mathsf{T}}(\Pi_{\mathsf{LQR}}, \mathcal{Z}, \Phi, \ell) \overset{(i)}{\leq} \sum_{t=1}^{T} \beta_{\mathsf{RERM},t} + 2\mathfrak{R}_T^{\mathsf{seq}}(\ell^\Phi \circ \Pi_{\mathsf{LQR}}) + \frac{\kappa k d}{\lambda}$$

$$\overset{(ii)}{\leq} c_\beta \lambda_K \cdot T + 2\mathfrak{R}_T^{\mathsf{seq}}(\ell^\Phi \circ \Pi_{\mathsf{LQR}}) + \frac{\kappa k d}{\lambda} , \tag{63}$$

where (i) follows from the fact that $\mathbb{E}[\sigma_i] = 1/\lambda$ and (ii) follows from noting that each of the dynamic stability parameters is upper bounded by $c_\beta \lambda_K$.

To obtain a bound on the sequential Rademacher complexity of the class, observe that by Lemma 6, we have that the loss $\ell^\Phi$ is bounded by $B_{\mathsf{max}}$ and Lipschitz with respect to policies $K$ with constant $L_{\mathsf{Lip}}$. Using a standard covering number argument, one can get an $\epsilon$-net of the class $\Pi_{\mathsf{LQR}}$ in the frobenius norm with atmost $O(dk(\frac{1}{\epsilon})^{dk})$ elements. Given this cover, one can upper bound the complexity as

$$\mathfrak{R}_T^{\mathsf{seq}}(\ell_*^\Phi \circ \Pi_{\mathsf{LQR}}) \leq c B_{\mathsf{max}} \sqrt{kd \cdot T \log(kdTL_{\mathsf{Lip}})}$$

for some universal constant $c > 0$. Setting $\lambda = O(1/\sqrt{T})$ concludes the proof of the corollary. $\qquad\square$

**Lemma 6.** *For the counterfactual loss $\ell_t^\Phi(K, \zeta_{1:t-1}, z_t)$ defined in equation (53) and policy class $\Pi_{\mathsf{LQR}}$, we have*

a $\ell_t^\Phi$ *is bounded by* $|\ell^\Phi(K, \zeta_{1:t-1,z_t})| \leq C(1+\kappa^2) \cdot \left(\frac{\kappa W}{\gamma}\right)^2 := B_{\mathsf{max}}.$

b $\ell_t^\Phi$ *is Lipschitz with respect to $K$ with*

$$|\ell_t^\Phi(K_1) - \ell_t^\Phi(K_2)| \leq C\left(2C_x^2\kappa + (\kappa^2+1)\left(\frac{2\kappa^4 C_x \sigma_B W}{\gamma^2} + \frac{\kappa^3 C_x^2 \sigma_B}{\gamma}\right)\right) \cdot \|K_1 - K_2\|_2.$$

*Proof.* We will establish both the parts separately.

**Proof for part (a).** Consider the counterfactual loss $\ell_t^\Phi$ at time $t$

$$|\ell^\Phi(K, \zeta_{1:t-1}, z_t)| = \langle Q_t + K^\top R_t K, \tilde{x}_t \tilde{x}_t^\top \rangle$$

$$\leq \|Q_t + K^\top R_t K\|_2 \cdot \|\tilde{x}_t\|_2^2$$

$$\leq C(1+\kappa^2) \cdot \left(\frac{\kappa W}{\gamma}\right)^2 , \tag{64}$$

where the last inequality follows by using the fact that the policy $K$ is $(\kappa, \gamma)$-strongly stable and that $\|x\|_2 \leq C_x := \frac{\kappa W}{\gamma}$.

**Proof for part (b).** Consider any two linear policies $K_1, K_2 \in \Pi_{\mathsf{LQR}}$. The difference in the counterfactual losses is given by

$$|\ell_t^\Phi(K_1, \zeta_{1:t-1}, z_t) - \ell_t^\Phi(K_2, \zeta_{1:t-1}, z_t)| = |\langle Q + K_1^\top R K_1, \widetilde{X}_1 \rangle - \langle Q + K_2^\top R K_2, \widetilde{X}_2 \rangle|$$

$$\leq C(\kappa^2 + 1)\|\widetilde{X}_{1,t} - \widetilde{X}_{2,t}\|_2 + 2C_x^2 C\kappa \|K_1 - K_2\|_2, \quad (65)$$

where we have used the notation $\widetilde{X}_t = \tilde{x}_t \tilde{x}_t^\top$ to denote the covariance at time $t$ and the final inequality follows by noting that $\mathrm{tr}(Q), \mathrm{tr}\, R \leq C$ and $\|K_i\|_2 \leq \kappa$ for $i = \{1, 2\}$. Let us denote by $\tilde{K} = A + BK$ the effective state transisiotn matrix. We now focus on the term corresponding to the difference of the covariances $\widetilde{X}_{1,t} - \widetilde{X}_{2,t}$.

$$\widetilde{X}_{1,t} - \widetilde{X}_{2,t} = \left( \tilde{K}_1 \tilde{x}_{1,t-1} \zeta_{t-1}^\top + \zeta_{t-1} \tilde{x}_{1,t-1}^\top \tilde{K}_1^\top + \tilde{K}_1 \widetilde{X}_{1,t-1} \tilde{K}_1^\top \right) - \left( \tilde{K}_2 \tilde{x}_{2,t-1} \zeta_{t-1}^\top + \zeta_{t-1} \tilde{x}_{2,t-1}^\top \tilde{K}_2^\top + \tilde{K}_2 \widetilde{X}_{2,t-1} \tilde{K}_2^\top \right)$$

$$= \underbrace{(\tilde{K}_1 - \tilde{K}_2) \tilde{x}_{1,t-1} \zeta_{t-1}^\top + \zeta_{t-1} \tilde{x}_{1,t-1}^\top (\tilde{K}_1 - \tilde{K}_2)^\top}_{\Delta_{1,t-1}} + \underbrace{\tilde{K}_2 (\tilde{x}_{1,t-1} - \tilde{x}_{2,t-1}) \zeta_{t-1}^\top + \zeta_{t-1} (\tilde{x}_{1,t-1} - \tilde{x}_{2,t-1})^\top \tilde{K}_2^\top}_{\Delta_{2,t-1}}$$

$$+ \underbrace{(\tilde{K}_1 - \tilde{K}_2) \widetilde{X}_{1,t-1} \tilde{K}_1^\top + \tilde{K}_2 \widetilde{X}_{1,t-1} (\tilde{K}_1 - \tilde{K}_2)^\top}_{\Delta_{3,t-1}} + \tilde{K}_2 (\widetilde{X}_{1,t-1} - \widetilde{X}_{2,t-1}) \tilde{K}_2^\top$$

$$= \sum_{i=1}^3 \sum_{s=2}^{t-1} \tilde{K}_2^{t-1+s} \Delta_{i,s} (\tilde{K}_2^{t-1+s})^\top. \quad (66)$$

In order to show establish the Lipschitzness of the loss function $\ell^\Phi$, we will obtain a bound on each of the three error terms comprising $\Delta_i$ separately now.

Bound on $\Delta_1$. For the term corresponding to $\Delta_1$, observe that both the state $\tilde{x}$ and the disturbance $\zeta$ are bounded vectors. Using the $(\kappa, \gamma)$-strong stability of the policy $K_1$, we have

$$\left\| \sum_{s=2}^{t-1} \tilde{K}_2^{t-1+s} \Delta_{1,s} (\tilde{K}_2^{t-1+s})^\top \right\|_2 \leq \frac{\kappa^2 C_x \sigma_B W}{\gamma} \cdot \|K_1 - K_2\|_2. \quad (67)$$

Bound on $\Delta_2$. For the second term, observe that

$$\|\tilde{x}_{1,t} - \tilde{x}_{2,t}\|_2 \leq \frac{\kappa \sigma_B C_x}{\gamma} \cdot \|K_1 - K_2\|_2.$$

With this, we can bound the second term in equation (66) as

$$\left\| \sum_{s=2}^{t-1} \tilde{K}_2^{t-1+s} \Delta_{2,s} (\tilde{K}_2^{t-1+s})^\top \right\|_2 \leq \frac{\kappa^4 C_x \sigma_B W}{\gamma^2} \cdot \|K_1 - K_2\|_2. \quad (68)$$

Bound on $\Delta_3$. For the final term, note that $\|\widetilde{X}_t\|_2 \leq C_x^2$. With this, we can bound the term corresponding to $\Delta_3$ as

$$\left\| \sum_{s=2}^{t-1} \tilde{K}_2^{t-1+s} \Delta_{2,s} (\tilde{K}_2^{t-1+s})^\top \right\|_2 \leq \frac{\kappa^3 C_x^2 \sigma_B}{\gamma} \cdot \|K_1 - K_2\|_2. \quad (69)$$

Combining the bounds obtained in equations (67), (68) and (69), with the upper bound in equation (66) establishes the desired claim. $\qquad \square$