[Reviews · NeurIPS 2020]

Review 1

Summary and Contributions: The paper studies a learning problem in a game between a learner and an adversary in a dynamic environment. The actions of the adversary influences the loss of the learner and also the dynamics of the environments making the state-evolution dynamics potentially adversarial. The goal of the learner is to minimise the 'policy regret' expressed as the difference of the loss from a policy and that incurred by the best 'counterfactual loss' (that uses fixed policy in every round). The authors (aim to) provide sufficient conditions for the learnability of the problem and give non-constructive upper bounds on the minimax rate. The upper bounds are expressed in terms of the complexity o f the expressiveness (Rademacher Complexity) of the policy class and dynamic mixability property of the regularised Empirical Risk Minimisers. The authors establish lower bound for the problem to show that their bounds are tights up to constants. The authors recover the regret bounds of many known setups from analysis of their general setup.

Strengths: 1. The problem of online learning in a dynamic environment is interesting. 2. The authors establish the optimality of the upper bound with a matching lower bound 3. Many of the earlier results are recovered with the analysis of their general setup

Weaknesses: 1. Main claim of the paper is development of sufficient conditions for learnability. What is this condition. I did not find explicit mention of the conditions in the paper 2. In Proposition 1, authors say "Let $\mathcal{Q}$ and $\mathcal{P}$..., satisfy necessary conditions for the minimax theorem to hold". For what type of sets the hypothesis holds? What restrictions this put on the setup is unclear. No discussion on this. 3. The authors say "We provide lower bounds that show that our sufficient conditions are tight for a large class of problem instances showing that both the terms in our upper bounds are indeed necessary." No characterisation of this large class of problem is given. Update: I read the response of the authors. I am still not convinced that sufficient conditions are clear. In the response, authors say "whenever sequential Rademacher complexity of the underlying policy class _x0005_ as well as the dynamic stability 6 parameters are o(T), the corresponding problem instance is learnable." But this clarity was not at all there in the earlier draft as admitted by the authors. Since the regret is bounded as the sum of Rademacher complexity and dynamic stability, it is obvious that they have to be o(T) for the problem to be learnable. What will be interesting and relevant is when these terms are o(T). I don't see conditions for this being stated in the paper. I feel this lack of clarity makes the paper weak.

Correctness: Not sure about are the the sufficient conditions the authors have claimed. The reviewer couldn't verify all the proofs

Clarity: The paper is well written to a large extent. However, the the paper should clearly state all the conditions explicitly for better readability.

Relation to Prior Work: The work builds on [22]. The authors mentions that tools of [22] are useful but they are not by themselves sufficient. They needed to account for dynamic stability parameter in the new analysis. The reviewer could not verify how challenging it was to incorporate them in the bounds given the assumptions that \{\beta_t\} parameters are known.

Reproducibility: Yes

Additional Feedback: The paper can benefit if following points are clarified further. 1. As a motivation for the setup it is said that "a vast majority of these works have focused on known and fixed models of state evolution, often restricting the scope to linear dynamical systems." Is it not that the authors also considers known state evolution? 2. In Definition 1, the meaning of \mathbf{\xi}_i(\epsilon) is not defined. 3. Instead of giving multiple example in Section 5, one example with more detail would help appreciate the general setup.


Review 2

Summary and Contributions: The paper studies the problem of "online learning with dynamics" and the relevant "policy regret" benchmark. To briefly describe this problem, we first recall that in the standard notion of regret, it is the difference of the cost suffered up to time T, minus the minimum cost suffered if the learner sticks to a certain strategy up to time T, *assuming that the cost and environment is not affected by the switch to sticking with the optimal strategy*. Policy regret can, informally, be said to the a variant of regret, but with that assumption removed. Instead, the cost and environment now can depend on the policy choice. In this paper, this dependence is made explicitly via a dynamical system. The paper considers a very general setting, and upper bound the optimal policy regret using something called "sequential Rademacher Complexity" (Def. 1) and Dynamic Stability (Def. 2). The main result is stated in Theorem 1. The authors also provide somewhat matching lower bounds in Theorem 2. The above very general result is then applied to various more specific (and well-known) problems, including online Markov Decision Process. The authors show that under this general framework, they can recover many optimal policy regret bounds for those specific problems.

Strengths: Here I am assuming that the proofs are correct (which, given the horizons of prerequisites required to understand the paper and the short review time given, I could not verify). Since the paper provides concrete results regarding very general settings of bounding policy regrets in "online learning with dynamics", which is a fairly fundamental problem in its own right, the results sound to be a milestone and are so important that it might be included into a textbook about the topic. So I have not doubt that the paper should be accepted if the proofs are correct.

Weaknesses: None as I spotted.

Correctness: I cannot verify since the time given for review is short, and the analyses (which are completely in the appendix) appear very involved.

Clarity: Nice, but I will appreciate if the authors can explain more about the key concepts (like Rademacher complexity and its sequential counterpart, dynamic stability). But I understand this is unlikely feasible due to page limit.

Relation to Prior Work: Yes.

Reproducibility: Yes

Additional Feedback: Below are some questions which I hope the authors can answer, so that I can have more confidence on the results of the paper. 1) As the authors point out in Line 226, the two key terms in the upper bound are Term I (that concerns dynamic stability and Term II (that concerns sequential Rademacher complexity). Can you provide an intuitive answer why for the three examples in Section 5, these two terms can be bounded by O(\sqrt{T}). To be more specific, is there any underlying property about the loss function that helps to explain the O(\sqrt{T}) bounds are valid? 2) There is a third term in Theorem 1 that is a static term that concerns the supremum regularizer value. From my experience the supremum over all policies can be infinite, and then Theorem 1 will be useless. Do you have any counter argument against this? ====== Other minor comment(s): 3) "Policy" is not formally defined in this paper. In particular, policy is typically a function that takes an input and outputs a choice. So what is the input in your setting? 4) The abstract should mention the keyword "Rademacher complexity". ====== Post-rebuttal: I am satisfied with the rebuttal. I am interested to know more about "One reason why such rates are common in online learning is the connection of the sequential Rademacher complexity with uniform convergence of martingale difference sequences in the corresponding Banach space (see [2] for details)." If the paper is accepted and space is allowed, I suggest to elaborate on this more thoroughly.


Review 3

Summary and Contributions: In this paper, the authors study the problem of online learning with dynamics, where the notion of policy regret is used as the measure. Authors analyze the policy regret in the minimax perspective and offer the upper bound for learnability, which consists of two terms: the stability term of ERM algorithm and the sequential Rademacher complexity term in the stateful environment. Meanwhile, the authors also offer the low bounds and validate the tightness of the upper bound in the instance-dependent scenario. Finally, authors give some applications of the proposed upper bounds, which can recover or even obtain tighter bound than previous studies. The main contribution of the paper is to introduce a novel and more general analytic framework for online learning with dynamics. The proposed upper bound can be applied for arbitrary online dynamical systems, which is not investigated in previous studies.

Strengths: The main advantage of this paper is that author propose a general analytic framework for online learning with dynamics. The proposed upper bound is very interesting, as it states the learnability in a fully general scenario: for any losses and general functions classes. Meanwhile, there is no restrictions on linear dynamic systems. This upper bound is shown to be tight and can recovers any previous studies. I think it is worthy to be known for the community.

Weaknesses: My main question is that is it possible to obtain the lower bound in terms of the stability of ERM algorithm (not the mini-batch ERM) and the sequential Rademacher complexity, which can better match the upper bound.

Correctness: The claim seems correct, but I do not go over the detailed proof.

Clarity: This paper is well written with a clear structure.

Relation to Prior Work: The work discusses the previous work from two aspects. The first is from the online learning with dynamics, where previous works require the constraints on linearized dynamic systems, fixed and known models of state evolution, or simplistic policy classes. By contrast, this work offers a general framework. Authors also discuss the learnability in classical online learning. It is clear that this work is different from the previous contributions.

Reproducibility: Yes

Additional Feedback: As mentioned in the weakness part, the question is that is it possible to obtain the lower bound in terms of \beta_{ERM} plus sequential Rademacher complexity? Meanwhile, there is a minor typo. Line 253: equation 2 --> equation (2) =======Post rebuttal session========= I remain my score after reading the other reviews , author feedback, and reviewer discussions.


Review 4

Summary and Contributions: The authors consider the problem of online learning with an environmental state which is affected by the learner's actions and affects the loss of each action/instance pair. The authors analyse the learnability of the game by deriving upper and lower bounds on the "value" of the game. The authors then give several special cases of this game that appear in the literature.

Strengths: The upper and lower bounds on the value of the game appear novel and the game unifies many problems that appear in the literature. They use their general upper bound to give upper bounds for the value of these existing games.

Weaknesses: This paper presents no algorithm for the problem (and certainly not an efficient one). All they have is bounds on the existence of an algorithm.

Correctness: As far as I can see the claims and method are correct. However, I have not read the proofs in the appendix.

Clarity: I found this paper confusing in that the discussion was not clear enough. I found the following confusions: 1) Line 62. It should be made clear that Z is the product of two sets. Also it should be made clear the x_t is a state (i.e. x_t\in X). 2) Line 124. It is said that the (additive) noise has zero mean. The restricts the set of states, X, to be a subset of a real vector space. The fact that the X is a subset of a real vector space should be made clear when X is defined. 3) Equation 3. The double brackets are not defined - I had to look at the proof in the appendix to figure out what they meant.

Relation to Prior Work: This is discussed.

Reproducibility: Yes

Additional Feedback: After rebuttal: I have incorporated the author feedback into the review. After viewing the other reviewers comments I have increased by score to 6 but have low confidence. The confidence is low because I have based my score on other reviewers' opinions.

[Author Response · NeurIPS 2020]

We thank the reviewers for their thoughtful feedback. We are pleased that they found the online learning with dynamics
problem interesting and appreciated the generality of our analysis. Below, we provide clarifications to their concerns.

**Reviewer R1.** – (**Sufficient conditions for learnability**) As noted on Line 72, an online learning with dynamics
problem is said to be learnable if the policy regret $\mathrm{Reg}_T^{\mathsf{pol}} = o(T)$. Our main results in Theorem 1 and Proposition 2
show that whenever sequential Rademacher complexity of the underlying policy class $\Pi$ as well as the dynamic stability
parameters are $o(T)$, the corresponding problem instance is learnable. These comprise the sufficient conditions for
learnability for the online learning with dynamics problem; we shall make this connection clearer in our updated draft.
– (**Lower bounds**) While the lower bound in Theorem 2 shows the existence of a particular instance, the results of
Proposition 3 hold for a broad range of problem instances – given any choice of decision functions $\mathcal{F}$, adversary
instance $\mathcal{Z}$ and loss $\ell$, there exists an instance with policy class $\Pi_{\mathcal{F}}$ and dynamics $\Phi$ such that the upper bounds on
regret are tight. These comprise the broad class of instances for which our learnability characterization is tight.
– (**Conditions for Minimax theorem**) Our application of the minimax theorem requires the space of probability
distributions $\mathcal{P}$ and $\mathcal{Q}$ to be weakly-compact (see [1, Appendix A] for a detailed discussion). Such an assumption is
fairly general – A simple corollary of Prokhorov's theorem establishes that for *any* compact metric space $(\mathcal{X}, d)$, the
metric space $(\mathcal{P}(\mathcal{X}), d_P)$ is compact where $d_P$ is the Prokhorov metric. We will elaborate further in our revised draft.
– (**Comparison with [22]**) We would like to highlight that both our problem definition as well as our techniques for
obtaining the upper bounds generalize those of [22]. The presence of an underlying dynamics adds an additional
complexity to the problem which requires the dual analysis to balance loss minimization and dynamic stability. This
creates a strong coupling between the strategies at any time $t$ with the future strategies - such a coupling and the
associated difficulties are absent in the dual analysis for the classical online learning framework. Indeed, the greedy dual
algorithm used in the analysis by [22] is dynamically unstable and would lead to $O(T)$ policy regret in our framework.
– (**Dynamics function**) We would like to highlight that the learner having access to the function $\Phi$ does not imply
that it knows the underlying dynamics. Indeed, one can encode all the aspects of the dynamics in the variable $\zeta_t$ and
$\Phi$ then acts as an applicator of these dynamics. Furthermore, recall that the learner gets to observe $\zeta_t$ only at time
$t$. For example, consider linear dynamical systems parameterized by matrices $(A, B)$. By letting $\zeta_t = (A_t, B_t)$, our
framework can capture time varying linear dynamical systems with the learner observing these matrices at time $t$.
– (**Examples**) We chose to highlight several different examples in Section 5 to demonstrate the generality of our results
and show that it recovers regret bounds for a wide class of problems, each of which is an individual research paper.

**Reviewer R2.** – (**Regularization term**) The boundedness for a large class of regularizers follows from the boundedness
of the underlying policy class – for e.g., policy classes with parameter belonging to a compact domain. Further, several
well-studied regularization functions (e.g. entropic penalty) are themselves bounded and do not affect learnability.
– ($\mathbf{T^{\frac{1}{2}}}$**-regret for examples**) There are definitely example instances in our framework wherein the regret is not $T^{\frac{1}{2}}$; for
instance, our lower bound example construction (Line 295) has regret scaling as $T^{\frac{2}{3}}$. However, since our focus in the
examples section is on recovering regret bounds for classical problems, we recover rates scaling as $T^{\frac{1}{2}}$. One reason
why such rates are common in online learning is the connection of the sequential Rademacher complexity with uniform
convergence of martingale difference sequences in the corresponding Banach space (see [2] for details).
–(**Policy Def.**) A policy is a mapping from state space $\mathcal{X}$ to a distribution over actions; we will include this in our draft.

**Reviewer R3.** – (**Lower bounds**) We would like to highlight that our lower bounds in Thm 2 (eq. (7a) and (7b)) can
actually be combined to obtain one which matches the upper bound of Thm 1 – we will update this in our draft. Given
this, we believe that obtaining such a lower bound in the general setup of Prop. 3(b) is an interesting future direction.

**Reviewer R4.** – (**Clarification regarding dual game**) We would like to clarify that the RERM we consider in the
proof of Theorem 1 is indeed for the dual game – it is straightforward to observe from equation 3 (Line 217) that the
policy at time $t$ involves an expectation with respect to $z_t \sim p_t$, which is clearly not available in the primal game.
Further, it is a well-known fact that primal ERM (which is effectively a follow-the-leader algorithm) does not guarantee
sub-linear regret in the classical online learning setup with no dynamics and our paper definitely does not propose it as
a solution to the online learning with dynamics problem.
– (**Non-algorithmic framework**) We disagree with the assessment that a non-algorithmic complexity theoretic approach
to studying such a fundamental problem is not a sufficient contribution. There have been decades of work in the
field of learning theory and statistical machine learning which have studied fundamental problems (like classification,
regression) from a complexity perspective leading to characterization of learnability in the form of VC dimension,
Littlestone dimension, Rademacher complexity amongst others. We believe that the study of fundamental questions
concerning learnability is the first step towards systematic algorithmic approaches for these classes of problems and is
therefore an important contribution in the learning community.

[1] A. Rakhlin, K.Sridharan, and A. Tewari. Online learning via sequential complexities. JMLR 2015.

[2] A. Rakhlin, K. Sridharan, and A. Tewari. Sequential complexities and uniform martingale laws of large numbers. PTRF 2015.


[Meta-Review · NeurIPS 2020]

Nice theoretical contribution to an interesting and relevant topic for the community. One of the main issues of this paper is clarity, however. We trust that you will work very hard to improve clarity for the final submission, as suggested in the reviews.